# Annual variation in event-scale precipitation $\delta^2 H$ at Barrow, AK reflects vapor source region

Annie L. Putman[1,2], Xiahong Feng[1], Leslie J. Sonder[1], and Eric S. Posmentier[1]

[1]Department of Earth Sciences, Dartmouth College, Hanover, NH, USA.
[2]Department of Geology & Geophysics, University of Utah, Salt Lake City, UT, USA.

*Correspondence to:* Annie Putman (putmanannie@gmail.com)

**Abstract.** In this study, precipitation isotopic variations at Barrow, AK, USA are linked to conditions at the moisture source region, along the transport path, and at the precipitation site. Seventy precipitation events between January 2009 and March 2013 were analyzed for $\delta^2 H$ and deuterium excess. For each precipitation event, vapor source regions were identified with the Lagrangian air parcel tracking program, HYSPLIT, in back-cast mode. The results show that the vapor source region
migrated annually, with the most distal (proximal) and southerly (northerly) vapor source regions occurring during the winter (summer). This may be related to equatorial expansion and poleward contraction of the Polar circulation cell and the extent of Arctic sea ice cover. Annual cycles of vapor source region latitude and $\delta^2 H$ in precipitation were in phase; depleted (enriched) $\delta^2 H$ values were associated with winter (summer) and distal (proximal) vapor source regions. Precipitation $\delta^2 H$ responded to variation in vapor source region as reflected by significant correlations between $\delta^2 H$ with the following three parameters: 1)
total cooling between lifted condensation level and precipitating cloud at Barrow, $\Delta \bar{T}_{cool}$, 2) meteorological conditions at the evaporation site quantified by 2 m dew point, $\bar{T}_d$, and 3) whether the vapor transport path crossed the Brooks and/or Alaskan ranges, expressed as a Boolean variable, $mtn$. These three variables explained 54 % of the variance (p <0.001) in precipitation $\delta^2 H$ with a sensitivity of -3.51 $\pm$ 0.55 ‰ °C$^{-1}$ (p < 0.001) to $\Delta \bar{T}_{cool}$, 3.23 $\pm$ 0.83 ‰ °C$^{-1}$ (p < 0.001) to T$_d$, and -32.11 $\pm$ 11.04 ‰ (p = 0.0049) depletion when $mtn$ is true. The magnitude of each effect on isotopic composition also varied with
vapor source region proximity. For storms with proximal vapor source regions (where $\Delta \bar{T}_{cool}$ <7 °C), $\Delta \bar{T}_{cool}$ explained 3 % of the variance in $\delta^2 H$, $\bar{T}_d$ alone accounted for 43 %, while $mtn$ explained 2 %. For storms with distal vapor sources ($\Delta \bar{T}_{cool}$ >7°C), $\Delta \bar{T}_{cool}$ explained 22 %, $\bar{T}_d$ explained only 1 %, and $mtn$ explained 18 %. The deuterium excess annual cycle lagged by 2-3 months the $\delta^2 H$ cycle, so the direct correlation between the two variables is weak. Vapor source region relative humidity with respect to the sea surface temperature, $\bar{h}_{ss}$, explained 34 % of variance in deuterium excess, (-0.395 $\pm$ 0.067 ‰ % $^{-1}$,
p < 0.001). The patterns in our data suggest that on an annual scale, isotopic ratios of precipitation at Barrow may respond to changes in the southerly extent of the Polar circulation cell, a relationship that may be applicable to interpretation of long term climate change records like ice cores.

# 1 Introduction

Changes to spatial patterns of water vapor transport and precipitation are an important component of incipient climate change (Santer et al., 2007; Marvel and Bonfils, 2013). The Arctic exhibits a particularly strong hydrologic response, including a notable increase in Arctic precipitation (Min et al., 2008; Bintanja and Selten, 2014; Kopec et al., 2016). Current and future changes

in the hydrologic cycle may impact fresh water resources, natural disasters, and earth's radiation balance, due to changes in timing, extent, and duration of snow or cloud cover (Liu et al., 2012).

Like changes in the timing or amount of precipitation, changes in the relative abundance of heavy-isotope substituted water molecules in precipitation (e.g., $^1H_2^{16}O$ vs. $^1H_2^{18}O$ and $^1H^2H^{16}O$) may reflect effects of changing climate on the hydrologic cycle. Historically, researchers have measured the isotopic ratios of precipitation on monthly or longer timescales and attempted

to explain isotopic variations over time, altitude, and latitude (Rindsberger et al., 1983; Cappa et al., 2003; Liu et al., 2010). Empirical analysis has focused on weather and climate conditions at the precipitation site (Dansgaard, 1964). Models developed to understand the spatial and temporal variability of water stable isotopes include evaporation and Rayleigh distillation models (Merlivat and Jouzel, 1979; Jouzel and Merlivat, 1984), models examining the balance of vertical mixing and meridional advection (Hendricks et al., 2000; Noone, 2008), and isotope-enabled general circulation models (GCMs) (e.g., Jouzel

et al., 1987; Yoshimura et al., 2008; Dee et al., 2015).

Variation in condensation temperatures and sub-cloud humidity has been shown to explain substantial variation in the measured isotopic ratios of precipitation (Aemisegger et al., 2015; Stewart, 1975) over short timescales. Until recently, few isotope models have considered meteorological conditions at the vapor source, in part because the evaporation site could not be unambiguously identified. Not knowing the vapor source prevents comprehensive examination of the full vapor history. Recently

developed Lagrangian air parcel tracking programs with quantitative source and trajectory meteorology have enabled estimation of evaporation sites and thus have become a useful tool for interpreting precipitation isotope ratios (Ichiyanagi and Yamanaka, 2005; Treble et al., 2005; Strong et al., 2007; Sodemann et al., 2008a; Wang et al., 2013; Good et al., 2014).

The objective of this study is to understand how source and trajectory meteorology contributes to event-scale variations in the precipitation isotopic ratios and how such contributions vary over time (e.g., seasonally). To do this, we investigate the

isotopic ratios of precipitation from event-scale sampling at Barrow, Alaska, USA. Barrow is one of nine sites that comprise the pan-Arctic Isotopic Investigation of Sea Ice and Precipitation in the Arctic Climate System campaign (iisPACS, (Feng, 2011)) The work presented here uses intensive observations at Barrow under the Atmospheric Radiation Monitoring (ARM) program. Specifically, we use millimeter wavelength cloud radar (MMCR) to identify the altitude and rate of condensation in the precipitating clouds in order to initialize Lagrangian air parcel tracking. Using direct cloud observations means that

the backward trajectories are initiated at the appropriate time and from a distribution of altitudes representative of the actual heights of condensation. Such an initial distribution of air parcels is unique to our study. We distribute air parcels in proportion to the condensation rate, so that for a given event, each air parcel represents an equal fraction of precipitated water (Putman, 2013). This simplifies calculating the average vapor source, transport, and condensation conditions, which we use to interpret the observed precipitation isotope ratios. Although this research focuses on precipitation data from a single location, the results

may indicate a link between atmospheric circulation and precipitation isotope systematics across the sea-ice-sensitive high latitudes.

## 2  Methods

Event-scale precipitation samples were collected from 70 precipitation events at Barrow, AK, between January 2009 and April
2013. Below we describe methods for sample collection and measurement of $\delta^2 H$ and $\delta^{18}O$ of precipitation, identification of vapor source regions, and characterization of evaporation and transport conditions using meteorological data from the source regions.

### 2.1  Sample collection and isotopic analysis

The sampling equipment was installed on a skydeck within the North Slope of Alaska facility of the Atmospheric Radiation
Measurement (ARM) program. If the precipitation was rain, a rain funnel was used to collect the sample. If the precipitation was snow, the fresh snow was scooped into a plastic bag from a designated surface on the skydeck. The collection surface was five meters above the ground on the tower, ensuring minimal contribution of windblown snow from previous events. Samples were gathered less than 24 hours after the event ended and often as soon as snow ended. Though it is possible that snow may have been altered by sublimation before collection, we assume that the degree of alteration of surface snow was
minimal relative to the amount of snow gathered. Furthermore, the frequent cloudiness and darkness of Barrow mean that for most events, sunlight-driven sublimation was insignificant. Liquid samples were stored in tightly sealed 30 mL Nalgene bottles below 5 °C and shipped in batches every three months to the Stable Isotope Laboratory at Dartmouth College. When not in transit, samples were refrigerated. Samples were analyzed within six months of collection.

Upon arrival at Dartmouth the samples were prepared for analysis of hydrogen and oxygen isotopic ratios with a Delta Plus
XL Isotope Ratio Mass Spectrometer (IRMS). For hydrogen measurements, the IRMS was connected to an HDevice reduction furnace: a reactor tube filled with a volumetric 1:1 mix of 100 mesh and 300 mesh chromium powder and set at 850 °C. One μL of sample was injected into the HDevice, and the water was allowed to react for two minutes in the hot chromium chamber, reducing to hydrogen gas, which was then introduced to the dual inlet system of the mass spectrometer and measured by the IRMS. For oxygen isotope measurements, the IRMS was coupled to a GasBench. A 500 μL aliquot of liquid sample was
placed in a vial, flushed with a mixture of 0.3 % $CO_2$ in helium, and allowed to equilibrate for at least 18 hours at 25 °C. The isotopic ratios of the $CO_2$ were measured by the IRMS. For both the oxygen and hydrogen measurements, the measured value was converted to the water-isotope equivalent by calibration with known standards. Isotopic ratios ($^2$H/$^1$H and $^{18}$O/$^{16}$O), are reported in delta notation: the per mil (‰) deviation from the international standard VSMOW on the VSMOW-SLAP scale, defined as $\delta = [\dfrac{R_{SA} - R_{ST}}{R_{ST}}]$, where $R_{SA\,or\,ST} = \dfrac{^2[H]}{^1[H]}$ or $\dfrac{[^{18}O]}{[^{16}O]}$. SA and ST indicate sample and standard, respectively. The
uncertainties of the reported values are within $\pm\,0.5$ and $\pm\,0.1$ ‰ (one standard error) for $\delta^2 H$ and $\delta^{18}O$, respectively.

## 2.2 Back trajectories

Back trajectories were performed with the air parcel tracking program HYSPLIT (Draxler and Hess, 1997, 1998; Draxler, 1999; Stein et al., 2015) using $1°$ resolution meteorological data from the Global Data Assimilation System (GDAS). To obtain a representative sampling of the vapor source region, the condensing air above Barrow, AK was subdivided into 1000 air parcels, each representing an equal amount of condensing water. We refer to the height of each air parcel as the 'air parcel arrival height'. Each of the 1000 air parcels was tracked backward in time for 10 days (240 hours). The vapor source location was defined as the place where the back trajectory of the air parcel sank into the planetary boundary layer (PBL). Relative to previous studies that tracked vapor change in an air parcel along the trajectory (e.g., Sodemann et al. (2008a)), we adopted a simpler procedure that assumes vapor in the air parcel is well represented by the air at the latest interaction with the PBL. This assumption is justified because mass movement in the PBL is dominated by vertical turbulence relative to horizontal advection. Figure 1 shows endpoints of all trajectories that sank into the PBL. However, only trajectories that ended over water with < 96% sea ice cover were used for calculations; parcels that sank where there was less than 96% sea ice cover were used for calculations. Parcels that never sank into the PBL or those that sank into the PBL over land or ice-covered ocean were ignored. Ocean-originating air parcels comprised about 71% of all trajectories.

Back trajectory analysis was performed for dates when precipitation was collected. The starting times for the back trajectories corresponded to times of maximum precipitation intensity, based on a combination of sampling records, surface analysis maps of Alaska available through the National Center for Environmental Prediction, and the returns of the millimeter wavelength cloud radar (MMCR) (Johnson and Jensen, 1996; Bharadwaj et al., 2011). Greater Doppler vertical velocities, reflectivities, and spectral widths from the MMCR broadly indicated more intense precipitation. Because the gridded meteorological files used for tracing the back trajectories had three-hour resolution, the chosen starting time represented average conditions over a three-hour period. If precipitation lasted for more than three hours, the most intense three hour time window was selected. If the precipitation was of approximately uniform intensity, the most temporally homogeneous three-hour time window was selected, with preference for time windows where precipitation occurred over the duration of the three hours.

The method for selecting the altitudes where the air parcels began their back trajectories is described in full in Putman (2013). Briefly, returns of the reflectivity and Doppler vertical velocity (Holdridge et al., 1994; Johnson and Jensen, 1996; Bharadwaj et al., 2011; Regional Climate Center, 2012) from the MMCR were processed with algorithms developed by Zhao and Garrett (2008) to estimate the precipitation rate profile ($g\ m^{-2}\ s^{-1}$) as a function of height. The precipitation rate profile was differentiated with respect to height, yielding the condensation rate profile ($g\ m^{-3}\ s^{-1}$) and then subdivided into the aforementioned 1000 air parcels so as to ensure that each parcel contained an equal fraction of total precipitation.

At both the air parcel initiation altitude above the precipitation site and the vapor source region, the meteorological data for our analysis came from the Global Data Assimilation System reanalysis gridded dataset. At the condensation site, we extracted from GDAS the air temperature at each height containing an air parcel. At the vapor source, we extracted the 2 m relative humidity and 2 m air temperature. Sea surface temperature data for the deuterium excess analysis came from the NOAA gridded sea surface temperature dataset (NOAA/OAR/ESRL PSD at Boulder Colorado USA, 2013).

## 2.3 Calculation of $\Delta \bar{T}_{cool}$, $\bar{T}_d$, and $mtn$

To quantify the relationship between the vapor source region and the isotopic composition of precipitation, we used three physically based metrics: the average amount of cooling during air parcel transport $\Delta \bar{T}_{cool}$, the average dew point at the vapor source region $\bar{T}_d$, which characterizes planetary boundary layer conditions, and the presence or absence of mountains along the transport path, described by the Boolean variable $mtn$. The first two metrics were calculated from the meteorological data at the vapor source and precipitation site. The third was assigned based on the air parcel trajectory.

### 2.3.1 $\Delta T_{cool}$

An estimate of air parcel cooling that produced condensation, $\Delta T_{cool}$, is a bulk metric quantifying the magnitude of Rayleigh distillation along the trajectory (Sodemann et al., 2008a). This approach simplifies the integration of cycles of warming and cooling that may occur along a trajectory to a net reduction in temperature. For each air parcel, we calculate $\Delta T_{cool}$ as the difference between the temperature at the air parcel lifted condensation level (LCL) above the source region $T_{LCL}$, and the condensation temperature, $T_c$, at the air parcel arrival height extracted from reanalysis above Barrow, AK, i.e.,

$$\Delta T_{cool} = T_{LCL} - T_c \tag{1}$$

To determine $Q_{sat,z}$, we start from the dry adiabatic lapse rate, (-9.8 °C km$^{-1}$). From this we determine the temperature $T_z$ at altitude $z$, starting with the 2 meter temperature $T_{2m}$.

The saturation vapor pressure at elevation $z$, $e_{sat,z}$ is then

$$e_{sat,z} = 0.6113 e^{[5423(\frac{1}{T_0} - \frac{1}{T_z})]} \tag{2}$$

where $T_0$ = 273.15 K (Stull, 2015). We may then write the saturation specific humidity, $Q_{sat,z}$, as

$$Q_z = 0.622 \frac{e_{sat,z} h_z}{P_z} \tag{3}$$

where $h_z$, the relative humidity at height $z$, is assumed to equal 1 (air is vapor-saturated) and the pressure at height $z$, $P_z$, is

$$P_z = 1013.25[1 - (2.25577 * 10^{-5})z]^{5.25588} \tag{4}$$

Calculating the 2 m specific humidity, $Q_{2m}$, is simply a special case of the general calculation: we use the 2 m temperature $T_{2m}$, fractional relative humidity $h_{2m}$, and pressure $P_{2m}$ from reanalysis in Equations 2 and 3, rather than using the dry adiabatic lapse rate, $h = 1$, and Equation 4, respectively.

Finally, we find the elevation where $Q_{2m}$ equals $Q_{sat,z}$. The temperature at this elevation is $T_{LCL}$.

$\Delta T_{cool}$ was calculated individually for each of the 1000 air parcels in an event. We report the mean of all air parcels that were traced to the marine PBL, $\Delta \bar{T}_{cool}$, as characteristic of the event.

### 2.3.2 $T_d$

We used the vapor source 2 m dew point $T_d$, to represent the conditions of the PBL in the vapor source region, because the relative proportions of the moist surface air and dry subsiding air determine the $T_d$ of the marine PBL. The choice of $T_d$ rather than sea surface temperature $T_{ss}$ and relative humidity $h_{2m}$ reflects our conviction that $T_d$ provides a better representation of conditions within the PBL, from which vapor with its characteristic $\delta^2 H$ will start its' trajectory to the precipitation site (see section 3.2) We approximate $T_d$ using

$$T_d = [\frac{1}{T_0} - 1.844 * 10^{-4} ln(\frac{e_{sat,2m}h_{2m}}{0.6113}]^{-1} \tag{5}$$

(Stull, 2015) with saturation vapor pressure, $e_{sat,2m}$, from Equation 2 and the 2 m air temperature, $T_{2m}$, and relative humidity $h_{2m}$ from reanalysis data.

$T_d$ was calculated for the vapor source indicated by each of the trajectories that was traced to the marine PBL; the mean of these $T_d$ values is reported as a single value, $\bar{T}_d$, characteristic of the event.

### 2.3.3 $mtn$

Vapor originating in the Gulf of Alaska typically must be transported over the Alaska and Brooks Ranges to contribute to precipitation at Barrow, whereas vapor originating anywhere in the Arctic Ocean, Bering Strait, or western North Pacific typically does not encounter major orographic obstacles during its transport to Barrow. The orographic effect on isotope ratios of precipitation was quantified with a Boolean variable, '$mtn$', defined as whether (1) or not (0) most air parcels crossed the Alaskan and/or Brooks ranges during transport to Barrow. The value of $mtn$ was assigned manually based on the general pattern of transport observed in the trajectory plots.

## 3 Results and discussion

In this section we discuss the vapor source annual cycle and statistical relationships between the isotopic composition of precipitation, vapor source region, and the variables $\Delta \bar{T}_{cool}$, $\bar{T}_d$, and $mtn$, that characterize the relationship of vapor source and transport to the isotope values measured at Barrow, AK.

### 3.1 Vapor source region annual cycle

The vapor source regions for precipitation at Barrow changed seasonally (Figure 1). Vapor fueling winter (December, January, February) precipitation originated furthest south, typically in the Gulf of Alaska, and for most winter events, trajectories crossed the Alaskan and Brooks Ranges. In spring (March, April, May) the vapor for roughly half the precipitation events came from the North Pacific and traveled over the mountain ranges, as in winter. The vapor for the remaining precipitation events generally came from the southwest of Barrow, from the Bering Strait and Chukchi Sea. Vapor source regions for summer

(June, July and August) precipitation were the most northerly, typically the Chukchi Sea or Bering Strait. Synoptic systems moving counterclockwise around the Arctic Ocean characterized summer air parcel transport. In fall (September, October, November), vapor also came from the Chukchi and Beaufort Seas, but with air parcel transport from the east to Barrow, the reverse of the spring and summer parcel transport patterns. The Gulf of Alaska provided vapor for a few fall events, with air parcel transport over the Brooks and/or Alaskan mountain ranges, as in winter.

In association with the latitudinal variation in the vapor source region, the temperature difference along the trajectory $\Delta \bar{T}_{cool}$ and vapor source dew point $\bar{T}_d$ also varied (Figure 2). The mean latitude of the vapor source region, $\bar{V}_{Lat}$, and $\Delta \bar{T}_{cool}$ varied inversely, with more cooling being associated with lower $\bar{V}_{Lat}$, i.e. greater meridional transport. For any given season $\bar{T}_d$ was warmer in the south, and cooler in the north. There are also seasonal differences; at any latitude $\bar{T}_d$ was warmer in summer and cooler in winter.

The migration of the mean latitude of the vapor source region can be tied to the seasonal cycling of solar insolation in the northern hemisphere via two mechanisms. Decreased solar insolation during winter drives expansion of the northern Polar circulation cell, which increases sea ice cover, and cold temperatures and snow cover prevent evapotranspiration. Both sea ice cover which diminishes the vapor contributions of the Arctic Ocean, and inhibited evapotranspiration allow for enhanced representation of southerly vapor sources. Increased summer insolation drives poleward contraction of the circulation cell, diminishes sea ice coverage, and warmer temperatures favors evapotranspiration such that the average vapor source area migrates north. Feng et al. (2009) documented similar vapor source migration over a much larger scale, in association with the annual north-south migration of circulation cells.

There is evidence for prior millenium-scale shifts in the southern extent of the polar circulation cell (Feng et al., 2007). Aspects of the link between seasonal variability in general circulation and seasonal vapor source cycling may be generalizable to interannual and even millennial timescales. This is relevant to modern changes in the hydrologic cycle as Marvel and Bonfils (2013) suggest that a poleward displacement of circulation cells is already occurring due to recent climate change. Additionally, changes in the isotopic composition of precipitation resulting from systematic vapor source migrations associated with changing climate may allow for interpretation of long-term isotopic records in terms of changes in atmospheric circulation, including but not limited to the precipitation site temperature.

## 3.2 The influence of vapor source on precipitation $\delta^2 H$

The local meteoric water line (with 95 % confidence intervals) is $\delta^2 H = 7.78(\pm 0.12)\delta^{18}O + 7.18(\pm 2.61)$. Figure 3 shows that the measured $\delta^2 H$ values of the 70 precipitation events fall between -280‰ and -50‰, with a pattern of summer enrichment and winter depletion that follows the well-established annual cycle for mid- and high latitudes (Feng et al., 2009; Bonne et al., 2014). Figure 3 also shows the interannual, seasonal and event-scale variability captured by the dataset where the spline captures 65% of the annual and interannual variance. The average annual cycle of the precipitation $\delta^2 H$ is strong; the spline fit explains 60% of variance in the data. The mean latitude of the vapor source exhibits a weak seasonal pattern, where the spline explains 19% of the variance. The seasonal cycles of $\delta^2 H$ and vapor source latitude are in phase, as shown in Figure 4, though the inter-event variability in both variables can be as large as the seasonal variability.

The phase relationship between $\delta^2H$ and the north-south migration of the vapor source region occurs because the vapor source region governs three critical metrics that affect the $\delta^2H$ of precipitation: 1) the temperature difference between vapor source region and precipitation site, quantified by air parcel cooling $\Delta\bar{T}_{cool}$, 2) the moisture source conditions, quantified in this work by $\bar{T}_d$, and 3) the mean air parcel transport path. A linear combination of $\Delta\bar{T}_{cool}$, $\bar{T}_d$, and $mtn$ statistically represents the event-scale variation in $\delta^2H$ with an $R^2$ value of 0.54 (p < 0.001). Table 1 contains the partial regression slopes ($\beta$), p-values, and the unique variance explained by each variable. Below we discuss the physical mechanisms that may explain the influence of each of these metrics on $\delta^2H$.

In contrast with previous assumptions that local (precipitation site) surface temperature alone is a metric for Rayleigh distillation (e.g., Dansgaard, 1964), our study relates $\delta^2H$ to $\Delta\bar{T}_{cool}$, $\bar{T}_d$ and $mtn$. Using these metrics instead of local surface temperature allows us to circumvent two restrictive assumptions. First, we do not assume that $\delta^2H$ has a spatially and temporally stationary relationship to local temperature. Bowen (2008) demonstrated that this assumption does not hold. Rather, because meridional temperature gradients are an important driver of the isotope-temperature sensitivity (Hendricks et al., 2000), when the meridional temperature gradient fluctuates, a quantity that $\Delta\bar{T}_{cool}$ captures, the sensitivity of $\delta^2H$ to local temperature also fluctuates. As demonstrated by Figure 5, the presence of mountains along the vapor transport path will deplete the isotope ratio of the precipitation relative to a uniform altitude transport, all other meteorological conditions being equivalent. The second restriction associated with using local surface temperature as a metric of Rayleigh distillation is the assumption that vapor for all precipitation events comes from a single, homogeneous source. It requires that the $\delta^2H$ of the water vapor, and thus the initial condensate, is constant in space and time. However, global measurements from the Tropospheric Emissions Spectrometer (Good et al., 2015) indicate that the vapor in the planetary boundary layer over the ocean varies with space and season, confirming previous land and ship measurements (e.g., Uemura et al., 2008; Kurita, 2011; Steen-Larsen et al., 2014). Likewise, our results indicate that vapor may come from a heterogeneous source region or variety of source regions (Figure 1) and the initial condensate, based on the evaporation conditions, should be expected to vary. The effect of a meteorologically heterogeneous source region(s) is captured by $\bar{T}_d$.

As expected, $\Delta\bar{T}_{cool}$ accounts for the largest proportion of variance in $\delta^2H$ (28.7%) among the explanatory variables. Our multiple regression yields a sensitivity of -3.51‰ °C$^{-1}$ for $\delta^2H$ with respect to $\Delta\bar{T}_{cool}$ (Table 1). Because Rayleigh distillation is considered the main source of spatial variation in $\delta^2H$, comparison with the sensitivities calculated from a simple Rayleigh model contextualize our result. In such a model, a saturated air parcel with specified temperature and vapor $\delta^2H$ is cooled iteratively in 1°C steps. At each temperature step, the condensation amount, remaining vapor, precipitation $\delta^2H$ and vapor $\delta^2H$ are calculated. No re-evaporation or non-equilibrium conditions are considered. We determined condensation in this air parcel for both adiabatic decompression and isobaric radiative cooling using equilibrium isotope fractionation factors from Majoube (1971). Because the association between precipitation $\delta^2H$ and $\Delta\bar{T}_{cool}$ during a Rayleigh process varies (Dansgaard, 1964), the sensitivity range for moist adiabatic cooling from 10 °C to -15 °C, with a lapse rate of -6.5 °C km$^{-1}$, ranges between -3.46‰ °C$^{-1}$ to -5.45‰ °C$^{-1}$, while moist isobaric radiative cooling across the same temperature range yields sensitivities from -5.47‰ °C$^{-1}$ to -7.88‰ °C$^{-1}$. The sensitivity exhibited by our data is just above the low end of the range determined for moist adiabatic cooling and was substantially lower than the range using isobaric cooling. The similarity between our data

and the moist adiabatic model results suggest that moist adiabatic cooling was likely the dominant mechanism for precipitation during air parcel transport to Barrow, although scatter in the $\delta^2 H$ data could also be due to variable contributions of radiative cooling. The relatively low observed sensitivity relative to both theoretical sensitivities may be explained by additions of vapor to air parcels during poleward meridional transport, which were not considered by our back trajectory scheme, but are supported

by the two-stream isentropic vapor source transport model (Noone, 2008).

     Our multiple linear regression attributes a substantial fraction of the variance in $\delta^2 H$ to variations in $\bar{T}_d$ (10.5%, Table 1), which is used to represent the source conditions. We prefer $T_d$ to the classical variables $T_{ss}$ and $h$ for determining isotopic evaporative fluxes. This choice is based on our understanding that the meteorological variable $T_d$ characterizes the bulk vapor content and isotopic ratio of the marine PBL, independent of the vapor temperature. When advected to the free troposphere, it

is this vapor that will form precipitation. Additionally, through equilibrium fractionation $T_d$ also determines the isotopic ratio of the first condensate at the LCL, where Rayleigh distillation begins.

     Within the marine PBL, several inter-related factors/processes are at work to determine the starting point of a Rayleigh trajectory. The first is the isotopic flux of evaporation from the sea surface. Most studies estimate this flux using the classic model by Craig and Gordon (1965). In that model, three variables control the evaporative flux: the sea surface temperature,

$T_{ss}$, $\delta^2 H$ above the laminar layer, and the humidity $h_{ss}$ above the laminar layer (e.g., at 2 m), defined relative to $T_{ss}$. Though $h_{ss}$ is not a measured quantity nor one that is normally modeled, it is determined by $T_d$ above the laminar layer and $T_{ss}$. Hence isotopic fluxes can be determined with the classical model using $T_{ss}$, and $T_d$ and $\delta^2 H$ above the laminar layer as input variables. From a physical point of view, $T_{ss}$ determines the amount of equilibrium fractionation at the water-air interface. $T_d$ and vapor $\delta^2 H$, as well as $T_{ss}$, control kinetic fractionation as vapor diffuses across the laminar layer. It should be noted

that when $T_{ss}$ is large, $T_d$ tends to be large as well, as a result of their change with latitude and season. $T_d$ and $\delta^2 H$ are also correlated, which will be discussed below. Therefore, all three variables controlling the evaporative flux, $T_{ss}$, and $h_{ss}$ and $\delta^2 H$ above the laminar layer, are associated directly or indirectly with $T_d$, making $T_d$ a good indicator of evaporation conditions.

     The second process is convergence. At a moisture source location, low level air is moist due to evaporation near the sea surface. Convergence and uplift transports low-level moist air into the free troposphere where it mixes with dry, isotopically

depleted air descending from surrounding regions resulting in strong humidity and temperature gradients near the sea surface (below 2 m). In contrast, the specific humidity and isotopic ratios in the bulk of the PBL above 2 m are relatively constant, resulting from the relative contributions of vertical transport of moist low-level and descending air (Fan, 2016). $T_d$ and $\delta^2 H$ at 2 m both reflect the outcome of this mixing process, and so it follows that they are positively correlated.

     The third process is condensation at the LCL. The temperature of the air mass, which equals or is very slightly less than the

local dew point, determines the amount of isotopic fractionation and thus the isotopic ratio of the first condensate. It is this isotopic composition that defines the beginning of the Rayleigh part of the trajectory. Only $T_{d,2m}$, not $T_{ss}$ nor $h_{2m}$, is directly associated with the condensation temperature at the LCL (which differs only slightly from $T_{d,2m}$ due to the pressure difference between 2 m and the LCL and its effect on saturation specific humidity).

     Since all three processes before Rayleigh distillation are either directly or indirectly related to $T_d$, we consider $T_d$ a better

indicator for the source conditions than either $T_{ss}$ or $h$. It is difficult, however, to theoretically assess the sensitivity of precip-

itation $\delta^2 H$ to variations in source $T_d$, because this would require quantification of the theoretical relationship of $T_d$ to $\delta^2 H$ through each of the three processes and perhaps their combinations. We here report the first empirical sensitivity of 3.23‰ °C$^{-1}$ (Table 1) for $\delta^2 H$ relative to $T_d$. At the sea surface, for $T_{ss}$ between 0 and 25 °C, equilibrium fractionation as a function of temperature yields sensitivities between 1.1-1.6‰°C$^{-1}$ (Majoube, 1971). However, a large part of this fractionation may be

offset by condensation at the LCL. Consequently, the observed sensitivity probably reflects primarily the fraction of vapor contributed by dry, isotopically depleted descending air that converges within the PBL. Mixing with the dry air causes a decrease in $T_d$, which affects the $\delta^2 H$ of the PBL in two ways: 1) making the PBL air dry and isotopically depleted, and 2) isotopically depleting the evaporative flux by enhancing kinetic fractionation (an effect of low relative humidity). Both mechanisms produce a positive association between $\delta^2 H$ and $T_d$, consistent with the sign of our observed partial coefficient (Table 1).

Upon leaving the vapor source region, the isotopic composition of vapor depends on the trajectory taken. To reach Barrow, AK, air parcels originating in the Gulf of Alaska must cross the Alaska and/or Brooks Ranges, whereas air parcels from the Bering Strait or Chukchi Sea do not have to cross high topography. Our work shows that transport across mountain ranges resulted in significant $\delta^2 H$ depletion in Barrow precipitation. Transport of vapor over mountain ranges occurred more frequently during cold months, when the Gulf of Alaska and North Pacific were the dominant vapor source regions. Since the vapor source

location in winter is governed by the expansion of the Polar circulation cell, the projected northward displacement of subtropical highs and the Polar front (Marvel and Bonfils, 2013) in a warming climate may be associated with less vapor transported over the Alaskan and/or Brooks ranges during fall, winter and spring. Fewer events traveling over the Alaskan and/or Brooks ranges would correspond to a pronounced enrichment in measured $\delta^2 H$ at Barrow during cold months.

To study the importance of $\bar{T}_d$ and $mtn$ as explanatory variables with respect to cooling during transport ($\Delta\bar{T}_{cool}$), we

divided our data into subgroups, those with $\Delta\bar{T}_{cool}$ <7 °C (corresponding to short trajectories), and those with $\Delta\bar{T}_{cool}$ >7 °C (corresponding to long trajectories) and recalculated the statistics. Table 2 summarizes the results and Figure 6 shows the standard deviation of $\bar{T}_d$ by category. The breakpoint of 7 °C was chosen by testing different breakpoints and finding one that maximized the statistical power of the short trajectory regression while preserving the strong relationship between $\delta^2 H$ and $\bar{T}_d$. For the small $\Delta\bar{T}_{cool}$ subgroup, $\bar{T}_d$ explains almost half the variance in $\delta^2 H$ ($R^2 = 0.43$), whereas for the large $\Delta\bar{T}_{cool}$

subgroup, $\bar{T}_d$ explains very little variance ($R^2 = 0.007$). This difference implies enhanced isotopic modification over long trajectories. In contrast, the $\delta^2 H$ values of the small $\Delta\bar{T}_{cool}$ subgroup are not well explained by the Boolean variable $mtn$ ($R^2 = 0.03$), whereas $mtn$ explains about one-fifth of the variability of the large $\Delta\bar{T}_{cool}$ subgroup ($R^2 = 0.18$). For the small $\Delta\bar{T}_{cool}$ subgroup, $\Delta\bar{T}_{cool}$ $R^2 = 0.02$. For the large $\Delta\bar{T}_{cool}$ subgroup, $\Delta\bar{T}_{cool}$ explained a quarter ($R^2 = 0.22$) of the variance in $\delta^2 H$.

Because the events with smallest $\Delta\bar{T}_{cool}$ tended to occur in summer, the strong relationship between $\bar{T}_d$ and $\delta^2 H$ indicates

that precipitation $\delta^2 H$ in summer predominantly reflects variability in source conditions. The strong relationship between $mtn$ and the variation in $\delta^2 H$ for large $\Delta\bar{T}_{cool}$ indicates that precipitation $\delta^2 H$ in winter predominantly reflects whether most air parcels crossed the Alaska and/or Brooks mountain ranges. Notably, $\Delta\bar{T}_{cool}$ could significantly predict $\delta^2 H$ for long trajectory events, and it explained less variance than expected, given the emphasis on Rayleigh distillation in explaining spatial variation in precipiation stable isotopes.

Among the simple regressions, almost half the variance in $\delta^2H$ for events with $\Delta\bar{T}_{cool}$ <7 °C was explained by $\bar{T}_d$. This is a notable result, as the isotope composition of the initial vapor is not emphasized to the same degree as Rayleigh distillation in isotope hydrology. There are two reasons why $\bar{T}_d$ may explain so much variance for short trajectory events. First, storm events with minimal cooling during air parcel transport typically originated close to Barrow in the Arctic Ocean. A smaller vapor source area predicts less variation in $T_d$ among air parcels: a more homogeneous source. We quantify this effect by examining the distribution of intra-event $\bar{T}_d$ standard deviations ($\sigma\bar{T}_d$) for the short and long trajectory event subsets (Figure 6). Short trajectory ($\Delta\bar{T}_{cool}$ <7 °C) events had a median $\sigma\bar{T}_d$ of 2.79 °C, which was less than the long trajectory ($\Delta\bar{T}_{cool}$ >7 °C) median $\sigma\bar{T}_d$ of 4.68 °C. Less variability among air parcels in the short trajectory subset allowed the among-event relationship of $\delta^2H$ to $\bar{T}_d$ to emerge. In addition, some of the variability in measured precipitation $\delta^2H$ may be caused by processes occurring during transport, such as radiative cooling, air mass mixing, and different degrees of mountain-induced rainout. The opportunity for these effects to impact the precipitation isotope value increases with increasing transport distance, obscuring the relationship of the precipitation $\delta^2H$ to the $\delta^2H$ of the initial vapor at the source and therefore to $\bar{T}_d$.

The three chosen variables explain just over half (54%) the variance of $\delta^2H$. This is not surprising, considering that many other mechanisms can also influence the $\delta^2H$ of the vapor and precipitation. These mechanisms include (but are not limited to) condensation temperature, supersaturation in the mixed phase cloud, sub-cloud dryness, phase of precipitation, precipitation intensity, evapotranspiration of land sources, and the amount of sea ice at the vapor source. The effects of several of these factors, including condensation temperature, sub-cloud dryness, sea ice concentration at the vapor source, and phase of precipitation (rain vs. snow), were tested as additional explanatory variables in the multiple regression, but yielded statistically insignificant results with little to no additional variance explained. Clearly, compared with the three chosen variables, the effects of these variables are relatively minor, such that the statistical power is not sufficient to reveal their significance.

### 3.3  The influence of vapor source on deuterium excess

Deuterium excess (d-excess, or $d$) of precipitation is often used to investigate source region conditions such as $T_{ss}$ and $h$ that affect evaporation (Dansgaard, 1964). Empirical studies have linked marine boundary layer vapor deuterium excess ($d = \delta^2H - \delta^{18}O$) to $T_{ss}$ and $h$ or $h_{ss}$ (Uemura et al., 2008; Kurita, 2011; Steen-Larsen et al., 2014). These results agree qualitatively or semi-quantitatively with theoretical predictions (Merlivat and Jouzel, 1979). However, in order for source vapor $d$ values to be preserved in precipitation, $d$ must be conserved through condensation and post-condensation processes. This assumption may not be realistic for several reasons. First, even simple equilibrium Rayleigh distillation does not yield constant $d$ values in precipitation (Dansgaard, 1964). Second, non-equilibrium processes associated with snow formation may substantially alter $d$ (Jouzel and Merlivat, 1984). Third, evaporation or sublimation under the cloud base and/or at the snow surface tends to decrease $d$ (Stichler et al., 2001).

While studies indicate that $d$ in vapor contains vapor source information (Steen-Larsen et al., 2014; Bonne et al., 2015; Steen-Larsen et al., 2015), direct comparison of precipitation $d$ to vapor source conditions via Lagrangian back trajectory vapor source estimation has produced complicated results. For example, Sodemann et al. (2008b) found that while the $d$ of precipitation contains identifiable source information, it 'does not directly translate into the source region $\bar{T}_{ss}$'. In a study of

vapor sources for precipitation in Antarctica, Wang et al. (2013) noted that the classical interpretation of measured $d$ would predict that the highest average $d$ found at Dome Argus would correspond to the warmest (most northerly) vapor sources. However, precipitation at Dome Argus was linked to southerly (cooler) vapor sources. The authors suggested the high $d$ value was due to the vapor pressure deficit of dry air blowing off sea ice. Likewise Good et al. (2014) attributed the significant

correlation between high $d$ and source relative humidity ($h$) for precipitation collected at four northeast U.S. locations during Superstorm Sandy to oceanic evaporation into a dry continental air mass that was entrained into the superstorm.

Our study reveals a relatively more conclusive relationship between vapor source and event-scale precipitation $d$, as summarized by four simple regressions against $h_{2m}$, $h_{sst}$, $T_{ss}$, and $T_d$ shown in Table 3. Though $d$ is not significantly predicted by $\bar{h}_{2m}$ (p = 0.86) it is significantly predicted by $\bar{h}_{ss}$ (p < 0.001, $R^2$ = 0.34), with a slope of -0.4‰% $^{-1}$. This value is consistent

with the -0.4 to -0.6‰ % $^{-1}$ range reported in the literature for vapor (Uemura et al., 2008; Pfahl and Wernli; Bonne et al., 2014). $\bar{T}_{ss}$ is also a significant predictor (p = 0.0023) though the variance explained is 12 % and the sign of the coefficient is negative, opposite to expectations. If $d$ is regressed against both $\bar{T}_{ss}$ and $\bar{h}_{ss}$, the multiple regression is significant (p < 0.001, not shown in Table 3) and explains 36 % of variance, most of which is due to the strong relationship with $h_{ss}$. The vapor source region dew point, $\bar{T}_d$, significantly predicts $d$ (p <0.001) and explains a non-trivial portion of the variance ($R^2$ = 0.24)

with a negative slope (-0.53‰°C $^{-1}$). This is an interesting result with respect to the utility of $T_d$, a measurable quantity, and is consistent with our earlier argument that $T_d$ is strongly related to $h_{ss}$. Both variables provide a better representation of source conditions than $T_{ss}$ and/or $h_{2m}$. A low value of $h_{ss}$ or $T_d$ corresponds to a strong influence of descending dry air within the PBL, which enhances kinetic isotopic fractionation and produces a high value of $d$. This mechanism explains the negative correlation between $d$ and $T_d$, and is expected for the relationship between $d$ and $h_{ss}$. Alternatively, the vapor in descending

air may have a high value of $d$ (Fan, 2016), or both mechanisms may contribute to this result.

Our dataset also shows systematic seasonal variations in $d$. Figure 7 shows that $d$ cycles annually, with the maximum occurring in October or November and lagging the annual maximum of $\delta^2 H$ by 2-3 months (or $\sim 90°$). This phase relationship explains the lack of linear association between $d$ and $\bar{T}_{ss}$ and $\bar{h}_{2m}$ because the two latter variables are both in phase with $\delta^2 H$. Systematic seasonal variations in precipitation $d$ occur in the Northern Hemisphere (Feng et al., 2009), particularly in the

Arctic (White et al., 1988; Johnsen et al., 1989; Kurita, 2011; Kopec et al., 2016). These studies suggest that the conditions producing $d$ variation have systematic annual variations in their magnitude and relative importance.

## 4  Conclusions

The vapor source regions identified by HYSPLIT for storms at Barrow, AK, USA exhibited interannual, annual, and substantial inter-event variability. On average, vapor came from the North Pacific and Gulf of Alaska, the most southerly vapor source

areas, in cold months when the Polar circulation cell extended southward. Vapor came from the Bering, Chukchi and Beaufort seas, the most northerly sources, in warm months when the Polar cell contracted northward. The cycle of winter depletion and summer enrichment exhibited by the $\delta^2 H$ of the precipitation followed the annual changes in the latitude of the vapor source region, as a result of source region controls on evaporation, transport, and condensation conditions. However, substantial intra-

season variability occurred in both source and $\delta^2 H$, indicating scatter in the seasonal relationship. A linear combination of the average vapor source region dew point ($\bar{T}_d$, $\beta = 3.23\,‰\,°C^{-1}$), average cooling of the air parcels during transport ($\Delta\bar{T}_{cool}$, $\beta = -3.51\,‰\,°C^{-1}$) and passage of air parcels over mountains or not ($mtn$, $\beta = -32.11\,‰$ when $mtn = 1$) explained 54 % of the event-scale variance in $\delta^2 H$. For the subset of events where $\Delta\bar{T}_{cool}$ was $< 7\,°C$ (short trajectories), $\bar{T}_d$ alone explained 43 % of the variance in $\delta^2 H$. For the subset of events where $\Delta\bar{T}_{cool}$ was $> 7\,°C$ (long trajectories), $\bar{T}_d$ did not significantly predict $\delta^2 H$, but $mtn$ alone explained 18 % of the variance in $\delta^2 H$. Neither the average vapor source relative humidity, $\bar{h}_{2m}$, nor the average vapor source sea surface temperature, $\bar{T}_{ss}$, nor both combined, significantly explained the variations in deuterium excess. The vapor source region relative humidity with respect to sea surface temperature, $\bar{h}_{sst}$, explained 34 % of the variance in $d$, with the expected negative sensitivity, and the source dew point, $\bar{T}_d$, explained a nontrivial proportion of 22%. Our results suggest that $T_d$ is related to $h_{ss}$, and that both variables are more indicative of PBL conditions that directly affect vapor supplied to the free troposphere than $T_{ss}$ or $h_{2m}$. Deuterium excess also exhibited a systematic seasonal variation with maximum $d$ in October and minimum $d$ in March; though additional study is needed to identify the mechanism responsible for the annual cycle.

Our study highlights how variations in stable isotopes of precipitation measured on an event-by-event basis can be interpreted in the context of the vapor source. The mechanisms identified, most notably the north-south migration of the vapor source region in phase with expansion and contraction of the Polar circulation cell, may also operate on times scales longer than that of our study, and may be a source of variation in isotopes measured in ice cores, pedogenic carbonates, and speleothems.

## 5   Data availability

The processed data used for this research are available as a supplement to the manuscript. Raw and partially processed results of the back trajectory runs may be obtained from Annie Putman (putmanannie@gmail.com).

*Acknowledgements.*  This project was supported by the National Science Foundation Grant 1022032, the Intensive Operational Period (IOP) Program of the Atmosphere Radiation Measurement, and Dartmouth College. The authors thank Walter Brower and Jimmy Ivanhoff for their sample collection efforts at the ARM NSA station, and Ben Kopec, J.L. Bonne and an anonymous reviewer for their valuable comments.

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

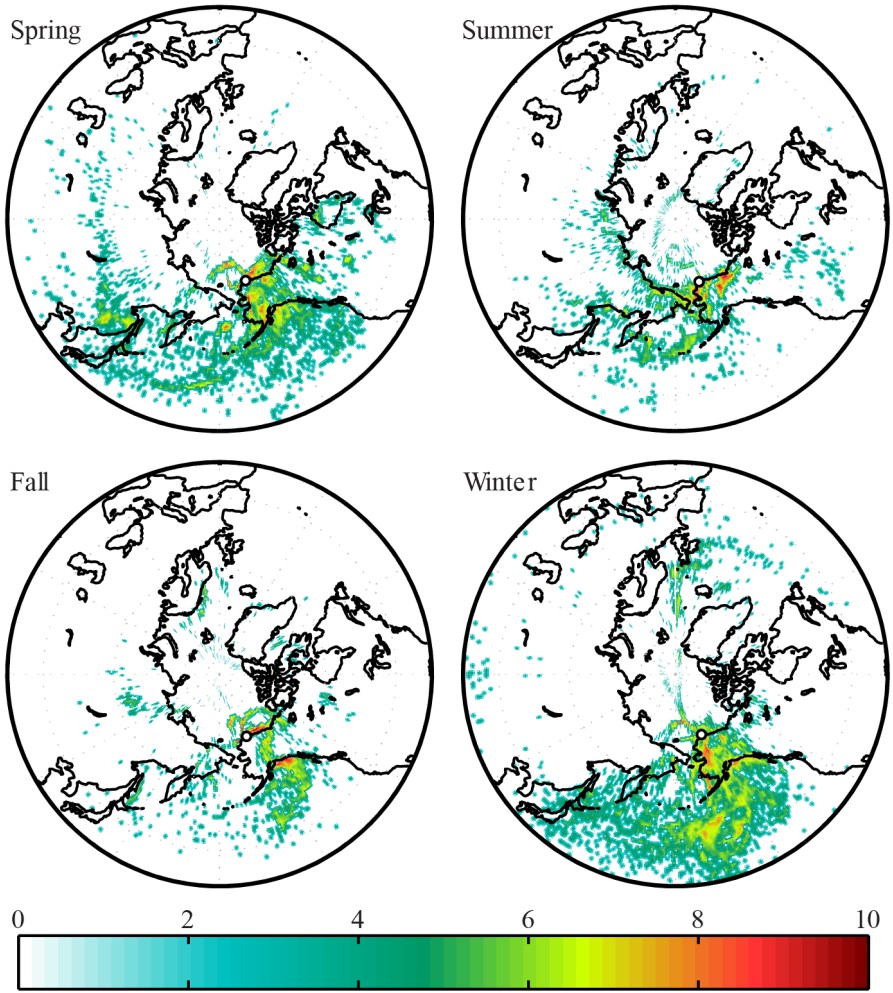

**Figure 1.** Spatial distribution of the vapor source region by season. Color indicates the relative frequency that a pixel was identified by HYSPLIT as a vapor source. Red indicates the most frequent vapor source for a given season, whereas dark blue indicates few air parcels were traced to that location. Because different numbers of events occurred in each season, each season's color scale is normalized to the total number of air parcels tracked during that season. The figure indicates that some air parcels originate over land, but these were not included in calculations.

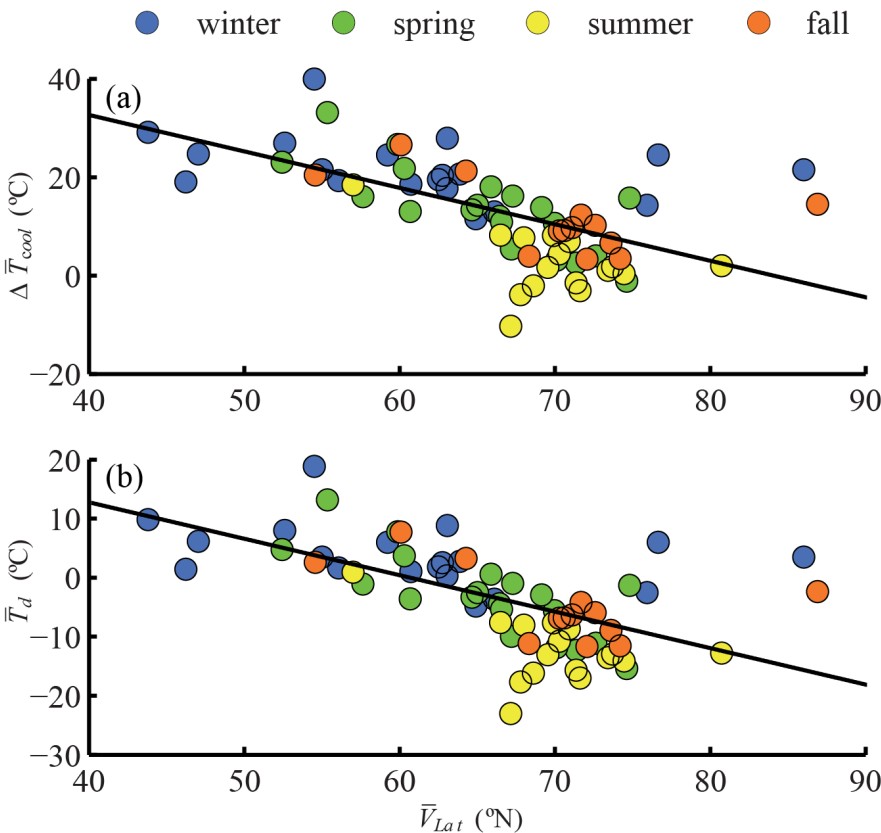

**Figure 2.** (a) Covarying behavior of mean vapor source region latitude, $\bar{V}_{Lat}$, and mean air parcel cooling during transport, $\Delta\bar{T}_{cool}$. (b) Covariation of the mean vapor source region latitude, $\bar{V}_{Lat}$, and dew point, $\bar{T}_d$. Both $\Delta\bar{T}_{cool}$ and $\bar{T}_d$ influence the $\delta^2 H$ of precipitation at Barrow, AK. Lines are best-fits; scatter from them is due, in part, to seasonal variation in latitudinal temperature gradients and vapor source conditions.

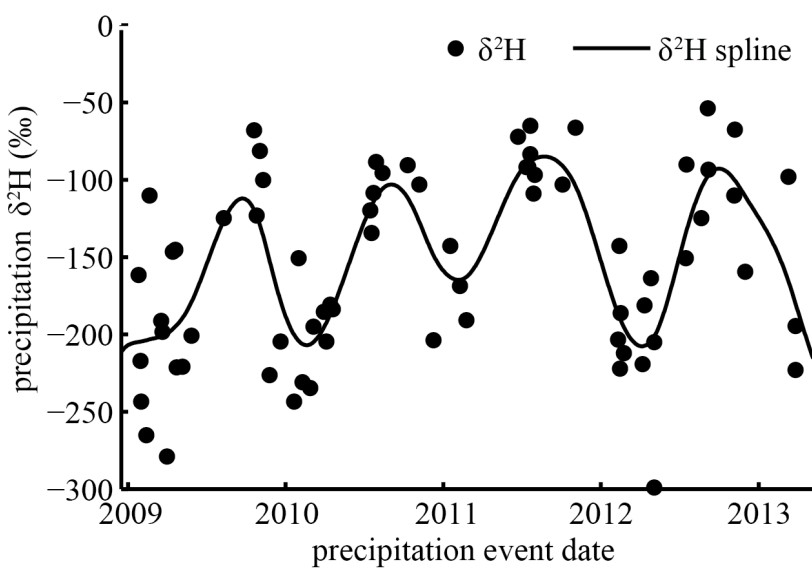

**Figure 3.** Measured $\delta^2 H$ in precipitation at Barrow, AK, exhibits variability on interannual, annual, and event time scales. The spline fit, which highlights seasonal variations, explains 65% of variance in the data with a root mean squared error of 39.7‰. Of the three timescales, annual variability shows the greatest amplitude, though variability among events is also substantial. Maximum enrichment corresponds roughly to the warmest months (June, July, August), and maximum depletion corresponds roughly to the coldest months (December, January, February).

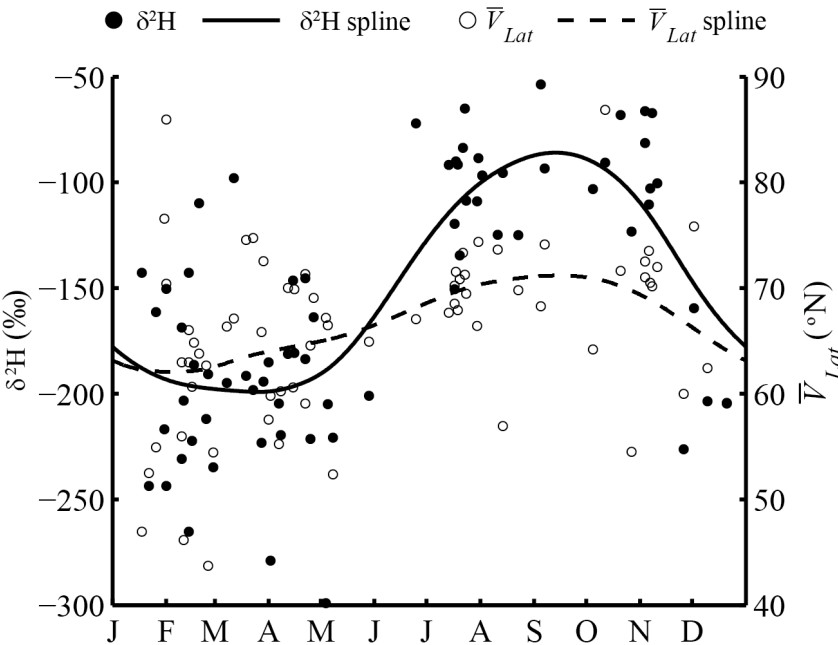

**Figure 4.** Measured $\delta^2 H$ of Barrow precipitation and mean latitude of the vapor source both exhibit an annual cycle and are in phase. The circles depict raw data, while curves are spline fits to the data. The spline fits have $R^2$ values of 0.60 and 0.19 for the $\delta^2 H$ and $\bar{V}_{Lat}$ respectively. For both datasets, the variability exhibited among events is of the same the order of magnitude as the seasonal variability.

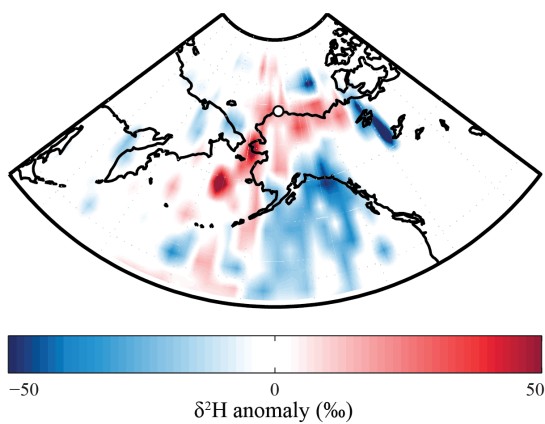

**Figure 5.** To demonstrate the effect of air parcel transport path, the residual $\delta^2 H$ of Barrow precipitation is plotted at the vapor source. The residual $\delta^2 H$ is determined by subtracting the spline shown in Figure 4 from the $\delta^2 H$ of each precipitation event. The vapor source locations, which have $1\,°$ by $1\,°$ resolution, are smoothed for clarity. Vapor from the Bering Strait or Chukchi Sea tends to produce precipitation that is enriched relative to the average. Likewise, vapor from the Gulf of Alaska tends to produce precipitation that is depleted relative to the average. This variation in vapor source reflects a difference in transport path. Vapor originating from the Gulf of Alaska must rise to cross over the Alaska Range, inducing orographic precipitation and isotopic depletion relative to air masses that do not encounter orographic obstacles.

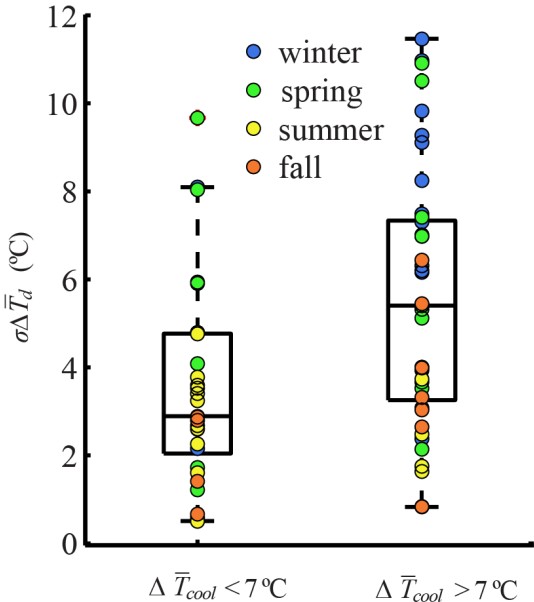

**Figure 6.** Distribution of standard deviations ($\sigma$) of $\bar{T}_d$ for events with $\Delta\bar{T}_{cool}$ <7 °C(short trajectories) and $\Delta\bar{T}_{cool}$ >7 °C (long trajectories). Colors indicate seasons. In general, small $\Delta\bar{T}_{cool}$ was associated with small $\sigma\bar{T}_d$. The variation in standard deviation is related to season, where warmer months tend to have smaller $\sigma\bar{T}_d$ and cooler months tend to have larger $\sigma\bar{T}_d$.

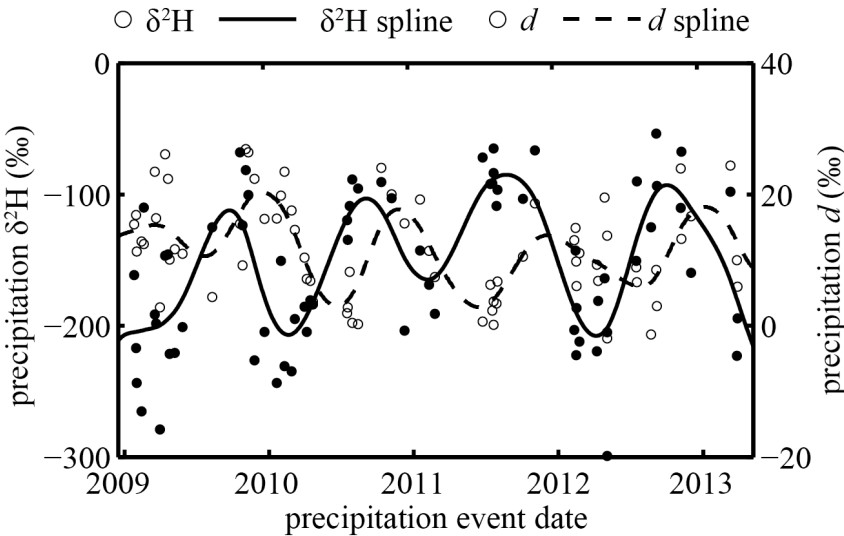

**Figure 7.** Annual maxima and minima in deuterium excess, $d$, lag those of $\delta^2 H$ by 2 - 3 months, such that the maxima are in fall and minima in spring.

**Table 1.** Response variable: $\delta^2 H$. Variation in $\delta^2 H$ is explained by a multiple linear regression ($R^2 = 0.54$) of air parcel cooling during transport ($\Delta \bar{T}_{cool}$), moisture source conditions ($\bar{T}_d$) and orographic obstacles in vapor transport path ($mtn$). Values of $\beta$ are the partial coefficients of the regression and $S.E.$ is the standard error. The variance estimate for each explanatory variable is calculated as the square of the semi-partial correlation for that variable with $\delta^2 H$. The variances reported do not sum to the total variance explained because the explanatory variables are not perfectly orthogonal.

| independent variable (slope units) | $\beta$ ($\pm$ S.E.) | p-value | variance estimate |
|---|---|---|---|
| intercept | -95.33 (8.62) | < 0.001 | |
| $\Delta \bar{T}_{cool}$ (‰°C$^{-1}$) | -3.51 (0.55) | < 0.001 | 0.287 |
| $\bar{T}_d$ (‰°C$^{-1}$) | 3.23 (0.83) | < 0.001 | 0.105 |
| $mtn$ ( ‰ when $mtn = 1$) | -32.11 (11.04) | 0.0049 | 0.059 |

**Table 2.** Three simple linear regressions against $\delta^2 H$ where $\beta$ is the regression coefficient and $S.E.$ is the standard error. Source conditions parameterized by $\bar{T}_d$ explain most variation in $\delta^2 H$ for small $\Delta \bar{T}_{cool}$, while topographic highs below the trajectory ($mtn$) explain substantial variation for large $\Delta \bar{T}_{cool}$. $\Delta \bar{T}_{cool}$ explains variability significantly only for the long transport subgroup ($\Delta \bar{T}_{cool} > 7$°C).

| Independent variable (slope units) | $\Delta \bar{T}_{cool} < 7\,°C$ $\beta$ ($\pm$ S.E.) | p-value | $R^2$ | $\Delta \bar{T}_{cool} > 7\,°C$ $\beta$ ($\pm$ S.E.) | p-value | $R^2$ |
|---|---|---|---|---|---|---|
| Intercept (‰) | -111.4 (8.6) | < 0.001 | | -115.1 (18.9) | < 0.001 | |
| $\Delta \bar{T}_{cool}$ (‰°C$^{-1}$) | -1.68 (2.1) | 0.428 | 0.03 | -3.44 (0.97) | < 0.001 | 0.22 |
| Intercept (‰) | -104.9 (6.75) | < 0.001 | | -176.5 (8.4) | < 0.001 | |
| $\bar{T}_d$ (‰°C$^{-1}$) | 2.89 (0.74) | < 0.001 | 0.43 | 1.04 (1.8) | 0.58 | 0.007 |
| Intercept (‰) | -115.2 (9.1) | < 0.001 | | -147.6 (11.9) | < 0.001 | |
| $mtn$ ( ‰ when $mtn = 1$) | -16.6 (24.6) | 0.51 | 0.02 | -49.8 (15.5) | 0.0025 | 0.18 |

**Table 3.** Explaining deuterium excess ($d$) using simple regressions against various metrics that characterize source conditions. $\beta$ is the regression coefficient and $S.E.$ is the standard error. We show results from simple linear regressions with four different independent variables: evaporation site relative humidity ($\bar{h}_{2m}$), evaporation site relative humidity relative to sea surface temperature ($\bar{h}_{sst}$), sea surface temperature ($\bar{T}_{ss}$), and 2 m dew point ($\bar{T}_d$).

| Independent variable (slope units) | $\beta$ ($\pm$ S.E.) | p-value | $R^2$ |
|---|---|---|---|
| $\bar{h}_{2m}$ (‰%$^{-1}$) | 0.027 (0.157) | 0.86 | 0.0 |
| $\bar{h}_{sst}$ (‰%$^{-1}$) | -0.395 (0.067) | $< 0.001$ | 0.34 |
| $\bar{T}_{ss}$ (‰°C$^{-1}$) | -1.17 (0.37) | 0.0023 | 0.12 |
| $\bar{T}_d$(‰°C$^{-1}$) | -0.56 (0.13) | $< 0.001$ | 0.22 |