# Peer review of "Annual variation in event-scale precipitation $\delta^2 H$ at Barrow, AK reflects vapor source region"

_Atmospheric Chemistry and Physics, 2016_

## Referee Comment (RC1) · J.-L. Bonne (Referee) · 26 Sep 2016

**General comments**

This paper presents a new dataset of isotopic composition of precipitation sampled in the Barrow, AK, Arctic station, together with an innovative method to analyze and interpret its seasonal and event time scales variations. The authors propose interesting tools to use the Lagrangian atmospheric backtrajectory model for a quantitative and statistical evaluation of the observed isotopic variations. They conclude that the seasonal variations of water isotopic values are partly due to migration of the moisture origins. They focus on the influence of three parameters which are shown to explain a large part of the observed variations of the isotopic composition: the cooling along atmospheric transport, the dew point at the moisture source and the presence of moun-

tains along the transport.

The authors made a good effort to provide a rich interpretation of their observations. The manuscript is well organized and provides necessary tables and figures. There are, however, several issues regarding discussion of the results. I recommend accepting the article after the authors address the points listed below.

**Specific comments**

**Modeling:**

The moisture source modeling used in this paper relies on strong assumptions. However, potential errors caused by these assumptions are poorly pointed out. A companion paper describing the method is currently under review and might contain these information. As this paper is not yet readable, one would need a summary of these information and eventually more details in the method description or in supplementary material, in particular concerning the points addressed below.

Contrary to Sodemann et al. 2008b method, the moisture source modeling used here does not take into account variations of the specific humidity in the air parcels along the trajectory. Processes such as the lost moisture through precipitation or reevaporation of already condensed droplets along transport are not taken into account, but could have a strong impact on the isotopic composition. Can you give more details on the potential errors inherent to this moisture sources modeling?

Also concerning the moisture sources modeling, moisture uptakes are assimilated to air masses sinking into the planetary boundary layer (PBL) above the ocean surface. Nothing is written about the potential presence of sea ice above the ocean in the region where the PBL is reached, which could however have a strong influence on the evaporation. Do you also take into account the sea ice cover in the region were the air parcels sink into the planetary boundary layer? For example: the moisture sources for the winter events are originating from a very wide range of latitudes. If most sources

are originating from the south, some sources are coming from high latitudes, up to 85°N (see winter sources latitudes on Figure 2). Can we really expect strong evaporation in those regions, over a potentially closed ocean? Have you checked the presence of sea ice in the moisture sources regions for this type of events?

**Interpretation of results:**

Concerning the interpretation of results at the seasonal scale, the seasonal variations described in the article are mostly the result of the relative preponderance of different types of synoptic scale events across the seasons. The intra-seasonal variability of the different events is often on the same order of magnitude than the variations of seasonal averages, which is too rarely pointed by the author. The clarity of the explanations might benefit from a more stronger distinction of the synoptic scale and seasonal scale variations.

**Technical corrections**

**Abstract:**

The abstract is quite long and could be more concise.

P.1, L.1 to 5: The first three sentences of the abstract could rather be at the beginning of the introduction, as they don't describe the work presented in this article but general situation of research in the domain.

P.1, L.8: "occurred" > "occuring"

**Methods:**

P.3, L.13-14: There might be an effect of sublimation of snow which could influence the isotopic composition of water, particularly for sunny periods, even within 24 hours. Did you make some experiments to test the evolution of fresh snow on your sampling site?

P.3, L.14: At which temperature were the samples stored, and how long?

P.4, L.1-4: Considering that a moisture source is corresponding to an air parcel sinking into the PBL is a strong assumption. More justifications of this method would be expected. If this method is described in Putman et al. (2015), add a reference here.

P.4, L11-12: By "the most temporally homogeneous thee-hour time window", do you mean homogeneity in the precipitation amount or in the meteorological records? Do you have particular criteria to define the preference for the middle of the event? Were the event times defined automatically or manually?

P.5., L.5: "The same was done for an array...": explicit that this is to calculate $Q_{sat,z}$ and define $Q_{sat,z}$.

P.5, L.5-6: Explicit $h_z$, $T_z$, $P_z$: fractional relative humidity, temperature and pressure at elevation z.

P.5, L.25: Are mtn values assigned manually or automatically? If automatic, then explicit the criteria.

**Results and discussion**

P. 6, L.6-15: This is a very qualitative description of Figure 1. The mean latitude of moisture sources could be introduced before and used to give quantitative aspects to this description. This description focuses on the seasonal averages of the moisture sources, but Figure 2 shows a very strong variability at the event time scale, which can be of a larger order of magnitude than the variations of the seasonal average for the mean latitude of the moisture source. For example, some events in winter have moisture sources located as north as in summer, or even further north. The normalisation of the maps from Figure 1 can also give an impression of wider or more local moisture sources depending on the total number of events and the difference between each event. Is this description of moisture sources regions still valid for absolute values without normalization to the number of events, or for individual events instead of the average of all events?

[Figure]

P. 6, L. 30-32: Not clear if the last sentence refers to Feng et al. (2007).

P. 7, L. 6-7: This sentence is really affirmative, whereas Figure 4 shows a very strong dispersion, particularly for the averaged $VLAT$. This affirmation should be tempered and a statistical evaluation of the spline fits and there correlations should be given, as well as the standard deviations of the data series. The seasonal scale might not be the better scale to look at.

P. 8, L.3: How did you choose the temperatures from 10C to -15C in you theoretical cooling experiments? What would be the effect on the slopes of a variation of these temperatures on the order of magnitude of the observed variations?

P. 8, L. 18: Rather write "more than 20C" instead of "> 20C".

P. 9, L. 1: "amount" instead of "amounts"?

P.9, L.5: How was the 7C criteria chosen? Is it close to the median of the distribution of $\Delta T_{cool}$?

P. 9, L.12: Insert a reference to figure 6 to show the repartition of small and large $\Delta T_{cool}$ across seasons.

P. 10, L.6: This is not directly about precipitation d-excess but can be of interest: some studies of water vapour d-excess in Arctic regions have depicted a partial conservation of the source d-excess signal under certain atmospheric transport conditions, with relations between observed d-excess and moisture source relative humidity.
Bonne, J.-L., Masson-Delmotte, V., Cattani, O., Delmotte, M., Risi, C., Sodemann, H., and Steen-Larsen, H. C.: The isotopic composition of water vapour and precipitation in Ivittuut, southern Greenland, Atmos. Chem. Phys., 14, 4419-4439, doi:10.5194/acp-14-4419-2014, 2014.
Bonne, J.-L., et al. (2015), The summer 2012 Greenland heat wave: In situ and remote sensing observations of water vapor isotopic composition during an atmospheric river event, J. Geophys. Res. Atmos., 120, 2970–2989, doi:10.1002/2014JD022602.

Steen-Larsen, H. C., A. E. Sveinbjörnsdottir, Th. Jonsson, F. Ritter, J.-L. Bonne, V. Masson-Delmotte, H. Sodemann, T. Blunier, D. Dahl-Jensen, and B. M. Vinther (2015), Moisture sources and synoptic to seasonal variability of North Atlantic water vapor isotopic composition, J. Geophys. Res. Atmos., 120, 5757–5774, doi:10.1002/2015JD023234.

**Conclusions**

P. 10, L. 29-31: This conclusion on the origins of moisture is valid for the average of the seasonal moisture sources, but should be tempered by pointing out the event to event variation of the moisture sources.

**References**

P.13, L. 32-36: Logically, the two papers numbering should be inverted (2008a and 2008b).

**Tables and figures**

Table 1 and 2: The legends do not clearly describe the contents of the tables. Why are different intercepts given for each variable in Table 2 and only one value in Table 1, if the only difference between the two tables are the division of all samples in two groups?

Figure 7: Parenthesis not closed in right y-axis label.

Figure 3 and 7: It would be more readable with x-axis ticks corresponding to the beginning of the years instead of the beginning of each December.

---

## Referee Comment (RC2) · Anonymous Referee #2 · 18 Oct 2016

Review of

*"Annual variation in precipitation $\delta^2 H$ reflects vapour source region at Barrow, AK"*

by A. L. Putman et al.

Paper published in ACPD on 11 August 2016

**1 General Comments**

This paper presents an interesting dataset of the event-scale $\delta^2$H and deuterium excess signature of precipitation from northern Alaska. The authors use a very simple back-trajectory-based analysis of the transport and moisture source conditions which they summarise in 3 main characteristics to interpret their data. These are 1) the moisture source dew point temperature at 2m, 2) the total cooling between the lifted condensation level at the moisture source and the precipitation level in the cloud at the measurement site (arrival temperature) and 3) whether the air parcels that are transported to the measurement site across the Brooks and/or the Alaskan ranges.

I recommend publication of this overall well-written manuscript, but I have four major concerns that should be addressed beforehand as well as a many specific comments listed below:

1 **Moisture source identification and particularly the implicit assumptions made:**
see specific comments 3-7.

2 **Choice of the parameters that explain the variance of the isotope signature of precipitation in Barrow:**
For me the choice of the parameters that were used to explain the precipitation isotope signal in Barrow seems random. It makes sense to look at moisture source and transport conditions but in my opinion there is no reason for completely neglecting the local conditions. Particularly at Barrow, the precipitation phase (liquid or snow) probably plays an important role for the end isotope composition of the precipitation event as it determines whether there is isotopic exchange (for rain drops, see specific comment 2) or not (for snowfall) with the local vapour. Also precipitation intensity plays an important role. The authors have some detailed information about the precipitation structure from their radar data and could use this to try to further understand the local processes. If this is done in an other paper, then this should be clearly stated. Also I do not fully support the choice of the variable $T_d$ as representative for the moisture source conditions (see specific comment 12).

3 **Expansion of the northern polar circulation cell and its link to moisture source location**
The link between the event-based moisture source location of precipitation and the polar circulation cell is described in a very qualitative way. A link between the weather systems driving the moisture transport at the event timescale leading to precipitation at Barrow and the more climatological description of the polar circulation is not obvious and not trivial to make. The formulations used throughout the paper should be more careful and kept as hypotheses.

4 **Critical discussion of results in view of the existing literature**:
in particular see specific comments 24 and 27.

**2 Specific comments**

1. p. 1, title: It would be nice to include in the title the fact that it is event-scale precipitation samples that the authors analyse in this paper. Something like: "Annual variation in event-scale precipitation $\delta^2$H reflects vapour source region at Barrow, AK". Also Barrow, AK could be replaced by northern Alaska.

2. p. 18-23: The local conditions during cloud formation and during precipitation also play an important role for the isotope composition of precipitation. For rainfall for example below cloud effects (evaporation and exchange with ambient vapour) can have a strong impact on the isotope composition of precipitation (20-40‰ for $\delta^2$H, see Pfahl et al. (2012), Aemisegger et al. (2015)).

3. p. 3, L. 29: The reanalysis dataset (wind fields) that is used for the trajectory calculation should be mentioned here as well as its horizontal resolution.

4. p. 4, L. 2: What do the authors mean with "The first time"? Is the time reference forward or backward? Does that mean the first time when following the trajectory back from the arrival point? And does that mean that one trajectory can have only 1 associated moisture source? This would be a very strong assumption about the moisture source location. Uptakes of moisture can happen all along an air parcel's trajectory (see Sodemann et al. (2008)) and they can sometimes be linked to surface evaporation even though they are not in the boundary layer (PBL), particularly over land. If for each trajectory only the latest passage in the PBL before arrival at the measurement site is considered then this means that the authors assume very strong mixing. This would imply that the air parcel basically looses all its previous humidity by mixing out and takes up only humidity that has just been evaporated at this location. The isotope signature of the air parcel thus is fully determined by the freshly evaporated water. This strong assumption has to be explicitly stated.

5. p. 4, L. 4: The authors say that air parcels that sank below the PBL over land were ignored? Why then do they find a lot of moisture sources over continental Alaska in Figure 1? This is confusing.

6. p. 4, L. 4: Were 71% of all trajectories ignored or kept for the analysis?

7. p. 4, L. 5-12: For me it is not entirely clear how the trajectory starting dates were chosen. Why do the authors choose only a three hours period instead of the whole precipitation event? Why are the individual dates not weighted by the locally measured precipitation intensity to take into account that when the precipitation intensity is higher the trajectories of that date contribute more to the isotope signal? What means the "most homogeneous" three-hour time window? And why with preference to the "middle" of the event? The selection criteria should be more oriented to the quantitative contribution of moisture to precipitation in my opinion.

8. p. 4, L. 13: Does "where" mean the starting altitude? The method that is shortly described in this paragraph sounds original and the idea is interesting but it assumes that the reanalysis dataset's wind field and precipitation rate profile are equivalent with the true fields. The reanalysis data error particularly with respect to the representation of small and microscale processes are ignored. Starting trajectories from different locations around the measurement site would allow to take into account the uncertainty arising from the reanalysis data

9. p. 4, L. 25: How did the authors calculate $\bar{T}_d$ and are the average moisture source conditions computed as an arithmetic mean without taking into account the evaporative contribution to the air parcel's humidity at the different source locations?

10. p. 4, L. 30: The authors should make clear that their $\Delta T_{\text{cool}}$ is only an estimate of the total cooling that the air parcel has experienced. The same remark for the possibility of multiple moisture sources for one air parcel (see specific comment 4) is valid for cooling and precipitation along an air parcel trajectory. A trajectory can produce rain all along its path and can go through several cycles of cooling and warming. The total cooling would be obtained by integrating the temperature changes along the trajectory.

11. p. 5, L. 3-9: this way of computing $T_{\text{LCL}}$ is confusing for me. Where does Eq. 2 come from? See Bolton (1980) and Lawrence (2005).

12. p. 5, L. 12-16: The idea to use $T_d$ as a summary variable for both relative humidity with respect to sea surface temperature ($h_{\text{SST}}^{2\,\text{m}}$) and SST-effects seems not justified to me from a physical point of view. The influence of SST on $T_d$ is only indirect and a strong coupling of the ocean surface conditions with near-surface air characteristics is not necessarily given particularly at the event timescale. From a theoretical perspective and for all isotope-enabled numerical modelling experiments it is the Craig-Gordon model and thus the other two variables that are used to determine $d$ of the fresh evaporate. So I am not convinced that it is sensible to introduce a third variable that does not contain more information than the specific humidity at 2 m. Furthermore, it should be made clear in the manuscript that it is not the 2 m relative humidity that is important for the non-equilibrium fractionation part during surface evaporation but the humidity gradient towards the surface which is represented by the relative humidity at 2 m with respect to sea surface temperature ($h_{\text{SST}}^{2\,\text{m}}$). The authors should make a stronger case for why they use $T_d$ rather than the classical variables. Also the sentence "$T_d$ depends on the specific humidity of saturated air at the sea surface and on the

amount of dry air from aloft that has subsided and mixed into low altitude air" is a confusing statement.

13. p. 5, L. 17: Where does Eq. 3 come from? What is the impact of the simplification involved, the authors should add a chapter reference to Stull (2015). Why did they not use Stull (2015), Equation 4.15a or b or extract directly $T_d$ from the reanalysis dataset?

14. p. 5, L. 23: How was $mtn$ defined? Using an objective criterion or subjectively by looking at the trajectory plots?

15. p. 6, L. 2: remove parentheses.

16. p. 6, L. 10-11: It would be useful to add the geographical names in one of the panels in Figure 1.

17. p. 6, L. 16-20: Is it really the variation in the moisture source latitude that is relevant or the mean transport distance? I am not convinced about the role of Figure 2. Also see major comment 3.

18. p. 6, L. 21-32: For me this relatively long paragraph is a general discussion of the possible link between polar atmospheric circulation and the location of vapour sources and not a result from this study. Either the link with the findings in this paper should be illustrated more clearly or this section should be strongly shortened or even left out. See also my general comment 3: the link between the different timescales that are involved here is not trivial to make at this stage, a more open formulation should be chosen here.

19. p. 6, L. 24-26: In Europe several studies found that during summer the regional moisture recycling and the contribution from continental evaporation is much more important than in winter (see Sodemann and Zubler (2010) and Aemisegger et al. (2014)). Even though on p.4 L.3 the authors say that "only trajectories that sank into the PBL over the ocean" a substantial contribution of evaporation from continental Alaska is found in Spring but also in the other seasons in Figure 1. This possible contribution of continental evaporation should also be discussed as its moisture source isotope signature is different than the one from ocean evaporation.

20. p. 6, L. 3: Add mid- to high latitudes here, other studies could be cited as well (e.g. Bonne et al. (2014))

21. p. 7, L. 10-11: References to figures are confusing.

22. p. 7, L. 13: Do the authors mean the regression slopes? It would be useful to add the units of the slopes in all tables. Also in Table 3 it would be useful to add the explanation on what $\beta$ and S.E. are.

23. p. 7, L. 27: Here and elsewhere the references should be listed chronologically.

24. p. 7, L. 34 - p. 8, L. 10: Here more detailed explanations on the theoretical cooling/Rayleigh experiment are needed to be able to follow. Also the sensitivity range of $\delta^2 H$ to the diagnosed cooling should be put into context and compared to literature values.

25. p. 7, L. 21: Table 1: do the regression slopes from Table 1 result from multiple linear regression?

26. p. 9, L. 23: "within storm" is a confusing term here as it suggests that the precipitation is due to the passage of a cyclone, which is not always the case. I would suggest using "intra-event" instead.

27. p. 10, L. 17: I am surprised at the $d$–$h$ slope which is not at all in agreement (opposite sign and different order of magnitude) with other literature values ($\sim$-0.6‰%$^{-1}$ to -0.32‰%$^{-1}$, though a difference with literature values is that $h_{2\,m}$ is used and not $h_{SST}^{2\,m}$). This mismatch should be explained and the relevant literature should be cited (Pfahl and Wernli, 2008; Steen-Larsen et al., 2014; Aemisegger et al., 2014). Also the $d$-SST regression slope is of opposite sign to what we would expect from the Craig-Gordon model.

28. p. 10, L. 20: What is the theoretical expectation for the sign of the correlation between $d$ and $T_d$? This should be explained in more detail. I do not agree with the statement made here, I would expect a negative $d$-$T_d$ slope from theory since the physical relation between relative humidity and $T_d$ should generally lead to a positive correlation between the latter two (see e.g. Lawrence (2005)).

29. Figures 3 and 4: more details are needed on the used spline fits. Also the strong inter-event variability that is sometimes of similar amplitude as the seasonal cycle should be discussed.

30. Figure 5: the role of this Figure is unclear to me, it is only referenced once and not further discussed in the text. Either this Figure should be better embedded in the text or it should be left out. If it is kept: is this figure an average over all events?

**References**

Aemisegger, F., S. Pfahl, H. Sodemann, I. Lehner, S. I. Seneviratne, and H. Wernli, 2014: Deuterium excess as a proxy for continental moisture recycling and plant transpiration. *Atmos. Chem. Phys.*, **14 (8)**, 4029–4054, doi:10.5194/acp-14-4029-2014.

Aemisegger, F., J. K. Spiegel, S. Pfahl, H. Sodemann, W. Eugster, and H. Wernli, 2015: Isotope meteorology of cold front passages: A case study combining observations and modeling. *Geophysical Research Letters*, **42 (13)**, 2015GL063 988, doi:10.1002/2015GL063988.

Bolton, D., 1980: The Computation of Equivalent Potential Temperature. *Monthly Weather Review*, **108 (7)**, 1046–1053, doi:10.1175/1520-0493(1980)108¡1046:TCOEPT¿2.0.CO;2.

Bonne, J.-L., V. Masson-Delmotte, O. Cattani, M. Delmotte, C. Risi, H. Sodemann, and H. C. Steen-Larsen, 2014: The isotopic composition of water vapour and precipitation in Ivittuut, southern Greenland. *Atmospheric Chemistry and Physics*, **14 (9)**, 4419–4439, doi:10.5194/acp-14-4419-2014.

Lawrence, M. G., 2005: The Relationship between Relative Humidity and the Dewpoint Temperature in Moist Air: A Simple Conversion and Applications. *Bulletin of the American Meteorological Society*, **86 (2)**, 225–233, doi:10.1175/BAMS-86-2-225.

Pfahl, S. and H. Wernli, 2008: Air parcel trajectory analysis of stable isotopes in water vapor in the eastern Mediterranean. *Journal of Geophysical Research: Atmospheres*, **113 (D20)**, D20 104, doi:10.1029/2008JD009839.

Pfahl, S., H. Wernli, and K. Yoshimura, 2012: The isotopic composition of precipitation from a winter storm – a case study with the limited-area model COSMOiso. *Atmos. Chem. Phys.*, **12 (3)**, 1629–1648, doi:10.5194/acp-12-1629-2012.

Sodemann, H., C. Schwierz, and H. Wernli, 2008: Interannual variability of Greenland winter precipitation sources: Lagrangian moisture diagnostic and North Atlantic Oscillation influence. *Journal of Geophysical Research: Atmospheres*, **113 (D3)**, D03 107, doi:10.1029/2007JD008503.

Sodemann, H. and E. Zubler, 2010: Seasonal and inter-annual variability of the moisture sources for Alpine precipitation during 1995–2002. *International Journal of Climatology*, **30 (7)**, 947–961, doi:10.1002/joc.1932.

Steen-Larsen, H. C., et al., 2014: Climatic controls on water vapor deuterium excess in the marine boundary layer of the North Atlantic based on 500 days of in situ, continuous measurements. *Atmospheric Chemistry and Physics*, **14 (15)**, 7741–7756, doi:10.5194/acp-14-7741-2014.

Stull, R., 2015: *Practical Meteorology: An Algebra-Based Survey of Atmospheric Science, Chapter 4, Water Vapor.* Univ. of British Columbia.

---

## Author Comment (AC1) · 1 Dec 2016

J.-L. Bonne (Referee) jean-louis.bonne@awi.de

General comments
This paper presents a new dataset of isotopic composition of precipitation sampled in the Barrow, AK, Arctic station, together with an innovative method to analyze and interpret its seasonal and event time scales variations. The authors propose interesting tools to use the Lagrangian atmospheric backtrajectory model for a quantitative and statistical evaluation of the observed isotopic variations. They conclude that the seasonal variations of water isotopic values are partly due to migration of the moisture origins. They focus on the influence of three parameters which are shown to explain a large part of the observed variations of the isotopic composition: the cooling along atmospheric transport, the dew point at the moisture source and the presence of mountains along the transport. The authors made a good effort to provide a rich interpretation of their observations. The manuscript is well organized and provides necessary tables and figures. There are, however, several issues regarding discussion of the results. I recommend accepting the article after the authors address the points listed below.
*The authors thank J.L. Bonne for his helpful and insightful comments. Substantial effort was put toward fully addressing the critiques.*

Specific comments
Modeling:
The moisture source modeling used in this paper relies on strong assumptions. However, potential errors caused by these assumptions are poorly pointed out. A companion paper describing the method is currently under review and might contain these information. As this paper is not yet readable, one would need a summary of these information and eventually more details in the method description or in supplementary material, in particular concerning the points addressed below. Contrary to Sodemann et al. 2008b method, the moisture source modeling used here does not take into account variations of the specific humidity in the air parcels along the trajectory. Processes such as the lost moisture through precipitation or reevaporation of already condensed droplets along transport are not taken into account, but could have a strong impact on the isotopic composition. Can you give more details on the potential errors inherent to this moisture sources modeling? Also concerning the moisture sources modeling, moisture uptakes are assimilated to air masses sinking into the planetary boundary layer (PBL) above the ocean surface. Nothing is written about the potential presence of sea ice above the ocean in the region where the PBL is reached, which could however have a strong influence on the evaporation. Do you also take into account the sea ice cover in the region were the air parcels sink into the planetary boundary layer? For example: the moisture sources for the winter events are originating from a very wide range of latitudes. If most sources are originating from the south, some sources are coming from high latitudes, up to 85◦N (see winter sources latitudes on Figure 2). Can we really expect strong evaporation in those regions, over a potentially closed ocean? Have you checked the presence of sea ice in the moisture sources regions for this type of events?

*The reviewer has a point, although we still consider our approach adequate for our purpose. We point out that assessment of "the potential errors inherent to this moisture sources modeling" is difficult for any model because true observations of moisture sources for a given event do not exist. Any assessment would be model dependent. Admittedly, the method of tracking the air parcels' moisture evolution through time, such as the one used by Sodemann et al., (2008a), is a more sophisticated way of identifying the source near the PBL than the method presented in this paper, but the model by Sodemann et al. (2008a) is not designed for the event scale moisture tracking in that 1) their parcels do not always start within precipitating clouds, and 2) the vertical distribution of parcels does not reflect condensation rates (precipitation events). Arguably, it would be ideal to combine the two methods. However, we think that for our purpose, which is to characterize the moisture source regions for observed and measured precipitation events, the initial starting points of parcels are most important, because the height of these parcels often primary dictate the source area, in our experience.*

*Details about modeling that have been added include enhanced discussion of the validity of our 'moisture source' decisions, in particular, our decision to use the last interaction with the PBL as the vapor source. We feel that this choice is justified, though less precise than Sodemann et al (2008a)because the dominance of turbulent transport relative to advective transport within the PBL (p. 4 l. 8-14).*

*As well, the sea ice presence and concentration is recorded. Air parcels were allowed to sink over sea ice, but were only considered to be a vapor source if the sea ice concentration was < 96% in order to allow for the presence of leads to contribute vapor to the PBL (p. 4 l. 13).*

Interpretation of results:
Concerning the interpretation of results at the seasonal scale, the seasonal variations described in the article are mostly the result of the relative preponderance of different types of synoptic scale events across the seasons. The intra-seasonal variability of the different events is often on the same order of magnitude than the variations of seasonal averages, which is too rarely pointed by the author. The clarity of the explanations might benefit from a more stronger distinction of the synoptic scale and seasonal scale variations.
*This is a very good point, and useful to understand when interpreting monthly, seasonal and interannual variability. Language to clarify the similarity in magnitude of the event to event and seasonal variability has been added on p. 7 l. 31-33, and the captions of Figs 3 and 4.*

Technical corrections

Abstract: The abstract is quite long and could be more concise.
P.1, L.1 to 5: The first three sentences of the abstract could rather be at the beginning of the introduction, as they don't describe the work presented in this article but general situation of research in the domain.
*The content of these sentences is covered in the introduction, so they were removed from the abstract, which helps to reduce the length of the abstract.*
P.1, L.8: "occurred" > "occuring"
*This was changed in the text.*

*As well, we have added the following to the beginning of the abstract. "In this study, precipitation isotopic variations are linked to conditions at the moisture source region, along the transport path, and at the site of precipitation. Seventy precipitation events..." (p. 1 l. 1-2)*

Methods:
P.3, L.13-14: There might be an effect of sublimation of snow which could influence the isotopic composition of water, particularly for sunny periods, even within 24 hours. Did you make some experiments to test the evolution of fresh snow on your sampling site?
*The reviewer is correct that this is possible. However, for much of the season when Barrow receives snow, there is little sun to drive sublimation, and the Arctic tends to be quite cloudy. There is also little evidence of sublimation from the data distribution along the meteoric water line. Nonetheless, clarification was added on p. 3, l. 13-16. We have also added a paragraph at the end of Section 3.2 (p. 11, l. 13-20) with a list of all potential sources of error that contribute to the unexplained variance in d2H.*

P.3, L.14: At which temperature were the samples stored, and how long?
*Clarification was added to the text. Samples were stored at less than 5 C, shipped every 3 months, and analyzed within 6 months. (p. 3, l. 17-18)*
P.4, L.1-4: Considering that a moisture source is corresponding to an air parcel sinking into the PBL is a strong assumption. More justifications of this method would be expected. If this method is described in Putman et al. (2015), add a reference here.
*The assumption has been more clearly stated and justified in the text (p.4, l. 8-14). Also see our response, at the beginning, to "Specific comments: Modeling"*
P.4, L11-12: By "the most temporally homogeneous three-hour time window", do you mean homogeneity in the precipitation amount or in the meteorological records? Do you have particular criteria to define the preference for the middle of the event? Were the event times defined automatically or manually?
*This section has been clarified. Homogeneity is in reference to the radar returns, which give us an idea of precipitation intensity. Event times were selected manually, based on multiple streams of evidence: radar returns, sampling records and surface analysis maps.(p. 4, l. 16-24)*
P.5., L.5: "The same was done for an array...": explicit that this is to calculate Qsat,z and define Qsat,z.
*The section on p. 5, l. 21-24 was updated to reflect the calculation was performed on an array of altitudes.*
P.5, L.5-6: Explicit hz, Tz, Pz: fractional relative humidity, temperature and pressure at elevation z.

*The text has been rephrased to state this explicitly (p. 5, l. 21-23).*
P.5, L.25: Are mtn values assigned manually or automatically? If automatic, then explicit the criteria.
*Mtn was assigned manually based on maps of trajectory results. Text has been updated for clarity on p.6, l. 17-18..*

Results and discussion
P. 6, L.6-15: This is a very qualitative description of Figure 1. The mean latitude of moisture sources could be introduced before and used to give quantitative aspects to this description. This description focuses on the seasonal averages of the moisture sources, but Figure 2 shows a very strong variability at the event time scale, which can be of a larger order of magnitude than the variations of the seasonal average for the mean latitude of the moisture source. For example, some events in winter have moisture sources located as north as in summer, or even further north. The normalisation of the maps from Figure 1 can also give an impression of wider or more local moisture sources depending on the total number of events and the difference between each event. Is this description of moisture sources regions still valid for absolute values without normalization to the number of events, or for individual events instead of the average of all events?
*The authors agree that there is substantial variation among events, even within a given season. This has been indicated in the revised version on p. 7 l. 31-33, and the captions of Figs 3 and 4.*

P. 6, L. 30-32: Not clear if the last sentence refers to Feng et al. (2007).
*Modified sentence to 'Feng et al. (2007) documented southward migration of the polar cell during the last glacial maximum.' for clarity. (p. 7, l. 17)*
P. 7, L. 6-7: This sentence is really affirmative, whereas Figure 4 shows a very strong dispersion, particularly for the averaged VLAT. This affirmation should be tempered and a statistical evaluation of the spline fits and there correlations should be given, as well as the standard deviations of the data series. The seasonal scale might not be the better scale to look at.
*This is true, in particular for the $V_{lat}$ variable. The variance captured by the spline fits has been added, and the text has been adjusted to better describe the similarity in magnitude between among-event variability and mean seasonal variability. (p. 7 l. 29-31, and the captions of Figs 3 and 4.)*
P. 8, L.3: How did you choose the temperatures from 10C to -15C in you theoretical cooling experiments? What would be the effect on the slopes of a variation of these temperatures on the order of magnitude of the observed variations?
*The temperature range encompasses the temperature change experienced by most trajectories. Making the warmest temp even warmer would yield slightly shallower slopes, and making colder temperatures even colder would yield steeper slopes. The coldest average final temperatures in our dataset are substantially below -15C, though the majority of each trajectory occurs within the 10 to -15C temperature range, and all events except one begin in the selected range.(p. 8, l. 27-29)*
P. 8, L. 18: Rather write "more than 20C" instead of "> 20C".
*Changed.*
P. 9, L. 1: "amount" instead of "amounts"?
*Changed.*
P.9, L.5: How was the 7C criteria chosen? Is it close to the median of the distribution of ΔTcool?
*The 7C criterion was chosen to preserve the power of the short trajectories while encapsulating the relationship to local conditions, as described in p. 10, l. 21. Though we have presented the results as categorical, this is for simplicity. It is likely that this feature of isotope systematics is actually continuous.*
P. 9, L.12: Insert a reference to figure 6 to show the repartition of small and large ΔTcool across seasons.
*Added phrase to sentence: '...and Figure 6 indicates that all the seasons are distributed across the two categories.'*
P. 10, L.6: This is not directly about precipitation d-excess but can be of interest: some studies of water vapour d-excess in Arctic regions have depicted a partial conservation of the source d-excess signal under certain atmospheric transport conditions, with relations between observed d-excess and moisture source relative humidity.

Bonne, J.-L., Masson-Delmotte, V., Cattani, O., Delmotte, M., Risi, C., Sodemann, H., and Steen-Larsen, H. C.: The isotopic composition of water vapour and precipitation in Ivittuut, southern Greenland, Atmos. Chem. Phys., 14, 4419-4439, doi:10.5194/acp- 14-4419-2014, 2014.

Bonne, J.-L., et al. (2015), The summer 2012 Greenland heat wave: In situ and remote sensing observations of water vapor isotopic composition during an atmospheric river event, J. Geophys. Res. Atmos., 120, 2970–2989, doi:10.1002/2014JD022602.

Steen-Larsen, H. C., A. E. Sveinbjörnsdottir, Th. Jonsson, F. Ritter, J.-L. Bonne, V. Masson-Delmotte, H. Sodemann, T. Blunier, D. Dahl-Jensen, and B. M. Vinther (2015), Moisture sources and synoptic to seasonal variability of North Atlantic water vapor isotopic composition, J. Geophys. Res. Atmos., 120, 5757–5774, doi:10.1002/ 2015JD023234.
*Interesting work, thank you for the citations. The discussion of d-excess has been updated to include these publications, on p. 11, l. 31-32.*

Conclusions
P. 10, L. 29-31: This conclusion on the origins of moisture is valid for the average of the seasonal moisture sources, but should be tempered by pointing out the event to event variation of the moisture sources.
*This caveat was added to the conclusions "However, substantial intra-season variability occurs in both source and d2H, indicating scatter in the seasonal relationship." (p. 12, l. 30-31)*

References
P.13, L. 32-36: Logically, the two papers numbering should be inverted (2008a and 2008b).
*Changed.*

Tables and figures
Table 1 and 2: The legends do not clearly describe the contents of the tables. Why are different intercepts given for each variable in Table 2 and only one value in Table 1, if the only difference between the two tables are the division of all samples in two groups?
*The captions of the tables have been updated to explain that Table 1 contains the results from a single multivariable regression, while Table 2 contains the results from 3 simple linear regressions.*

Figure 7: Parenthesis not closed in right y-axis label.
*Fixed.*
Figure 3 and 7: It would be more readable with x-axis ticks corresponding to the beginning of the years instead of the beginning of each December.
*Fixed.*

---

## Author Comment (AC2) · 1 Dec 2016

Review of
*"Annual variation in precipitation $\delta^2 H$ reflects vapour source region at Barrow, AK"*
by A. L. Putman et al.
Paper published in ACPD on 11 August 2016

**1 General Comments**

This paper presents an interesting dataset of the event-scale $\delta_2 H$ and deuterium excess signature of precipitation from northern Alaska. The authors use a very simple back-trajectory-based analysis of the transport and moisture source conditions which they summarise in 3 main characteristics to interpret their data. These are 1) the moisture source dew point temperature at 2m, 2) the total cooling between the lifted condensation level at the moisture source and the precipitation level in the cloud at the measurement site (arrival temperature) and 3) whether the air parcels that are transported to the measurement site across the Brooks and/or the Alaskan ranges. I recommend publication of this overall well-written manuscript, but I have four major concerns that should be addressed beforehand as well as a many specific comments listed below:

*Thanks to the reviewer for the useful points and ideas. We have considered the suggestions, addressed the questions and revised the paper accordingly, and we hope that the revisions are satisfactory to this reviewer.*

1 **Moisture source identification and particularly the implicit assumptions made:**
see specific comments 3-7.

*The authors argue that the method employed in this paper is adequate for our purpose. Please see our responses for comments 3-7 for the full discussion.*

2 **Choice of the parameters that explain the variance of the isotope signature of precipitation in Barrow:**
For me the choice of the parameters that were used to explain the precipitation isotope signal in Barrow seems random. It makes sense to look at moisture source and transport conditions but in my opinion there is no reason for completely neglecting the local conditions. Particularly at Barrow, the precipitation phase (liquid or snow) probably plays an important role for the end isotope composition of the precipitation event as it determines whether there is isotopic exchange (for rain drops, see specific comment 2) or not (for snowfall) with the local vapour. Also precipitation intensity plays an important role. The authors have some detailed information about the precipitation structure from their radar data and could use this to try to further understand the local processes. If this is done in an other paper, then this should be clearly stated. Also I do not fully support the choice of the variable $T_d$ as representative for the moisture source conditions (see specific comment 12).

*This is a good point, and one that was considered by the authors before settling on the variables reported. Indeed, half the variance in d2H cannot be explained by the 3 variables chosen! The main reason that other variables (including but not limited to precipitation phase, sub-cloud dryness, precipitation intensity, evaporation below the cloud base, supersaturation in the cloud, and storm event type) were not included is because, the statistical power of the limited number of events we were able to consider is not sufficiently high to go after each of those potentially very important variables. When such variables were included in the analysis, they did not explain any more variance. This may be because the isotopic responses to them are not related to d2H variations, or are related but not sufficiently above noise. For example, sub-cloud dryness may be important for some but not all events, dryness may occur during both high and low d2H events, but the power or the size of the signal may be limited. Nevertheless, to respond to this point, we added a paragraph at the end of Section 3.2 (p. 11 l. 13-20) that includes a list of variables potentially contributing to the 46% of the unexplained variance in d2H.*

3 **Expansion of the northern polar circulation cell and its link to moisture source location**
The link between the event-based moisture source location of precipitation and the polar circulation cell is described in a very qualitative way. A link between the weather systems driving the moisture transport at the event timescale leading to precipitation at Barrow and the more climatological description of the polar circulation is not obvious and not trivial to make. The formulations used throughout the paper should be more careful and kept as hypotheses.

*The authors acknowledge that the relationship between vapor source and circulation is not simple, and the link between the annual and longer timescales is a possibility, not a certainty. However, the work does substantiate the idea that isotope values measured in ice cores may reflect changes in circulation patterns as well as local temperature, which is how they are often interpreted. The phrasing of these statements has been re-formulated in all discussion to suggest hypothetical as opposed to likely links. (p. 7 l. 17-23, p.13, l. 9-12)*

4 **Critical discussion of results in view of the existing literature**:
in particular see specific comments 24 and 27.
*More discussion has been added to Sections 3.2 and 3.3. In 3.2, which discusses the influence of vapor source on measured precipitation isotopes, greater clarification of the simple Rayleigh model used to contextualize our results has been added, as well as a comprehensive sources of error paragraph (p. 11 l. 13-20) and an expanded discussion of the utility of Td (p.9 l. 7-35, p 10 l 1-10) in characterizing the source. In section 3.3, the d-excess results are discussed in greater depth in light of the suggested papers (p. 11, l. 31). In particular, we have added discussion of the relationship between local water vapor and evaporation conditions (p. 12 l. 13-17).*

**2 Specific comments**
1. p. 1, title: It would be nice to include in the title the fact that it is event-scale precipitation samples that the authors analyse in this paper. Something like: "Annual variation in event-scale precipitation $_2$H reflects vapour source region at Barrow, AK". Also Barrow, AK could be replaced by northern Alaska.
*Title changed as suggested.*

2. p. 18-23: The local conditions during cloud formation and during precipitation also play an important role for the isotope composition of precipitation. For rainfall for example below cloud effects (evaporation and exchange with ambient vapour) can have a strong impact on the isotope composition of precipitation (20-40h for $\delta_2$H, see Pfahl et al. (2012), Aemisegger et al. (2015)).
*This is absolutely true, and we did experiment with including condensation temperature, precipitation type, and sub-cloud humidity in our regressions. The regression presented was the best model in terms of simplicity and variance explained by different parameters. One reason why these local factors may not have been significant influences to our dataset is because of event-to-event variability. e.g., in one case enrichment may be due to sub-cloud evaporation, but in another it may be due to condensation temperature, and within our dataset we did not have the statistical power to disentangle these competing mechanisms. The text has been updated on p. 11 l. 13-20. Also see our response to General Comments 2).*

3. p. 3, L. 29: The reanalysis dataset (wind fields) that is used for the trajectory calculation should be mentioned here as well as its horizontal resolution.
*The information has been added on p. 4 l.3-4.*

4. p. 4, L. 2: What do the authors mean with "The first time"? Is the time reference forward or backward? Does that mean the first time when following the trajectory back from the arrival point? And does that mean that one trajectory can have only 1 associated moisture source? This would be a very strong assumption about the moisture source location. Uptakes of moisture can happen all along an air parcel's trajectory (see Sodemann et al. (2008)) and they can sometimes be linked to surface evaporation even though they are not in the boundary layer (PBL), particularly over land. If for each trajectory only the latest passage in the PBL before arrival at the measurement site is considered then this means that the authors assume very strong mixing. This would imply that the air parcel basically looses all its previous humidity by mixing out and takes up only humidity that has just been evaporated at this location. The isotope signature of the air parcel thus is fully determined by the freshly evaporated water. This strong assumption has to be explicitly stated.
*The 'first time' is in reference to back trajectories. Wording in the manuscript has been updated for clarity on p.4 l. 8-14. Yes, each trajectory has one associated vapor source. Though the method described in Sodemann (2008) is a substantially more sophisticated way of identifying the vapor source, it is not necessary in our work for three reasons. 1) For a given parcel, the spatial range over which the parcel moves up and down across the PBL, is small compared to the region covered by 1000 total parcels of an event. The latter is primarily dictated by the vertical distribution of the initial parcels' altitude 2) Our work analyzes the influence of marine source areas on Barrow precipitation. Marine surface conditions are relatively homogeneous, which point strengthens the argument in 1). 3)*

*Averaging at the precipitation site of condensate of 1000 trajectories from a wide spatial distribution of source locations implicitly accounts for mixing of moisture from distributed source locations. It is thus in effect equivalent to the more sophisticated Sodemann et al. (2008) model which used on average only 2.6 trajectories per column of air per time window, even if each of those trajectories combined source points at its inception.*

5. p. 4, L. 4: The authors say that air parcels that sank below the PBL over land were ignored? Why then do they find a lot of moisture sources over continental Alaska in Figure 1? This is confusing.
*The difference between data used for statistics and data used in figure is confusing. Text is added that notes that all trajectories are shown in the figure, but only specific trajectories are used for the calculations (p.4, l. 12-14). This is noted in the figure caption as well.*

6. p. 4, L. 4: Were 71% of all trajectories ignored or kept for the analysis?
*Changed to 'Ocean originating air parcels' to clarify (p. 4 l. 15)*

7. p. 4, L. 5-12: For me it is not entirely clear how the trajectory starting dates were chosen. Why do the authors choose only a three hours period instead of the whole precipitation event? Why are the individual dates not weighted by the locally measured precipitation intensity to take into account that when the precipitation intensity is higher the trajectories of that date contribute more to the isotope signal? What means the "most homogeneous" three-hour time window? And why with preference to the "middle" of the event? The selection criteria should be more oriented to the quantitative contribution of moisture to precipitation in my opinion.
*The three hour time period was chosen so that each event was treated the same. In some cases, multiple 3 hour windows were analyzed, and typically showed very similar results, so we concluded that choosing one three hour window was representative of the whole event. Selection of the specific three hour time period was not performed quantitatively. The qualitative selection used returns from the MMCR and KAZR. Higher Doppler vertical velocity and reflectivity indicate increased precipitation intensity. These criteria, in conjunction with surface analysis maps were used to determine the start and end times. The three hour window reflects the constraints of the reanalysis, i.e. the temporal resolution is three hours. For all of these questions, clarification has been added to the text on p. 4, l 15-24.*

8. p. 4, L. 13: Does "where" mean the starting altitude? The method that is shortly described in this paragraph sounds original and the idea is interesting but it assumes that the reanalysis dataset's wind field and precipitation rate profile are equivalent with the true fields. The reanalysis data error particularly with respect to the representation of small and microscale processes are ignored. Starting trajectories from different locations around the measurement site would allow to take into account the uncertainty arising from the reanalysis data
*Yes, "where" means altitude. This is clarified in the text (p. 4 l. 25). Yes, wind direction issues are very important for back trajectories, However, if winds are incorrect, incorporating a wider area will not make them more correct. Furthermore, because the resolution of the reanalysis data is 1x1 degree, using multiple locations may cover a wide region, hundreds of kilometers in size. Such a wide spatial scale could be less representative of local or small-region precipitation events, though it could be helpful for large precipitation events. We are not convinced that looking over a larger spatial location would improve the vapor source estimation. However, sending a large number of air parcels (~1000), as we have done, helps to deal with the wind issue. Wind errors in the reanalysis are a potential source of error for any Lagrangian back trajectory study, and are not unique to our study. Therefore, we hope people who are reading studies using Lagrangian back trajectories are in general cautious with this type of reanalysis product.*

9. p. 4, L. 25: How did the authors calculate $T_d$ and are the average moisture source conditions computed as an arithmetic mean without taking into account the evaporative contribution to the air parcel's humidity at the different source locations?
*Because of the way the condensation profile is divided, it is assumed that each parcel contributes an equal amount of vapor to the final precipitating cloud, so we did not weight them. The calculation of Td is described in detail on p. 6 l. 3-12.*

10. p. 4, L. 30: The authors should make clear that their $T_{cool}$ is only an estimate of the total cooling that the air parcel has experienced. The same remark for the possibility of multiple moisture sources for one air parcel (see

specific comment 4) is valid for cooling and precipitation along an air parcel trajectory. A trajectory can produce rain all along its path and can go through several cycles of cooling and warming. The total cooling would be obtained by integrating the temperature changes along the trajectory.

*The reviewer is correct, the cooling indicated by Tcool is the net cooling, not the integral of cycles of warming and cooling an air parcel may have experienced along its trajectory. This is a simplification. The following has been added to the text to clarify this point on p. 5 l. 10-11.*

11. p. 5, L. 3-9: this way of computing $T_{LCL}$ is confusing for me. Where does Eq. 2 come from? See Bolton (1980) and Lawrence (2005).

*The equations have been changed to those in Stull (2015). The previous equation was a linearized approximation that introduced minor, if not insignificant, discrepancy. To be more precise, the equations, calculations, and text have been updated to be consistent with Stull (2015), though the results and discussion require no change. Equations 2, 3, and 5 are affected.*

12. p. 5, L. 12-16: The idea to use $T_d$ as a summary variable for both relative humidity with respect to sea surface temperature ($h_{2m\,SST}$) and SST-effects seems not justified to me from a physical point of view. The influence of SST on $T_d$ is only indirect and a strong coupling of the ocean surface conditions with near-surface air characteristics is not necessarily given particularly at the event timescale. From a theoretical perspective and for all isotope-enabled numerical modelling experiments it is the Craig-Gordon model and thus the other two variables that are used to determine $d$ of the fresh evaporate. So I am not convinced that it is sensible to introduce a third variable that does not contain more information than the specific humidity at 2 m. Furthermore, it should be made clear in the manuscript that it is not the 2m relative humidity that is important for the non-equilibrium fractionation part during surface evaporation but the humidity gradient towards the surface which is represented by the relative humidity at 2m with respect to sea surface temperature ($h_{2m\,SST}$). The authors should make a stronger case for why they use $T_d$ rather than the classical variables. Also the sentence "$T_d$ depends on the specific humidity of saturated air at the sea surface and on the amount of dry air from aloft that has subsided and mixed into low altitude air" is a confusing statement.

*The idea of using Td is to indicate the moisture condition PBL, the moisture that forms the first condensate. This is not the same as the evaporative flux predicted by the Craig-Gordon model. Our group has done a significant amount of work to model and understand isotopic variations in the marine boundary layer (manuscripts in preparation), and our understanding continues to improve. We realized that our discussion about Td in the earlier version was not clear, and it is valid for this reviewer to solicit further explanation. We have completely rewritten section 3.2 that pertains to Td (p. 9 l. 6-35, p. 10 l. 1-9). Basically, the Craig-Gordon model only predicts the evaporative flux, not the vapor properties in the PBL. In addition, the Craig-Gordon model does not consider effects of convection on vapor isotopic ratios in the PBL. However, convection is very important process that 1) transports PBL air to the free troposphere, and 2) brings dry air from aloft to the PBL. The boundary layer air is therefore a mixture of evaporated vapor from the ocean surface, and the dry air from aloft. The extent of this mixing within the PBL is reflected by (2m) dew point, Td. Td is also important for indicating the evaporation condition in that it is more directly related to relative humidity with respect to the sea surface temperature than is the 2 m relative humidity. We hope the new discussion in the revised version is clearer.*

13. p. 5, L. 17: Where does Eq. 3 come from? What is the impact of the simplification involved, the authors should add a chapter reference to Stull (2015). Why did they not use Stull (2015), Equation 4.15a or b or extract directly $T_d$ from the reanalysis dataset?

*Calculation was done with 4.15b in Stull (2015).*

14. p. 5, L. 23: How was *mtn* defined? Using an objective criterion or subjectively by looking at the trajectory plots?

*Removed 'for the event, not to individual trajectories' and added 'observed in trajectory plots' to clarify how mtn was defined.*

15. p. 6, L. 2: remove parentheses.

*Parenthesis removed.*

16. p. 6, L. 10-11: It would be useful to add the geographical names in one of the panels in Figure 1.

*A nice idea, but would obscure the data presented in the plot because it would be too busy.*

17. p. 6, L. 16-20: Is it really the variation in the moisture source latitude that is relevant or the mean transport distance? I am not convinced about the role of Figure 2. Also see major comment 3.

*Because most vapor transport is from mid-latitudes to high latitudes, latitude is actually relevant for this site. Yes, distance might be another reasonable metric to investigate. However, latitude was chosen because latitude covaries with evaporation conditions, so it's more physically useful than distance.*

*Figure 2 is useful as it shows the relationship of latitude to vapor source and distillation. A similar figure could have been made for distance, but the outcomes would be very similar, as in our case distance variation is roughly the same as that of latitude.*

18. p. 6, L. 21-32: For me this relatively long paragraph is a general discussion of the possible link between polar atmospheric circulation and the location of vapour sources and not a result from this study. Either the link with the findings in this paper should be illustrated more clearly or this section should be strongly shortened or even left out. See also my general comment 3: the link between the different timescales that are involved here is not trivial to make at this stage, a more open formulation should be chosen here.

*The link between seasonal changes to general circulation and seasonal change in vapor source makes sense because general circulation is the background pattern from which weather events deviate. This paragraph only links the seasonality of vapor source with the seasonality of circulation patterns- nothing over longer timescales. However, the language has been updated to be less causal. (p. 7 l. 17-23)*

19. p. 6, L. 24-26: In Europe several studies found that during summer the regional moisture recycling and the contribution from continental evaporation is much more important than in winter (see Sodemann and Zubler (2010) and Aemisegger et al. (2014)). Even though on p.4 L.3 the authors say that "only trajectories that sank into the PBL over the ocean" a substantial contribution of evaporation from continental Alaska is found in Spring but also in the other seasons in Figure 1. This possible contribution of continental evaporation should also be discussed as its moisture source isotope signature is different than the one from ocean evaporation.

*Local evapotranspiration during spring and summer is likely an important vapor source. However, given the heterogeneity of the event conditions and sources, we did not have the statistical power to pull evapotranspiration out relative to the other factors. Nonetheless, in the discussion, evapotranspiration is added as a third potential mechanism of seasonal change (p. 7 l. 9-14), as it likely does contribute to the d2H measured in precipitation. Ignoring continental air may also contribute to the unexplained variance in the multiple regression in Section 3.2. We included discussion of this factor in the new version at the end of section 3.2 (p. 11 l. 13-20)*

20. p. 6, L. 3: Add mid- to high latitudes here, other studies could be cited as well (e.g. Bonne et al. (2014))
*The phrase is changed, and citation added.*

21. p. 7, L. 10-11: References to figures are confusing.
*The references to figures were removed.*

22. p. 7, L. 13: Do the authors mean the regression slopes? It would be useful to add the units of the slopes in all tables. Also in Table 3 it would be useful to add the explanation on what $\beta$ and S.E. are.
*Changed to regression slopes. The units, originally just of the variable, are now the units of the slope in all tables. Beta and S.E. are described in the text that refers to Table 3.*

23. p. 7, L. 27: Here and elsewhere the references should be listed chronologically.
*Checked references throughout manuscript and reordered where necessary.*

24. p. 7, L. 34 - p. 8, L. 10: Here more detailed explanations on the theoretical cooling/Rayleigh experiment are needed to be able to follow. Also the sensitivity range of _2H to the diagnosed cooling should be put into context and compared to literature values.
*The model is explained in greater detail given the following: 'In the simple model a saturated air parcel with a specified temperature and vapor d2H is cooled by 1C steps. At each temperature step the condensation amount, remaining vapor, d2H and vapor d2H are calculated. No re-evaporation or non-equilibrium conditions are considered.' The simple model is meant to contextualize the numbers. (p. 8 l. 27-29)*

25. p. 7, L. 21: Table 1: do the regression slopes from Table 1 result from multiple linear regression?

*Yes, this is stated in the text (original p. 7 l. 11) though it has been added to the Table caption now. The regressions in Tables 2 and 3 are now clearly indicated as simple linear regressions.*

26. p. 9, L. 23: "within storm" is a confusing term here as it suggests that the precipitation is due to the passage of a cyclone, which is not always the case. I would suggest using "intra-event" instead.
*Changed to intra-event.*

27. p. 10, L. 17: I am surprised at the $d$–$h$ slope which is not at all in agreement (opposite sign and different order of magnitude) with other literature values ($d$-0.6h%$^{-1}$ to -0.32h%$^{-1}$, though a difference with literature values is that $h_{2m}$ is used and not $h_{2m}$ SST). This mismatch should be explained and the relevant literature should be cited (Pfahl and Wernli, 2008; Steen-Larsen et al., 2014; Aemisegger et al., 2014). Also the $d$-SST regression slope is of opposite sign to what we would expect from the Craig-Gordon model.
*We too were also surprised at the outcome. Efforts were made in the text to explain it, focusing on the effect of the larger-scale humidity gradients and potential mix-phased cloud effects. However, considering that evaporative condition is only one process controlling the vapor properties in the PBL, as we explained earlier and also in the revised manuscript, this result is not entirely unreasonable. See our response for 28 below.*

28. p. 10, L. 20: What is the theoretical expectation for the sign of the correlation between $d$ and $T_d$? This should be explained in more detail. I do not agree with the statement made here, I would expect a negative $d$-$T_d$ slope from theory since the physical relation between relative humidity and $T_d$ should generally lead to a positive correlation between the latter two (see e.g. Lawrence (2005)).
*Our mistake. Thank you for catching this. We now discuss this negative relationship in the context of two processes, 1) dry air (low Td) causing larger kinetic fractionation and higher d, and 2) the descending air may have high d values. Both processes, independently and together yield negative association between d and Td. We now also state that this result is consistent with our argument that Td is more representative of the PBL conditions than are Tss and h. (p. 12 l 13-17)*

29. Figures 3 and 4: more details are needed on the used spline fits. Also the strong inter-event variability that is sometimes of similar amplitude as the seasonal cycle should be discussed.
*Details on the spline fits have been added to the figure caption, and the similarity in amplitude among seasonal and event variability is noted for both datasets. (Figs 3 and 4, and p. 7 l. 29-34)*

30. Figure 5: the role of this Figure is unclear to me, it is only referenced once and not further discussed in the text. Either this Figure should be better embedded in the text or it should be left out. If it is kept: is this figure an average over all events?
*Yes, this figure shows the average value of d2H of precipitation coming from a specific vapor source, which indicates that certain regions tend to be vapor sources for precipitation events that are either more or less enriched than average for that time of year. Mountains along the trajectory appear to be the mechanism at work in producing the spatial structure. The figure and its meaning are also discussed on p. 8 l. 14.*

**References**

Aemisegger, F., S. Pfahl, H. Sodemann, I. Lehner, S. I. Seneviratne, and H. Wernli, 2014: Deuterium excess as a proxy for continental moisture recycling and plant transpiration. *Atmos. Chem. Phys.*, **14 (8)**, 4029–4054, doi:10.5194/acp-14-4029-2014.

Aemisegger, F., J. K. Spiegel, S. Pfahl, H. Sodemann, W. Eugster, and H. Wernli, 2015: Isotope meteorology of cold front passages: A case study combining observations and modeling. *Geophysical Research Letters*, **42 (13)**, 2015GL063 988, doi:10.1002/2015GL063988.

Bolton, D., 1980: The Computation of Equivalent Potential Temperature. *Monthly Weather Review*, **108 (7)**, 1046–1053, doi:10.1175/1520-0493(1980)108¡1046:TCOEPT¿2.0.CO;2.

Bonne, J.-L., V. Masson-Delmotte, O. Cattani, M. Delmotte, C. Risi, H. Sodemann, and H. C. Steen-Larsen, 2014: The isotopic composition of water vapour and precipitation in Ivittuut, southern Greenland. *Atmospheric Chemistry and Physics*, **14 (9)**, 4419–4439, doi:10.5194/acp-14-4419-2014.

Lawrence, M. G., 2005: The Relationship between Relative Humidity and the Dewpoint Temperature in Moist Air: A Simple Conversion and Applications. *Bulletin of the American Meteorological Society*, **86 (2)**, 225–233, doi:10.1175/BAMS-86-2-225.

Pfahl, S. and H. Wernli, 2008: Air parcel trajectory analysis of stable isotopes in water vapor in the eastern Mediterranean. *Journal of Geophysical Research: Atmospheres*, **113 (D20)**, D20 104, doi: 10.1029/2008JD009839.

Pfahl, S., H. Wernli, and K. Yoshimura, 2012: The isotopic composition of precipitation from a winter storm – a case study with the limited-area model COSMOiso. *Atmos. Chem. Phys.*, **12 (3)**, 1629–1648, doi:10.5194/acp-12-1629-2012.

Sodemann, H., C. Schwierz, and H. Wernli, 2008: Interannual variability of Greenland winter precipitation sources: Lagrangian moisture diagnostic and North Atlantic Oscillation influence. *Journal of Geophysical Research: Atmospheres*, **113 (D3)**, D03 107, doi:10.1029/2007JD008503.

Sodemann, H. and E. Zubler, 2010: Seasonal and inter-annual variability of the moisture sources for Alpine precipitation during 1995–2002. *International Journal of Climatology*, **30 (7)**, 947–961, doi: 10.1002/joc.1932.

Steen-Larsen, H. C., et al., 2014: Climatic controls on water vapor deuterium excess in the marine boundary layer of the North Atlantic based on 500 days of in situ, continuous measurements. *Atmospheric Chemistry and Physics*, **14 (15)**, 7741–7756, doi:10.5194/acp-14-7741-2014.

Stull, R., 2015: *Practical Meteorology: An Algebra-Based Survey of Atmospheric Science, Chapter 4, Water Vapor*. Univ. of British Columbia.

*All of these suggested references have been checked and cited when appropriate.*

---

## Author Comment (AC3) · 1 Dec 2016

**Review of**
**"Annual variation in precipitation δ²H reflects vapour source region at Barrow, AK"**
by A. L. Putman et al.
Paper published in ACPD on 11 August 2016

*The following are additional responses to the following questions that concern the utility of h2m,SST. Additional responses are marked in green.*

12. p. 5, L. 12-16: The idea to use $T_d$ as a summary variable for both relative humidity with respect to sea surface temperature ($h_{2m\,SST}$) and SST-effects seems not justified to me from a physical point of view. The influence of SST on $T_d$ is only indirect and a strong coupling of the ocean surface conditions with near-surface air characteristics is not necessarily given particularly at the event timescale. From a theoretical perspective and for all isotope-enabled numerical modelling experiments it is the Craig-Gordon model and thus the other two variables that are used to determine $d$ of the fresh evaporate. So I am not convinced that it is sensible to introduce a third variable that does not contain more information than the specific humidity at 2 m. Furthermore, it should be made clear in the manuscript that it is not the 2m relative humidity that is important for the non-equilibrium fractionation part during surface evaporation but the humidity gradient towards the surface which is represented by the relative humidity at 2m with respect to sea surface temperature ($h_{2m\,SST}$). The authors should make a stronger case for why they use $T_d$ rather than the classical variables. Also the sentence "$T_d$ depends on the specific humidity of saturated air at the sea surface and on the amount of dry air from aloft that has subsided and mixed into low altitude air" is a confusing statement.

*The idea of using Td is to indicate the moisture condition PBL, the moisture that forms the first condensate. This is not the same as the evaporative flux predicted by the Craig-Gordon model. Our group has done a significant amount of work to model and understand isotopic variations in the marine boundary layer (manuscripts in preparation), and our understanding continues to improve. We realized that our discussion about Td in the earlier version was not clear, and it is valid for this reviewer to solicit further explanation. We have completely rewritten section 3.2 that pertains to Td (p. 9 l. 6-35, p. 10 l. 1-9). Basically, the Craig-Gordon model only predicts the evaporative flux, not the vapor properties in the PBL. In addition, the Craig-Gordon model does not consider effects of convection on vapor isotopic ratios in the PBL. However, convection is very important process that 1) transports PBL air to the free troposphere, and 2) brings dry air from aloft to the PBL. The boundary layer air is therefore a mixture of evaporated vapor from the ocean surface, and the dry air from aloft. The extent of this mixing within the PBL is reflected by (2m) dew point, Td. Td is also important for indicating the evaporation condition in that it is more directly related to relative humidity with respect to the sea surface temperature than is the 2 m relative humidity. When h2m, SST was used in the multiple regression instead of Td, it was a significant predictor of d2H. However, in both variance explained and AIC, the the multiple regression that incorporated Td performed better. Both because it performs better in the multiple regression, and because it is a measurable quantity, we prefer Td to h_2m,SST and have retained it in the paper. We hope the new discussion in the revised version is clearer.*

27. p. 10, L. 17: I am surprised at the $d$–$h$ slope which is not at all in agreement (opposite sign and different order of magnitude) with other literature values ($d$-0.6h%⁻¹ to -0.32h%⁻¹, though a difference with literature values is that $h_{2m}$ is used and not $h_{2m\,SST}$). This mismatch should be explained and the relevant literature should be cited (Pfahl and Wernli, 2008; Steen-Larsen et al., 2014; Aemisegger et al., 2014). Also the $d$-SST regression slope is of opposite sign to what we would expect from the Craig-Gordon model.

*We too were surprised at the outcome. Efforts were made in the text to explain it, focusing on the effect of the larger-scale humidity gradients and potential mix-phased cloud effects. However, considering that evaporative condition is only one process controlling the vapor properties in the PBL, as we explained earlier and also in the revised manuscript, this result is not entirely unreasonable. See our response for 28 below.*

*The statistics were performed with h2m,SST, and the results were significant (p < 0.001), and in the range of the values described above:- 0.39+/- 0.067 h%-1, with 34% variance explained. This has been added to the paper in this section. Note- because we updated the sea ice threshold to 96% (per specific comment on modeling by JL Bonne), some values of source variables have changed slightly. This is why there are updated numbers in the table.*

---

## Author Response (AR1)

**Authors' response to reviewer comments**

**Journal**: Atmospheric Chemistry and Physics
**Manuscript #:** acp-2016-539
**Title**: Annual variation in event-scale precipitation $\delta^2$H at Barrow, AK reflects vapor source region
**Authors**: Annie Putman, Xiahong Feng, Leslie J. Sonder, and Eric S. Posmentier
**Date**: Dec, 27, 2016

Dear Dr. Thomas Röckmann,

We are pleased to resubmit for publication the revised version of MS acp-2016-539, 'Annual variation in event-scale precipitation $\delta^2$H at Barrow, AK reflects vapor source region'. We appreciate the constructive criticism of the reviewers, and feel that the resulting paper is stronger as a result of their input. A detailed, point-by-point response to J.L. Bonne and Anonymous Reviewer #2 are followed by a list of relevant changes, and a marked up manuscript. Note that line numbers in this document refer to the final manuscript.

The authors feel that the work detailed in the following document represents a complete response to the concerns of the reviewers. We appreciate your time and consideration as you evaluate our revised manuscript and we look forward to hearing from you in the new year.

Kind Regards,

Annie Putman

Interactive comment on "Annual variation in precipitation δ 2H reflects vapor source region at Barrow, AK"
by Annie L. Putman et al.
J.-L. Bonne (Referee) jean-louis.bonne@awi.de

General comments
This paper presents a new dataset of isotopic composition of precipitation sampled in the Barrow, AK, Arctic
station, together with an innovative method to analyze and interpret its seasonal and event time scales variations.
The authors propose interesting tools to use the Lagrangian atmospheric backtrajectory model for a quantitative and
statistical evaluation of the observed isotopic variations. They conclude that the seasonal variations of water isotopic
values are partly due to migration of the moisture origins. They focus on the influence of three parameters which are
shown to explain a large part of the observed variations of the isotopic composition: the cooling along atmospheric
transport, the dew point at the moisture source and the presence of mountains along the transport. The authors made
a good effort to provide a rich interpretation of their observations. The manuscript is well organized and provides
necessary tables and figures. There are, however, several issues regarding discussion of the results. I recommend
accepting the article after the authors address the points listed below.
*The authors thank J.L. Bonne for his helpful and insightful comments. Substantial effort was put toward fully
addressing the critiques. Line numbers provided correspond to the final document.*

Specific comments
Modeling:
The moisture source modeling used in this paper relies on strong assumptions. However, potential errors caused by
these assumptions are poorly pointed out. A companion paper describing the method is currently under review and
might contain these information. As this paper is not yet readable, one would need a summary of these information
and eventually more details in the method description or in supplementary material, in particular concerning the
points addressed below. Contrary to Sodemann et al. 2008b method, the moisture source modeling used here does
not take into account variations of the specific humidity in the air parcels along the trajectory. Processes such as the
lost moisture through precipitation or reevaporation of already condensed droplets along transport are not taken into
account, but could have a strong impact on the isotopic composition. Can you give more details on the potential
errors inherent to this moisture sources modeling? Also concerning the moisture sources modeling, moisture uptakes
are assimilated to air masses sinking into the planetary boundary layer (PBL) above the ocean surface. Nothing is
written about the potential presence of sea ice above the ocean in the region where the PBL is reached, which could
however have a strong influence on the evaporation. Do you also take into account the sea ice cover in the region
were the air parcels sink into the planetary boundary layer? For example: the moisture sources for the winter events
are originating from a very wide range of latitudes. If most sources are originating from the south, some sources are
coming from high latitudes, up to 85◦N (see winter sources latitudes on Figure 2). Can we really expect strong
evaporation in those regions, over a potentially closed ocean? Have you checked the presence of sea ice in the
moisture sources regions for this type of events?

*The reviewer has a point, although we still consider our approach adequate for our purpose. We point out that
assessment of "the potential errors inherent to this moisture sources modeling" is difficult for any model because
true observations of moisture sources for a given event do not exist. Any assessment would be model dependent.
Admittedly, the method of tracking the air parcels' moisture evolution through time, such as the one used by
Sodemann et al., (2008a), is a more sophisticated way of identifying the source near the PBL than the method
presented in this paper, but the model by Sodemann et al. (2008a) is not designed for the event scale moisture
tracking in that 1) their parcels do not always start within precipitating clouds, and 2) the vertical distribution of
parcels does not reflect condensation rates (precipitation events). Arguably, it would be ideal to combine the two
methods. However, we think that for our purpose, which is to characterize the moisture source regions for observed
and measured precipitation events, the initial starting points of parcels are most important, because the height of
these parcels often primary dictate the source area, in our experience.*

*Details about modeling that have been added include enhanced discussion of the validity of our 'moisture source'
decisions, in particular, our decision to use the last interaction with the PBL as the vapor source. We feel that this
choice is justified, though less precise than Sodemann et al (2008a) because the dominance of turbulent transport
relative to advective transport within the PBL.*

*As well, the sea ice presence and concentration is recorded. Air parcels were allowed to sink over sea ice, but were only considered to be a vapor source if the sea ice concentration was < 96% in order to allow for the presence of leads to contribute vapor to the PBL.*

p.4, l. 7-14:
'Relative to previous studies that tracked vapor change in an air parcel along the trajectory (e.g., Sodemann et al. (2008a)), we adopted a simpler procedure that assumes vapor in the air parcel is well represented by the air at the latest interaction with the PBL. This assumption is justified because mass movement in the PBL is dominated by vertical turbulence relative to horizontal advection. Figure 1 shows endpoints of all trajectories that sank into the PBL. However, only trajectories that ended over water with < 96% sea ice cover were used for calculations; parcels that sank where there was less than 96% sea ice cover were used for calculations.'

Interpretation of results:
Concerning the interpretation of results at the seasonal scale, the seasonal variations described in the article are mostly the result of the relative preponderance of different types of synoptic scale events across the seasons. The intra-seasonal variability of the different events is often on the same order of magnitude than the variations of seasonal averages, which is too rarely pointed by the author. The clarity of the explanations might benefit from a more stronger distinction of the synoptic scale and seasonal scale variations.
*This is a very good point, and useful to understand when interpreting monthly, seasonal and interannual variability. Language to clarify the similarity in magnitude of the event to event and seasonal variability has been added.*

 P. 7 L. 34:
'… though the inter-event variability in both variables can be as large as the seasonal variability'

P. 12 L.28-29
'…exhibited interannual, annual, and substantial inter-event variability.'

P.12 L. 33
'However, substantial intra-season variability occurred in both source and $\delta_2$H, indicating scatter in the seasonal relationship'

Caption of Fig 3
'Of the three timescales, annual variability shows the greatest amplitude, though variability among events is also substantial.'

Caption of Fig 4
'For both datasets, the variability exhibited among events is of the same the order of magnitude as the seasonal variability.'

Technical corrections

Abstract: The abstract is quite long and could be more concise.
*The following sentences at the beginning and end of the abstract were removed.*

Removed 'Interpretation of variability in precipitation stable isotopic ratios often relies exclusively on empirical relationships to meteorological variables (e.g., temperature) at the precipitation site. Because of the difficulty of unambiguously determining the vapor source region(s), relatively fewer studies consider evaporation and transport conditions. Increasing accessibility of Lagrangian air parcel tracking programs now allows for an integrated look at the relationship between the precipitation isotope ratios and the evolution of moist air masses. In this study, 70 precipitation events occurring…'

Removed: We expect isotopes to respond similarly for longer-term climate-induced changes to the mean position of meridional circulation features, and expect that the most of the variation in isotopes measured in ice cores and other long term records are driven by changes in circulation, instead of fluctuations in local temperature.

P.1, L.1 to 5: The first three sentences of the abstract could rather be at the beginning of the introduction, as they don't describe the work presented in this article but general situation of research in the domain.

*The content of these sentences is covered in the introduction, so they were removed from the abstract, which helps to reduce the length of the abstract. They were replaced with the following:*

P.1 L. 1-2
"In this study, precipitation isotopic variations are linked to conditions at the moisture source region, along the transport path, and at the site of precipitation. Seventy precipitation events…"

P.1, L.8: "occurred" > "occuring"
*This was changed in the text.*
P.1 L. 5

Methods:
P.3, L.13-14: There might be an effect of sublimation of snow which could influence the isotopic composition of water, particularly for sunny periods, even within 24 hours. Did you make some experiments to test the evolution of fresh snow on your sampling site?
*The reviewer is correct that this is possible. However, for much of the season when Barrow receives snow, there is little sun to drive sublimation, and the Arctic tends to be quite cloudy. There is also little evidence of sublimation from the data distribution along the meteoric water line. Nonetheless, clarification was added in the referenced section.*

P.3 L.13-16
'…and often as soon as snow ended. Though it is possible that snow may have been altered by sublimation before collection, we assume that the degree of alteration of surface snow was minimal relative to the amount of snow gathered. Furthermore, the frequent cloudiness and darkness of Barrow mean that for most events, sunlight-driven sublimation was insignificant.'

P.3, L.14: At which temperature were the samples stored, and how long?
*Clarification was added to the text. Samples were stored at less than 5 C, shipped every 3 months, and analyzed within 6 months.*

P.3 L.16-19
'Liquid samples were stored in tightly sealed 30mL Nalgene bottles below 5 °C and shipped in batches every three months to the Stable Isotope Laboratory at Dartmouth College. When not in transit, samples were refrigerated. Samples were analyzed within six months of collection.'

P.4, L.1-4: Considering that a moisture source is corresponding to an air parcel sinking into the PBL is a strong assumption. More justifications of this method would be expected. If this method is described in Putman et al. (2015), add a reference here.
*The assumption has been more clearly stated and justified in the text. Also see our response, at the beginning, to "Specific comments: Modeling"*

P.4, L. 7-14
'Relative to previous studies that tracked vapor change in an air parcel along the trajectory (e.g., Sodemann et al. (2008a)), we adopted a simpler procedure that assumes vapor in the air parcel is well represented by the air at the latest interaction with the PBL. This assumption is justified because mass movement in the PBL is dominated by vertical turbulence relative to horizontal advection. Figure 1 shows endpoints of all trajectories that sank into the PBL. However, only trajectories that ended over water with < 96% sea ice cover were used for calculations; parcels that sank where there was less than 96% sea ice cover were used for calculations.'

P.4, L11-12: By "the most temporally homogeneous three-hour time window", do you mean homogeneity in the precipitation amount or in the meteorological records? Do you have particular criteria to define the preference for the middle of the event? Were the event times defined automatically or manually?
*This section has been clarified. Homogeneity is in reference to the radar returns, which give us an idea of precipitation intensity. Event times were selected manually, based on multiple streams of evidence: radar returns, sampling records and surface analysis maps.*

P.4 L.15-23
'Back trajectory analysis was performed for dates when precipitation was collected. The starting times for the back trajectories corresponded to times of maximum precipitation intensity, based on a combination of sampling records, surface analysis maps of Alaska available through the National Center for environmental Prediction, and the returns of the millimeter wavelength cloud radar (MMCR) (Johnson and Jensen, 1996; Bharadwaj et al., 2011). Greater Doppler vertical velocities, reflectivities, and spectral widths from the MMCR broadly indicated more intense precipitation. Because the gridded meteorological files 20 used for tracing the back trajectories had three-hour resolution, the chosen starting time represented average conditions over a three-hour period. If precipitation lasted for more than three hours, the most intense three hour time window was selected. If the precipitation was of approximately uniform intensity, the most temporally homogeneous three-hour time window was selected, with preference for time windows where precipitation occurred over the duration of the three hours.'

P5., L.5: "The same was done for an array...": explicit that this is to calculate Qsat,z and define Qsat,z.
*This section was rewritten for clarity, and to reflect the change in the equations used.*

P.5 L. 14-25
'To determine $Q_{sat;z}$, we start from the dry adiabatic lapse rate, (-9.8 °C km-1). From this we determine the temperature $T_z$ at altitude z, starting with the 2 meter temperature $T_{2m}$.
The saturation vapor pressure at elevation z, $e_{sat;z}$ is then

$$e_{sat;z} = 0.6113 \exp[5423 (1/T_0 - T/T_z)] \quad (2)$$

where $T_0 = 273.15K$ (Stull, 2015). We may then write the saturation specific humidity, $Q_{sat;z}$, as

$$Q_z = 0.622 \, e_{sat;z} \, h_z \, /P_z \quad (3)$$

where $h_z$, the relative humidity at height z, is assumed to equal 1 (air is vapor-saturated) and the pressure at height z, $P_z$, is

$$P_z = 1013.25 [1-(2:25577 \, e\text{-}5) \, z]^{5.25588} \quad (4)$$

Calculating the 2m specific humidity, $Q_{2m}$, is simply a special case of the general calculation: we use the 2m temperature $T_{2m}$, fractional relative humidity $h_{2m}$, and pressure $P_{2m}$ from reanalysis in Equations 2 and 3, rather than using the dry adiabatic lapse rate, h = 1, and Equation 4, respectively.

Finally, we find the elevation where $Q_{2m}$ equals $Q_{sat;z}$. The temperature at this elevation is $T_{LCL}$.'

P.5, L.5-6: Explicit hz, Tz, Pz: fractional relative humidity, temperature and pressure at elevation z.
*This suggestion was incorporated into the newly written section discussed in the previous point.*

P.5 L. 14-25

P.5, L.25: Are mtn values assigned manually or automatically? If automatic, then explicit the criteria.
*Mtn was assigned manually based on maps of trajectory results.*

P.6 L. 18-19
'The value of mtn was assigned manually based on the general pattern of transport observed in the trajectory plots.'

Results and discussion
P. 6, L.6-15: This is a very qualitative description of Figure 1. The mean latitude of moisture sources could be introduced before and used to give quantitative aspects to this description. This description focuses on the seasonal averages of the moisture sources, but Figure 2 shows a very strong variability at the event time scale, which can be of a larger order of magnitude than the variations of the seasonal average for the mean latitude of the moisture source. For example, some events in winter have moisture sources located as north as in summer, or even further north. The normalisation of the maps from Figure 1 can also give an impression of wider or more local moisture sources depending on the total number of events and the difference between each event. Is this description of

moisture sources regions still valid for absolute values without normalization to the number of events, or for individual events instead of the average of all events?

*The authors agree that there is substantial variation among events, even within a given season. This follows the response to 'interpretation of results', above.*

P. 7 L. 34:
'… though the inter-event variability in both variables can be as large as the seasonal variability'

Caption of Fig 3
'Of the three timescales, annual variability shows the greatest amplitude, though variability among events is also substantial.'

Caption of Fig 4
'The spline fits have $R_2$ values of 0.60 and 0.19 for the $d_2H$ and $V_{Lat}$ respectively. For both datasets, the variability exhibited among events is of the same the order of magnitude as the seasonal variability.'

P. 6, L. 30-32: Not clear if the last sentence refers to Feng et al. (2007).
*Modified sentence.*

P.7 L.19
'There is evidence for prior millenium-scale shifts in the southern extent of the polar circulation cell (Feng et al., 2007).'

P. 7, L. 6-7: This sentence is really affirmative, whereas Figure 4 shows a very strong dispersion, particularly for the averaged VLAT. This affirmation should be tempered and a statistical evaluation of the spline fits and there correlations should be given, as well as the standard deviations of the data series. The seasonal scale might not be the better scale to look at.

*This is true, in particular for the $V_{lat}$ variable. The variance captured by the spline fits has been added to the text, and the text has been adjusted to better describe the similarity in magnitude between among-event variability and mean seasonal variability.*

P. 7 L. 30-34
'Figure 3 also shows the interannual, seasonal and event-scale variability captured by the dataset where the spline captures 65% of the annual and interannual variance. The average annual cycle of the precipitation $\delta_2H$ is strong; the spline fit explains 60% of variance in the data. The mean latitude of the vapor source exhibits a weak seasonal pattern, where the spline explains 19% of the variance. The seasonal cycles of $\delta_2H$ and vapor source latitude are in phase, as shown in Figure 4,though the inter-event variability in both variables can be as large as the seasonal variability'

Caption of Fig 3
'The spline fit, which highlights seasonal variations, explains 65% of variance in the data with a root mean squared error of 39.7‰. Of the three timescales, annual variability shows the greatest amplitude, though variability among events is also substantial.'

Caption of Fig 4
'The spline fits have $R_2$ values of 0.60 and 0.19 for the $d_2H$ and $V_{Lat}$ respectively. For both datasets, the variability exhibited among events is of the same the order of magnitude as the seasonal variability.'

P. 8, L.3: How did you choose the temperatures from 10C to -15C in you theoretical cooling experiments? What would be the effect on the slopes of a variation of these temperatures on the order of magnitude of the observed variations?

*The temperature range encompasses the temperature change experienced by most trajectories. Making the warmest temp even warmer would yield slightly shallower slopes, and making colder temperatures even colder would yield steeper slopes. The coldest average final temperatures in our dataset are substantially below -15C, though the majority of each trajectory occurs within the 10 to -15C temperature range, and all events except one begin in the selected range.*

P. 8, L. 18: Rather write "more than 20C" instead of "> 20C".
*Sentence was deleted when discussion was updated.*

P. 9, L. 1: "amount" instead of "amounts"?
*Sentence was deleted when discussion was updated.*

P.9, L.5: How was the 7C criteria chosen? Is it close to the median of the distribution of ΔTcool?
*The 7C criterion was chosen to preserve the statistical power of the short trajectories while preserving the strong relationship of δ2H to Td. Though we have presented the results as categorical, this is for simplicity. It is likely that this feature of isotope systematics is actually continuous.*

P.10 L.22-24
'The breakpoint of 7 °C was chosen by testing different breakpoints and finding one that maximized the statistical power of the short trajectory regression while preserving the strong relationship between δ2H and Td.'

P. 9, L.12: Insert a reference to figure 6 to show the repartition of small and large ΔTcool across seasons.
*Added sentence:*

P.10 L.21-22
'Table 2 summarizes the results and Figure 6 shows the standard deviation of $\sigma T_d$ by category'

P. 10, L.6: This is not directly about precipitation d-excess but can be of interest: some studies of water vapour d-excess in Arctic regions have depicted a partial conservation of the source d-excess signal under certain atmospheric transport conditions, with relations between observed d-excess and moisture source relative humidity.

Bonne, J.-L., Masson-Delmotte, V., Cattani, O., Delmotte, M., Risi, C., Sodemann, H., and Steen-Larsen, H. C.: The isotopic composition of water vapour and precipitation in Ivittuut, southern Greenland, Atmos. Chem. Phys., 14, 4419-4439, doi:10.5194/acp- 14-4419-2014, 2014.

Bonne, J.-L., et al. (2015), The summer 2012 Greenland heat wave: In situ and remote sensing observations of water vapor isotopic composition during an atmospheric river event, J. Geophys. Res. Atmos., 120, 2970–2989, doi:10.1002/2014JD022602.

Steen-Larsen, H. C., A. E. Sveinbjörnsdottir, Th. Jonsson, F. Ritter, J.-L. Bonne, V. Masson-Delmotte, H. Sodemann, T. Blunier, D. Dahl-Jensen, and B. M. Vinther (2015), Moisture sources and synoptic to seasonal variability of North Atlantic water vapor isotopic composition, J. Geophys. Res. Atmos., 120, 5757–5774, doi:10.1002/ 2015JD023234.
*Interesting work, thank you for the citations. The discussion of d-excess has been updated to include these publications.*

P.11 L.31-32
'While studies indicate that d in vapor contains vapor source information (Steen-Larsen et al., 2014; Bonne et al., 2015; Steen-Larsen et al., 2015)…'

P.12 L. 9-11
'This value is consistent with the -0.4 to -0.6‰ %-1 range reported in the literature for vapor (Uemura et al., 2008; Pfahl and Wernli; Bonne et al.; 2014).'

Conclusions
P. 10, L. 29-31: This conclusion on the origins of moisture is valid for the average of the seasonal moisture sources, but should be tempered by pointing out the event to event variation of the moisture sources.
*This caveat was added to the conclusions.*

P.12 L.33

*'However, substantial intra-season variability occurred in both source and δ2H, indicating scatter in the seasonal relationship.'*

References
P.13, L. 32-36: Logically, the two papers numbering should be inverted (2008a and 2008b).
*Changed.*

Tables and figures
Table 1 and 2: The legends do not clearly describe the contents of the tables. Why are different intercepts given for each variable in Table 2 and only one value in Table 1, if the only difference between the two tables are the division of all samples in two groups?
*The captions of the tables have been updated to explain that Table 1 contains the results from a single multivariable regression, while Table 2 contains the results from 3 simple linear regressions.*

Caption of Tab 1
*'Variation in δ2H is explained by a multiple linear regression ($R^2$ = 0.54) of air parcel cooling during transport (ΔTcool), moisture source conditions (Td) and orographic obstacles in vapor transport path (mtn).'*

Caption of Tab 2
*'Three simple linear regressions against δ2H where β is the regression coefficient and S.E. is the standard error'*

Figure 7: Parenthesis not closed in right y-axis label.
*Fixed.*

Figure 3 and 7: It would be more readable with x-axis ticks corresponding to the beginning of the years instead of the beginning of each December.
*Fixed.*

**Review of**
**"Annual variation in precipitation δ²H reflects vapour source region at Barrow, AK"**
by A. L. Putman et al.
Paper published in ACPD on 11 August 2016

**1 General Comments**

This paper presents an interesting dataset of the event-scale $\delta_2$H and deuterium excess signature of precipitation from northern Alaska. The authors use a very simple back-trajectory-based analysis of the transport and moisture source conditions which they summarise in 3 main characteristics to interpret their data. These are 1) the moisture source dew point temperature at 2m, 2) the total cooling between the lifted condensation level at the moisture source and the precipitation level in the cloud at the measurement site (arrival temperature) and 3) whether the air parcels that are transported to the measurement site across the Brooks and/or the Alaskan ranges. I recommend publication of this overall well-written manuscript, but I have four major concerns that should be addressed beforehand as well as a many specific comments listed below:

*The authors thank the reviewer for the useful points and ideas. We have considered the suggestions, addressed the questions and revised the paper accordingly, and we hope that the revisions are satisfactory. Line numbers provided refer to the final manuscript.*

1 **Moisture source identification and particularly the implicit assumptions made:**
see specific comments 3-7.

*The authors argue that the method employed in this paper is adequate for our purpose. Please see our responses for comments 3-7 for the full discussion.*

2 **Choice of the parameters that explain the variance of the isotope signature of precipitation in Barrow:**
For me the choice of the parameters that were used to explain the precipitation isotope signal in
Barrow seems random. It makes sense to look at moisture source and transport conditions but in my opinion there is no reason for completely neglecting the local conditions. Particularly at Barrow, the precipitation phase (liquid or snow) probably plays an important role for the end isotope composition of the precipitation event as it determines whether there is isotopic exchange (for rain
drops, see specific comment 2) or not (for snowfall) with the local vapour. Also precipitation intensity plays an important role. The authors have some detailed information about the precipitation structure from their radar data and could use this to try to further understand the local processes.
If this is done in an other paper, then this should be clearly stated. Also I do not fully support the choice of the variable $T_d$ as representative for the moisture source conditions (see specific comment 12).

*This is a good point, and one that was considered by the authors before settling on the variables reported. Indeed, half the variance in δ2H cannot be explained by the 3 variables chosen! The main reason that other variables (including but not limited to precipitation phase, sub-cloud dryness, precipitation intensity, evaporation below the cloud base, supersaturation in the cloud, and storm event type) were not included is because, the statistical power of the limited number of events we were able to consider is not sufficiently high to go after each of those potentially very important variables. When such variables were included in the analysis, they did not explain any more variance. This may be because the isotopic responses to them are not related to δ2H variations, or are related but not sufficiently above noise. For example, sub-cloud dryness may be important for some but not all events, dryness may occur during both high and low δ2H events, but the power or the size of the signal may be limited. Nevertheless, to respond to this point, we added a paragraph at the end of Section 3.2 that includes a list of variables potentially contributing to the 46% of the unexplained variance in δ2H.*

P. 11 L. 13-20
'The three chosen variables explain just over half (54%) the variance of δ2H. This is not surprising, considering that many other mechanisms can also influence the δ2H of the vapor and precipitation. These mechanisms include (but are not limited to) condensation temperature, supersaturation in the mixed phase cloud, sub-cloud dryness, phase of precipitation, precipitation intensity, evapotranspiration of land sources, and the amount of sea ice at the vapor

source. The effects of several of these factors, including condensation temperature, sub-cloud dryness, sea ice concentration at the vapor source, and phase of precipitation (rain vs. snow), were tested as additional explanatory variables in the multiple regression, but yielded statistically insignificant results with little to no additional variance explained. Clearly, compared with the three chosen variables, the effects of these variables are relatively minor, such that the statistical power is not sufficient to reveal their significance.'

3 **Expansion of the northern polar circulation cell and its link to moisture source location**
The link between the event-based moisture source location of precipitation and the polar circulation cell is described in a very qualitative way. A link between the weather systems driving the moisture transport at the event timescale leading to precipitation at Barrow and the more climatological description of the polar circulation is not obvious and not trivial to make. The formulations used throughout the paper should be more careful and kept as hypotheses.
*The authors acknowledge that the relationship between vapor source and circulation is not simple, and the link between the annual and longer timescales is a possibility, not a certainty. However, the work does substantiate the idea that isotope values measured in ice cores may reflect changes in circulation patterns as well as local temperature, which is how they are often interpreted. The phrasing of these statements has been re-formulated in all discussion to suggest hypothetical as opposed to likely links.*

P. 7 L. 20-25
'Aspects of the link between seasonal variability in general circulation and seasonal vapor source cycling may be generalizable to interannual and even millennial timescales. This is relevant to modern changes in the hydrologic cycle as Marvel and Bonfils (2013) suggest that a poleward displacement of circulation cells is already occurring due to recent climate change. Additionally, changes in the isotopic composition of precipitation resulting from systematic vapor source migrations associated with changing climate may allow for interpretation of long-term isotopic records in terms of changes in atmospheric circulation, including but not limited to the precipitation site temperature.'

P. 13 L. 14-16
'The mechanisms identified, most notably the north-south migration of the vapor source region in phase with expansion and contraction of the Polar circulation cell, may also operate on times scales longer than that of our study, and may be a source of variation in isotopes measured in ice cores, pedogenic carbonates, and speleothems.'

4 **Critical discussion of results in view of the existing literature**:
in particular see specific comments 24 and 27.
*More discussion has been added to Sections 3.2 and 3.3. In 3.2, which discusses the influence of vapor source on measured precipitation isotopes, greater clarification of the simple Rayleigh model used to contextualize our results has been added, as well as a comprehensive sources of error paragraph, and an expanded discussion of the utility of $T_d$ in characterizing the source. In section 3.3, the d-excess results are discussed in greater depth in light of the suggested papers. In particular, we have added discussion of the relationship between local water vapor and evaporation conditions. The new or revised section are in order as listed below.*

P. 8 L. 25-29

[revised manuscript text omitted]

**2 Specific comments**

1. p. 1, title: It would be nice to include in the title the fact that it is event-scale precipitation samples that the authors analyse in this paper. Something like: "Annual variation in event-scale precipitation $\delta 2H$ reflects vapour source region at Barrow, AK". Also Barrow, AK could be replaced by northern Alaska.
*Title changed as suggested.*

New title:
'Annual variation in event-scale precipitation $\delta 2H$ at Barrow, AK reflects vapor source region'

2. p. 18-23: The local conditions during cloud formation and during precipitation also play an important role for the isotope composition of precipitation. For rainfall for example below cloud effects (evaporation and exchange with ambient vapour) can have a strong impact on the isotope composition of precipitation (20-40h for $\delta 2H$, see Pfahl et al. (2012), Aemisegger et al. (2015)).
*This is absolutely true, and we did experiment with including condensation temperature, precipitation type, and sub-cloud humidity in our regressions. The regression presented was the best model in terms of simplicity and variance explained by different parameters. One reason why these local factors may not have been significant influences to our dataset is because of event-to-event variability. e.g., in one case enrichment may be due to sub-cloud evaporation, but in another it may be due to condensation temperature, and within our dataset we did not have the statistical power to disentangle these competing mechanisms. Also see our response to General Comments 2).*

P. 11 L. 13-20
'The three chosen variables explain just over half (54%) the variance of $\delta 2H$. This is not surprising, considering that many other mechanisms can also influence the $\delta 2H$ of the vapor and precipitation. These mechanisms include (but are not limited to) condensation temperature, supersaturation in the mixed phase cloud, sub-cloud dryness, phase of precipitation, precipitation intensity, evapotranspiration of land sources, and the amount of sea ice at the vapor source. The effects of several of these factors, including condensation temperature, sub-cloud dryness, sea ice concentration at the vapor source, and phase of precipitation (rain vs. snow), were tested as additional explanatory variables in the multiple regression, but yielded statistically insignificant results with little to no additional variance explained. Clearly, compared with the three chosen variables, the effects of these variables are relatively minor, such that the statistical power is not sufficient to reveal their significance.'

3. p. 3, L. 29: The reanalysis dataset (wind fields) that is used for the trajectory calculation should be mentioned here as well as its horizontal resolution.
*The information has been added.*

P. 4 L. 3
'…using 1° resolution meteorological data from the Global Data Assimilation System (GDAS).'

4. p. 4, L. 2: What do the authors mean with "The first time"? Is the time reference forward or backward? Does that mean the first time when following the trajectory back from the arrival point? And does that mean that one trajectory can have only 1 associated moisture source? This would be a very strong assumption about the moisture source location. Uptakes of moisture can happen all along an air parcel's trajectory (see Sodemann et al. (2008)) and they can sometimes be linked to surface evaporation even though they are not in the boundary layer (PBL), particularly over land. If for each trajectory only the latest passage in the PBL before arrival at the measurement site is considered then this means that the authors assume very strong mixing. This would imply that the air parcel basically looses all its previous humidity by mixing out and takes up only humidity that has just been evaporated at

this location. The isotope signature of the air parcel thus is fully determined by the freshly evaporated water. This strong assumption has to be explicitly stated.

*The 'first time' is in reference to back trajectories; wording in the manuscript has been updated for clarity. Yes, each trajectory has one associated vapor source. Though the method described in Sodemann (2008) is a substantially more sophisticated way of identifying the vapor source, it is not necessary in our work for three reasons. 1) For a given parcel, the spatial range over which the parcel moves up and down across the PBL, is small compared to the region covered by 1000 total parcels of an event. The latter is primarily dictated by the vertical distribution of the initial parcels' altitude 2) Our work analyzes the influence of marine source areas on precipitation at Barrow, AK. Marine surface conditions are relatively homogeneous, which point strengthens the argument in 1). 3) Averaging at the precipitation site of condensate of 1000 trajectories from a wide spatial distribution of source locations implicitly accounts for mixing of moisture from distributed source locations. It is thus in effect equivalent to the more sophisticated Sodemann et al. (2008) model which used on average only 2.6 trajectories per column of air per time window, even if each of those trajectories combined source points at its inception.*

P. 4 L. 6-10
'The vapor source location was defined as the place where the back trajectory of the air parcel sank into the planetary boundary layer (PBL). Relative to previous studies that tracked vapor change in an air parcel along the trajectory (e.g., Sodemann et al. (2008a)), we adopted a simpler procedure that assumes vapor in the air parcel is well represented by the air at the latest interaction with the PBL. This assumption is justified because mass movement in the PBL is dominated by vertical turbulence relative to horizontal advection.'

5. p. 4, L. 4: The authors say that air parcels that sank below the PBL over land were ignored? Why then do they find a lot of moisture sources over continental Alaska in Figure 1? This is confusing.
*The difference between data used for statistics and data used in figure is confusing, and text has been added to clarify the distinction.*

P. 4 L. 11-14
'Figure 1 shows endpoints of all trajectories that sank into the PBL. However, only trajectories that ended over water with < 96% sea ice cover were used for calculations; parcels that sank where there was less than 96% sea ice cover were used for calculations. Parcels that never sank into the PBL or those that sank into the PBL over land or ice-covered ocean were ignored. Ocean-originating air parcels comprised about 71% of all trajectories.'

Fig 1, caption
'The figure indicates that some air parcels originate over land, but these were not included in calculations.'

6. p. 4, L. 4: Were 71% of all trajectories ignored or kept for the analysis?
*Changed for clarity.*

P. 4 L. 14
'Ocean-originating air parcels comprised about 71% of all trajectories.'

7. p. 4, L. 5-12: For me it is not entirely clear how the trajectory starting dates were chosen. Why do the authors choose only a three hours period instead of the whole precipitation event? Why are the individual dates not weighted by the locally measured precipitation intensity to take into account that when the precipitation intensity is higher the trajectories of that date contribute more to the isotope signal? What means the "most homogeneous" three-hour time window? And why with preference to the "middle" of the event? The selection criteria should be more oriented to the quantitative contribution of moisture to precipitation in my opinion.

*The three hour time period was chosen so that each event was treated the same. During analysis, we selected a few events to test the sensitivity of the vapor source to the selected time. In these cases, multiple 3 hour windows were analyzed for a single event, and typically showed very similar results. Thus, we concluded that choosing one three hour window was representative of the whole event. Selection of the specific three hour time period was not automated or quantitative. The qualitative selection used returns from the MMCR and KAZR. Higher Doppler vertical velocity and reflectivity indicate increased precipitation intensity. These criteria, in conjunction with information from surface analysis maps were used to determine the start and end times. The three hour window*

*reflects the constraints of the reanalysis, i.e. the temporal resolution is three hours. The text has been clarified with respect to these questions.*

P. 4 L. 15-23
'Back trajectory analysis was performed for dates when precipitation was collected. The starting times for the back trajectories corresponded to times of maximum precipitation intensity, based on a combination of sampling records, surface analysis maps of Alaska available through the National Center for Environmental Prediction, and the returns of the millimeter wavelength cloud radar (MMCR) (Johnson and Jensen, 1996; Bharadwaj et al., 2011). Greater Doppler vertical velocities, reflectivites, and spectral widths from the MMCR broadly indicated more intense precipitation. Because the gridded meteorological files used for tracing the back trajectories had three-hour resolution, the chosen starting time represented average conditions over a three-hour period. If precipitation lasted for more than three hours, the most intense three hour time window was selected. If the precipitation was of approximately uniform intensity, the most temporally homogeneous three-hour time window was selected, with preference for time windows where precipitation occurred over the duration of the three hours.'

8. p. 4, L. 13: Does "where" mean the starting altitude? The method that is shortly described in this paragraph sounds original and the idea is interesting but it assumes that the reanalysis dataset's wind field and precipitation rate profile are equivalent with the true fields. The reanalysis data error particularly with respect to the representation of small and microscale processes are ignored. Starting trajectories from different locations around the measurement site would allow to take into account the uncertainty arising from the reanalysis data
*Yes, 'where' means altitude. The authors agree that accurate wind directions are very important for accurate back trajectories. However, if winds in reanalysis are incorrect, incorporating a wider area will not make them more correct. Furthermore, because the resolution of the reanalysis data is 1x1 degree, using multiple locations may cover a wide region, hundreds of kilometers in size. Such a wide spatial scale could be less representative of local or small-region precipitation events, though it could be helpful for large-region precipitation events. We are not convinced that looking over a larger spatial location would improve the vapor source estimation. However, sending a large number of air parcels (~1000), as we have done, helps to deal with the wind issue. Finally, wind errors in the reanalysis are a potential source of error for any Lagrangian back trajectory study, and are not unique to our study. Therefore, we hope people who are reading studies using Lagrangian back trajectories are in general cautious with this type of reanalysis product.*

P. 4 L. 24-25
'The method for selecting the altitudes where the air parcels began their back trajectories is described in full in Putman (2013).'

9. p. 4, L. 25: How did the authors calculate $T_d$ and are the average moisture source conditions computed as an arithmetic mean without taking into account the evaporative contribution to the air parcel's humidity at the different source locations?
*The description of the calculation of Td is later in Methods section of the manuscript (Section 2.3.2). Because of the way the condensation profile is divided, it is assumed that each parcel contributes an equal amount of vapor to the final precipitating cloud, so we did not weight them.*

P. 6 L. 6-9
'We approximate $T_d$ using

$$T_d = [1/T_0 - 1.844*10^{-4} \ln(e_{sat;2m} h_{2m} 0.6113)]^{-1} \quad (5)$$

(Stull, 2015) with saturation vapor pressure, $e_{sat;2m}$, from Equation 2 and the 2m air temperature, $T_{2m}$, and relative humidity $h_{2m}$ from reanalysis data.'

P. 4 L. 27-29
'The precipitation rate profile was differentiated with respect to height, yielding the condensation rate profile (g m-3 s-1) and then subdivided into the aforementioned 1000 air parcels so as to ensure that each parcel contained an equal fraction of total precipitation.'

10. p. 4, L. 30: The authors should make clear that their $T_{cool}$ is only an estimate of the total cooling that the air parcel has experienced. The same remark for the possibility of multiple moisture sources for one air parcel (see specific comment 4) is valid for cooling and precipitation along an air parcel trajectory. A trajectory can produce rain all along its path and can go through several cycles of cooling and warming. The total cooling would be obtained by integrating the temperature changes along the trajectory.

*The reviewer is correct, the cooling indicated by ΔTcool is the net cooling, not the integral of cycles of warming and cooling an air parcel may have experienced along its trajectory. This is a simplification. This has been made explicit by additions and revisions to the text.*

P. 5 L 8-10
'An estimate of air parcel cooling that produced condensation, ΔTcool, is a bulk metric quantifying the magnitude of Rayleigh distillation along the trajectory (Sodemann et al., 2008a). This approach simplifies the integration of cycles of warming and cooling that may occur along a trajectory to a net reduction in temperature.'

11. p. 5, L. 3-9: this way of computing $T_{LCL}$ is confusing for me. Where does Eq. 2 come from? See Bolton (1980) and Lawrence (2005).

*The equations have been changed to those in Stull (2015) and the description of the calculation has been updated. The previous equation was a linearized approximation that introduced minor, if not insignificant, discrepancy. To be more precise, the equations, calculations, and text have been updated to be consistent with Stull (2015), though the results and discussion require no change. Equations 2, 3, and 5 are affected.*

P.5 L. 14-25
'To determine Qsat;z, we start from the dry adiabatic lapse rate, (-9.8 °C km-1). From this we determine the temperature Tz at altitude z, starting with the 2 meter temperature T2m.
The saturation vapor pressure at elevation z, esat;z is then

$$e_{sat;z} = 0.6113 \exp[5423 \, (1/T_0 - T/T_z)] \quad (2)$$

where T0 = 273.15K (Stull, 2015). We may then write the saturation specific humidity, Qsat;z, as

$$Q_z = 0.622 \, e_{sat;z} \, h_z / P_z \quad (3)$$

where hz, the relative humidity at height z, is assumed to equal 1 (air is vapor-saturated) and the pressure at height z, Pz, is

$$P_z = 1013.25 \, [1-(2{:}25577 \, e\text{-}5) \, z]^{5.25588} \quad (4)$$

Calculating the 2m specific humidity, Q2m, is simply a special case of the general calculation: we use the 2m temperature T2m, fractional relative humidity h2m, and pressure P2m from reanalysis in Equations 2 and 3, rather than using the dry adiabatic lapse rate, h = 1, and Equation 4, respectively.

Finally, we find the elevation where Q2m equals Qsat;z. The temperature at this elevation is $T_{LCL}$.'

12. p. 5, L. 12-16: The idea to use $T_d$ as a summary variable for both relative humidity with respect to sea surface temperature ($h_{2m}$ SST) and SST-effects seems not justified to me from a physical point of view. The influence of SST on $T_d$ is only indirect and a strong coupling of the ocean surface conditions with near-surface air characteristics is not necessarily given particularly at the event timescale. From a theoretical perspective and for all isotope-enabled numerical modelling experiments it is the Craig-Gordon model and thus the other two variables that are used to determine $d$ of the fresh evaporate. So I am not convinced that it is sensible to introduce a third variable that does not contain more information than the specific humidity at 2 m. Furthermore, it should be made clear in the manuscript that it is not the 2m relative humidity that is important for the non-equilibrium fractionation part during surface evaporation but the humidity gradient towards the surface which is represented by the relative humidity at 2m with respect to sea surface temperature ($h_{2m}$ SST). The authors should make a stronger case for why they use $T_d$ rather than the classical variables. Also the sentence "$T_d$ depends on the specific humidity of saturated air at the sea surface and on the amount of dry air from aloft that has subsided and mixed into low altitude air" is a confusing statement.

*The use of Td and related discussion in this paper reflects that our group has done a substantial amount of work to model and understand isotopic variations in the marine boundary layer (manuscripts in preparation), and our understanding continues to improve. We realize that our discussion about Td in the earlier version was not clear, and it is valid for this reviewer to solicit further explanation. For clarity for the reader, we have completely rewritten section 3.2 that pertains to Td, and we hope the new discussion in the revised version is clearer.*

*In short, the idea of using Td is to indicate the moisture conditions within the PBL, as this is the moisture that forms the first condensate. This is different from the evaporative flux predicted by the Craig-Gordon model. In addition, the Craig-Gordon model does not consider effects of convection on vapor isotopic ratios in the PBL. However, convection is an important process that 1) transports PBL air to the free troposphere, and 2) brings dry air from aloft to the PBL. The boundary layer air is therefore a mixture of evaporated vapor from the ocean surface, and the dry air from aloft. The extent of this mixing within the PBL is reflected by (2m) dew point, Td.*

*Td is also useful because it is directly related to relative humidity with respect to the sea surface temperature (h2m, SST), moreso than is the 2 m relative humidity. Indeed, when h2m, SST was used in the multiple regression instead of Td, it was a significant predictor of δ2H. However, in both variance explained and AIC, the multiple regression that incorporated Td performed better. Both because it performs better in the multiple regression, and because it is a measurable quantity, we prefer Td to h2m,SST and have retained it in the paper.*

P. 9-10 L. 7-35, 1-9
'We prefer Td to the classical variables Tss and h for determining isotopic evaporative fluxes. This choice is based on our understanding that the meteorological variable Td characterizes the bulk vapor content and isotopic ratio of the marine PBL, independent of the vapor temperature. When advected to the free troposphere, it is this vapor that will form precipitation. Additionally, through equilibrium fractionation Td also determines the isotopic ratio of the first condensate at the LCL, where Rayleigh distillation begins.

Within the marine PBL, several inter-related factors/processes are at work to determine the starting point of a Rayleigh trajectory. The first is the isotopic flux of evaporation from the sea surface. Most studies estimate this flux using the classic model by Craig and Gordon (1965). In that model, three variables control the evaporative flux: the sea surface temperature, Tss, δ2H above the laminar layer, and the humidity hss above the laminar layer (e.g., at 2 m), defined relative to Tss. Though hss is not a measured quantity nor one that is normally modeled, it is determined by Td above the laminar layer and Tss. Hence isotopic fluxes can be determined with the classical model using Tss, and Td and _2H above the laminar layer as input variables. From a physical point of view, Tss determines the amount of equilibrium fractionation at the water-air interface. Td and vapor δ2H, as well as Tss, control kinetic fractionation as vapor diffuses across the laminar layer. It should be noted that when Tss is large, Td tends to be large as well, as a result of their change with latitude and season. Td and δ2H are also correlated, which will be discussed below. Therefore, all three variables controlling the evaporative flux, Tss, and hss and δ 2H above the laminar layer, are associated directly or indirectly with Td, making Td a good indicator of evaporation conditions.

The second process is convergence. At a moisture source location, low level air is moist due to evaporation near the sea surface. Convergence and uplift transports low-level moist air into the free troposphere where it mixes with dry, isotopically depleted air descending from surrounding regions resulting in strong humidity and temperature gradients near the sea surface (below 2 m). In contrast, the specific humidity and isotopic ratios in the bulk of the PBL above 2m are relatively constant, resulting from the relative contributions of vertical transport of moist low-level and descending air (Fan, 2016). Td and δ2H at 2m both reflect the outcome of this mixing process, and so it follows that they are positively correlated.

The third process is condensation at the LCL. The temperature of the air mass, which equals or is very slightly less than the local dew point, determines the amount of isotopic fractionation and thus the isotopic ratio of the first condensate. It is this isotopic composition that defines the beginning of the Rayleigh part of the trajectory. Only Td;2m, not Tss nor h2m, is directly associated with the condensation temperature at the LCL (which differs only slightly from Td;2m due to the pressure difference between 2 m and the LCL and its effect on saturation specific humidity).

Since all three processes before Rayleigh distillation are either directly or indirectly related to Td, we consider Td a better indicator for the source conditions than either Tss or h. It is difficult, however, to theoretically assess the

sensitivity of precipitation δ2H to variations in source Td, because this would require quantification of the theoretical relationship of Td to δ2H through each of the three processes and perhaps their combinations. We here report the first empirical sensitivity of 3.23‰ °C-1 (Table 1) for δ2H relative to Td. At the sea surface, for Tss between 0 and 25 °C, equilibrium fractionation as a function of temperature yields sensitivities between 1.1-1.6‰°C-1 (Majoube, 1971). However, a large part of this fractionation may be offset by condensation at the LCL. Consequently, the observed sensitivity probably reflects primarily the fraction of vapor contributed by dry, isotopically depleted descending air that converges within the PBL. Mixing with the dry air causes a decrease in Td, which affects the δ2H of the PBL in two ways: 1) making the PBL air dry and isotopically depleted, and 2) isotopically depleting the evaporative flux by enhancing kinetic fractionation (an effect of low relative humidity). Both mechanisms produce a positive association between δ2H and Td, consistent with the sign of our observed partial coefficient (Table 1).'

13. p. 5, L. 17: Where does Eq. 3 come from? What is the impact of the simplification involved, the authors should add a chapter reference to Stull (2015). Why did they not use Stull (2015), Equation 4.15a or b or extract directly $T_d$ from the reanalysis dataset?
*Calculation was updated to eqn. 4.15b in Stull (2015). See response to 11.*

14. p. 5, L. 23: How was *mtn* defined? Using an objective criterion or subjectively by looking at the trajectory plots?
*It was determined manually by examining the trajectory plots.*

P. 6 L. 17-18
'The value of mtn was assigned manually based on the general pattern of transport observed in the trajectory plots.'

15. p. 6, L. 2: remove parentheses.
*Parenthesis removed.*

P. 6 L. 20-22
'In this section we discuss the vapor source annual cycle and statistical relationships between the isotopic composition of precipitation, vapor source region, and the variables ΔTcool, Td, and mtn, that characterize the relationship of vapor source and transport to the isotope values measured at Barrow, AK.'

16. p. 6, L. 10-11: It would be useful to add the geographical names in one of the panels in Figure 1.
*The authors appreciate the suggestion, but have chosen not to incorporate it because more text would make the figure too busy and obscure the data presented in the plot.*

17. p. 6, L. 16-20: Is it really the variation in the moisture source latitude that is relevant or the mean transport distance? I am not convinced about the role of Figure 2. Also see major comment 3.
*Because most vapor transport is from mid-latitudes to high latitudes, latitude is actually relevant for this site. Yes, distance might be another reasonable metric to investigate. However, latitude was chosen because latitude covaries with evaporation conditions, so it's more physically useful than distance.*

*Figure 2 is useful as it shows the relationship of latitude to vapor source and distillation. A similar figure could have been made for distance, but the outcomes would be very similar, as in our case distance variation is roughly the same as that of latitude.*

18. p. 6, L. 21-32: For me this relatively long paragraph is a general discussion of the possible link between polar atmospheric circulation and the location of vapour sources and not a result from this study. Either the link with the findings in this paper should be illustrated more clearly or this section should be strongly shortened or even left out. See also my general comment 3: the link between the different timescales that are involved here is not trivial to make at this stage, a more open formulation should be chosen here.
*The link between seasonal changes to general circulation and seasonal change in vapor source makes sense because general circulation is the background pattern from which weather events deviate. This paragraph only links the seasonality of vapor source with the seasonality of circulation patterns- nothing over longer timescales. However, the language has been updated to be less causal.*

P. 7 L.20-25

'Aspects of the link between seasonal variability in general circulation and seasonal vapor source cycling may be generalizable to interannual and even millennial timescales. This is relevant to modern changes in the hydrologic cycle as Marvel and Bonfils (2013) suggest that a poleward displacement of circulation cells is already occurring due to recent climate change. Additionally, changes in the isotopic composition of precipitation resulting from systematic vapor source migrations associated with changing climate may allow for interpretation of long-term isotopic records in terms of changes in atmospheric circulation, including but not limited to the precipitation site temperature.'

19. p. 6, L. 24-26: In Europe several studies found that during summer the regional moisture recycling and the contribution from continental evaporation is much more important than in winter (see Sodemann and Zubler (2010) and Aemisegger et al. (2014)). Even though on p.4 L.3 the authors say that "only trajectories that sank into the PBL over the ocean" a substantial contribution of evaporation from continental Alaska is found in Spring but also in the other seasons in Figure 1. This possible contribution of continental evaporation should also be discussed as its moisture source isotope signature is different than the one from ocean evaporation.
*Local evapotranspiration during spring and summer is likely an important vapor source. However, given the heterogeneity of the event conditions and sources, we did not have the statistical power to pull evapotranspiration out relative to the other factors. Nonetheless, in the discussion, evapotranspiration is added as a potential mechanism of seasonal change as it likely does contribute to the δ2H measured in precipitation. Furthermore, it has been included in the sources of error discussion at the end of Section 3.2.*

P. 7 L. 11-17
'The migration of the mean latitude of the vapor source region can be tied to the seasonal cycling of solar insolation in the northern hemisphere via two mechanisms. Decreased solar insolation during winter drives expansion of the northern Polar circulation cell, which increases sea ice cover, and cold temperatures and snow cover prevent evapotranspiration. Both sea ice cover which diminishes the vapor contributions of the Arctic Ocean, and inhibited evapotranspiration allow for enhanced 15 representation of southerly vapor sources. Increased summer insolation drives poleward contraction of the circulation cell, diminishes sea ice coverage, and warmer temperatures favors evapotranspiration such that the average vapor source area migrates north.'

P. 11 L. 13-20
'The three chosen variables explain just over half (54%) the variance of δ2H. This is not surprising, considering that many other mechanisms can also influence the δ2H of the vapor and precipitation. These mechanisms include (but are not limited to) condensation temperature, supersaturation in the mixed phase cloud, sub-cloud dryness, phase of precipitation, precipitation intensity, evapotranspiration of land sources, and the amount of sea ice at the vapor source. The effects of several of these factors, including condensation temperature, sub-cloud dryness, sea ice concentration at the vapor source, and phase of precipitation (rain vs. snow), were tested as additional explanatory variables in the multiple regression, but yielded statistically insignificant results with little to no additional variance explained. Clearly, compared with the three chosen variables, the effects of these variables are relatively minor, such that the statistical power is not sufficient to reveal their significance.'

20. p. 6, L. 3: Add mid- to high latitudes here, other studies could be cited as well (e.g. Bonne et al. (2014))
*The phrase is changed, and citation added.*

P. 7 L. 29-30
'…that follows the well-established annual cycle for mid- and high latitudes (Feng et al., 2009; Bonne 30 et al., 2014)'

21. p. 7, L. 10-11: References to figures are confusing.
*The references to figures were removed.*

P. 8 L. 2-4
'…1) the temperature difference between vapor source region and precipitation site, quantified by air parcel cooling ΔTcool, 2) the moisture source conditions, quantified in this work by Td, and 3) the mean air parcel transport path.'

22. p. 7, L. 13: Do the authors mean the regression slopes? It would be useful to add the units of the slopes in all tables. Also in Table 3 it would be useful to add the explanation on what *β* and S.E. are.

*Changed the text to regression slopes. β and S.E. are described in the text that refers to Table 3. The units, originally just of the variable, are now the units of the slope in all tables. This is reflected in the column label.*

P. 8 L. 5-6
'Table 1 contains the partial regression slopes (β), p-values, and the unique variance explained by each variable.'
Caption of Tab 1.
'Values of β are the partial coefficients of the regression and S.E. is the standard error.'

Caption of Tab 2
'…where β is the regression coefficient and S.E. is the standard error'

Caption of Tab 3
'β is the regression coefficient and S.E. is the standard error.'

23. p. 7, L. 27: Here and elsewhere the references should be listed chronologically.
*Checked references throughout manuscript and reordered where necessary.*

24. p. 7, L. 34 - p. 8, L. 10: Here more detailed explanations on the theoretical cooling/Rayleigh experiment are needed to be able to follow. Also the sensitivity range of $\delta_2H$ to the diagnosed cooling should be put into context and compared to literature values.
*The model is explained in greater detail. The simple model is meant to contextualize the numbers.*

P.8 L. 27-29
'In such a model, a saturated air parcel with specified temperature and vapor δ2H is cooled iteratively in 1°C steps. At each temperature step, the condensation amount, remaining vapor, precipitation δ2H and vapor δ2H are calculated. No re-evaporation or non-equilibrium conditions are considered.'

25. p. 7, L. 21: Table 1: do the regression slopes from Table 1 result from multiple linear regression?
*Yes, this is stated in the text though it has been added to the Table caption now. The regressions in Tables 2 and 3 are now clearly indicated as simple linear regressions.*

P. 8 L. 4-6
'A linear combination of ΔTcool, Td, and mtn statistically represents the event-scale variation in δ2H with an R2 5 value of 0.54 (p < 0.001). Table 1 contains the partial regression slopes (β), pvalues, and the unique variance explained by each variable.'

P. 8 L. 24-25
'Our multiple regression yields…'

Caption Tab 1
'Variation in δ2H is explained by a multiple linear regression…'

Caption Tab 2
'Three simple linear regressions…'

Caption Tab 3
'Explaining deuterium excess (d) using simple regressions against…'

26. p. 9, L. 23: "within storm" is a confusing term here as it suggests that the precipitation is due to the passage of a cyclone, which is not always the case. I would suggest using "intra-event" instead.
*Changed to intra-event.*

P. 11 L.6
'…the distribution of intra-event…'

27. p. 10, L. 17: I am surprised at the *d–h* slope which is not at all in agreement (opposite sign and different order of magnitude) with other literature values (*d*-0.6h%−1 to -0.32h%−1, though a difference with literature values is that $h_{2m}$ is used and not $h_{2m \, SST}$). This mismatch should be explained and the relevant literature should be cited (Pfahl and Wernli, 2008; Steen-Larsen et al., 2014; Aemisegger et al., 2014). Also the *d*-SST regression slope is of opposite sign to what we would expect from the Craig-Gordon model.

*We too were also surprised at the outcome with respect to h2m and SST. This comment prompted us to calculate the statistics again with respect to h2m,SST. The results were significant (p < 0.001), and in the range of the values described above:- 0.39+/- 0.067 h%-1, with 34% variance explained. This has been added to the paper in this section as the primary result and contextualized by the values reported in the suggested citations.*

*The non-significant relationship to h2m and SST supports our prior argument (Section 3.2, point 12) that the isotope flux predicted by Craig-Gordon is only one process controlling the vapor properties in the PBL and does not indicate the bulk conditions within the PBL. Thus, a statistical non-relationship with variables that control isotopic flux is reasonable. The statistically significant results with h2m, SST and Td support our argument that the isotope value reflects the bulk conditions of the PBL, which are the result of a combination of the evaporative flux and mixing with drier air from aloft. This is discussed further in point 28, below.*

P. 12 L. 9-11
'…it is significantly predicted by $h_{ss}$ (p < 0.001, $R_2$ = 0.34), with a slope of -0.4‰%-1. This value is consistent with the -0.4 to -0.6‰ %-1 range reported in the literature for vapor (Uemura et al., 2008; Pfahl and Wernli; Bonne et al.,2014)'

P. 9-10 L. 7-35, 1-9 (see point 12)

28. p. 10, L. 20: What is the theoretical expectation for the sign of the correlation between *d* and $T_d$? This should be explained in more detail. I do not agree with the statement made here, I would expect a negative *d*-$T_d$ slope from theory since the physical relation between relative humidity and $T_d$ should generally lead to a positive correlation between the latter two (see e.g. Lawrence (2005)).

*Yes, we agree that we should expect a negative Td-d slope. The authors thank the reviewer for pointing this out. We now discuss this negative relationship in the context of two processes, 1) dry air (low Td) causing larger kinetic fractionation and higher d, and 2) the descending air may have high d values. Both processes, independently and together yield negative association between d and Td. The new information about the relationship to h2m, SST from point 27 supports our argument that Td and h2m, SST are related, and both are more representative of the PBL conditions than are Tss and h.*

P. 12 L. 15-20
'…with a negative slope (-0.53‰°C-1). This is an interesting result with respect to the utility of $T_d$, a measurable quantity, and is consistent with our earlier argument that $T_d$ is strongly related to $h_{ss}$. Both variables provide a better representation of source conditions than $T_{ss}$ and/or $h_{2m}$. A low value of $h_{ss}$ or $T_d$ corresponds to a strong influence of descending dry air within the PBL, which enhances kinetic isotopic fractionation and produces a high value of d. This mechanism explains the negative correlation between d and $T_d$, and is expected for the relationship between d and $h_{ss}$. Alternatively, the vapor in descending air may have a high value of d (Fan, 2016), or both mechanisms may contribute to this result.'

29. Figures 3 and 4: more details are needed on the used spline fits. Also the strong inter-event variability that is sometimes of similar amplitude as the seasonal cycle should be discussed.
*Details on the spline fits have been added to the figure caption, and the similarity in amplitude among seasonal and event variability is noted for both datasets in various pertinent locations in the paper.*

P. 7 L. 30-34
'…where the spline captures 65% of the annual and interannual variance. The average annual cycle of the precipitation δ2H is strong; the spline fit explains 60% of variance in the data. The mean latitude of the vapor source exhibits a weak seasonal pattern, where the spline explains 19% of the variance. The seasonal cycles of _2H and vapor source latitude are in phase, as shown in Figure 4, though the inter-event variability in both variables can be as large as the seasonal variability.'

P. 12 L.28-29
'…exhibited interannual, annual, and substantial inter-event variability.'

P.12 L. 33
'However, substantial intra-season variability occurred in both source and $\delta_2$H, indicating scatter in the seasonal relationship'

Caption Fig 3
'The spline fit, which highlights seasonal variations, explains 65% of variance in the data with a root mean squared error of 39.7‰. Of the three timescales, annual variability shows the greatest amplitude, though variability among events is also substantial.'

Caption Fig 4
'The spline fits have $R_2$ values of 0.60 and 0.19 for the $\delta_2$H and $V_{Lat}$ respectively. For both datasets, the variability exhibited among events is of the same the order of magnitude as the seasonal variability.'

30. Figure 5: the role of this Figure is unclear to me, it is only referenced once and not further discussed in the text. Either this Figure should be better embedded in the text or it should be left out. If it is kept: is this figure an average over all events?

*Yes, this figure shows the average value of δ2H of precipitation coming from a specific vapor source, which indicates that certain regions tend to be vapor sources for precipitation events that are either more or less enriched than average for that time of year. Mountains along the trajectory appear to be the mechanism at work in producing the spatial structure.*

P. 8 L. 14-16
'As demonstrated by Figure 5, the presence of mountains along the vapor transport path will deplete the isotope ratio of the precipitation relative to a uniform altitude transport, all other meteorological conditions being equivalent.'

*All of these suggested references have been checked and cited when appropriate.*

**List of all relevant changes in manuscript**
Line numbers refer to the final document, not the markup. Substantial blocks of text that have been removed are noted. Though the text included here may not be entirely new, all have significant additions, or have undergone substantial structural or content change in response to comments from reviewers.

**Title:**
Added 'event-scale'
Rearranged to 'precipitation δ2H at Barrow, AK reflects vapor source region'

**Abstract**
Removed 'Interpretation of variability in precipitation stable isotopic ratios often relies exclusively on empirical relationships to meteorological variables (e.g., temperature) at the precipitation site. Because of the difficulty of unambiguously determining the vapor source region(s), relatively fewer studies consider evaporation and transport conditions. Increasing accessibility of Lagrangian air parcel tracking programs now allows for an integrated look at the relationship between the precipitation isotope ratios and the evolution of moist air masses. In this study, 70 precipitation events occurring…'

**P. 1 L. 1-2**
"In this study, precipitation isotopic variations are linked to conditions at the moisture source region, along the transport path, and at the site of precipitation. Seventy precipitation events…"
**P. 1 L. 5**
'…occurring…'
Removed: We expect isotopes to respond similarly for longer-term climate-induced changes to the mean position of meridional circulation features, and expect that the most of the variation in isotopes measured in ice cores and other long term records are driven by changes in circulation, instead of fluctuations in local temperature.

[revised manuscript text omitted]

Since all three processes before Rayleigh distillation are either directly or indirectly related to Td, we consider Td a better indicator for the source conditions than either Tss or h. It is difficult, however, to theoretically assess the sensitivity of precipitation δ2H to variations in source Td, because this would require quantification of the theoretical relationship of Td to δ2H through each of the three processes and perhaps their combinations. We here report the first empirical sensitivity of 3.23‰ °C-1 (Table 1) for δ2H relative to Td. At the sea surface, for Tss between 0 and 25 °C, equilibrium fractionation as a function of temperature yields sensitivities between 1.1-1.6‰°C-1 (Majoube, 1971). However, a large part of this fractionation may be offset by condensation at the LCL. Consequently, the observed sensitivity probably reflects primarily the fraction of vapor contributed by dry, isotopically depleted descending air that converges within the PBL. Mixing with the dry air causes a decrease in Td, which affects the δ2H of the PBL in two ways: 1) making the PBL air dry and isotopically depleted, and 2) isotopically depleting the evaporative flux by enhancing kinetic fractionation (an effect of low relative humidity). Both mechanisms produce a positive association between δ2H and Td, consistent with the sign of our observed partial coefficient (Table 1).'

**Removed:**
'This is because source meteorological conditions control the δ2 H of the water evaporated from the ocean surface. Studies typically attribute variations in _2 H values at the source to mean sea surface temperature Tss and mean 2m relative humidity h2m, which affect the magnitudes of equilibrium and kinetic fractionation, respectively (Craig and Gordon, 1965). Assuming that Tss influences 2m air temperature, so that the two temperatures correlate spatially, we may use the 2m dew point (Td ) at the vapor source to combine the effects of Tss and h2m . Either high Tss or high h2m results in high Td . Therefore, we expect Td to be positively associated with δ2 H in the original vapor in an airparcel at the vapor source. This δ2 H signal at the source is then carried to the precipitation site. Differences in T2m cause 20°C> of the range we report for Td , whereas h2m contributes 2-4°C.The substantial difference between the vapor source Td for Arctic compared with subtropical sources makes Td a more useful metric for characterizing the vapor source than either Tss or h2m alone.'

**P.10 L.22-24**
'The breakpoint of 7 °C was chosen by testing different breakpoints and finding one that maximized the statistical power of the short trajectory regression while preserving the strong relationship between δ2H and Td.'

**P.10 L.21-22**
'Table 2 summarizes the results and Figure 6 shows the standard deviation of $\sigma$Td by category'

**P. 11 L.6**
'…the distribution of intra-event…'

**P. 11 L. 13-20**
'The three chosen variables explain just over half (54%) the variance of δ2H. This is not surprising, considering that many other mechanisms can also influence the δ2H of the vapor and precipitation. These mechanisms include (but are not limited to) condensation temperature, supersaturation in the mixed phase cloud, sub-cloud dryness, phase of precipitation, precipitation intensity, evapotranspiration of land sources, and the amount of sea ice at the vapor source. The effects of several of these factors, including condensation temperature, sub-cloud dryness, sea ice concentration at the vapor source, and phase of precipitation (rain vs. snow), were tested as additional explanatory variables in the multiple regression, but yielded statistically insignificant results with little to no additional variance explained. Clearly, compared with the three chosen variables, the effects of these variables are relatively minor, such that the statistical power is not sufficient to reveal their significance.'

**P.11 L.31-32**
'While studies indicate that d in vapor contains vapor source information (Steen-Larsen et al., 2014; Bonne et al., 2015; Steen-Larsen et al., 2015)…'

**P.12 L. 9-11**
'This value is consistent with the -0.4 to -0.6‰ %-1 range reported in the literature for vapor (Uemura et al., 2008; Pfahl and Wernli; Bonne et al.; 2014).'

**P. 12 L. 15-20**
'This is an interesting result with respect to the utility of Td, a measurable quantity, and is consistent with our earlier argument that Td is strongly related to hss. Both variables provide a better representation of source conditions than Tss and/or h2m. A low value of hss or Td corresponds to a strong influence of descending dry air within the PBL, which enhances kinetic isotopic fractionation and produces a high value of d. This mechanism explains the negative correlation between d and Td, and is expected for the relationship between d and hss. Alternatively, the vapor in descending air may have a high value of d (Fan, 2016), or both mechanisms may contribute to this result.'

**P. 12 L.28-29**
'…exhibited interannual, annual, and substantial inter-event variability.'

**P.12 L. 33**

'However, substantial intra-season variability occurred in both source and δ2H, indicating scatter in the seasonal relationship'

**P. 13 L. 14-16**
'The mechanisms identified, most notably the north-south migration of the vapor source region in phase with expansion and contraction of the Polar circulation cell, may also operate on times scales longer than that of our study, and may be a source of variation in isotopes measured in ice cores, pedogenic carbonates, and speleothems.'

**Caption of Fig 1**
'The figure indicates that some air parcels originate over land, but these were not included in calculations.'

**Caption of Fig 3**
'The spline fit, which highlights seasonal variations, explains 65% of variance in the data with a root mean squared error of 39.7‰. Of the three timescales, annual variability shows the greatest amplitude, though variability among events is also substantial.'

**Caption of Fig 4**
'The spline fits have R2 values of 0.60 and 0.19 for the δ2H and VLat respectively. For both datasets, the variability exhibited among events is of the same the order of magnitude as the seasonal variability.'

**Caption of Tab 1**
'Values of β are the partial coefficients of the regression and S.E. is the standard error.'
'Variation in δ2H is explained by a multiple linear regression (R2 = 0.54) of air parcel cooling during transport (ΔTcool), moisture source conditions (Td) 
[revised manuscript text omitted]

---

## Referee Report (RR1)

I would like to thank the authors for their detailed response to my first review. Most of my concerns have been addressed in a satisfactory way. However, I have three remaining points that need to be more carefully discussed in the manuscript and that I strongly recommend to consider before final publication:

1) The use of Td as a physically meaningful moisture source variable: the author's answer and the respective changes to the manuscript following my earlier comment 12.

Several statements with respect to the use of Td as a relevant moisture source variable are very confusing or physically wrong. I list my concerns below. I copied the authors' changed text (blue) and added my comments to it (green).

The use of Td and related discussion in this paper reflects that our group has done a substantial amount of work to model and understand isotopic variations in the marine boundary layer (manuscripts in preparation), and our understanding continues to improve. We realize that our discussion about Td in the earlier version was not clear, and it is valid for this reviewer to solicit further explanation. For clarity for the reader, we have completely rewritten section 3.2 that pertains to Td, and we hope the new discussion in the revised version is clearer.

In short, the idea of using Td is to indicate the moisture conditions within the PBL, as this is the moisture that forms the first condensate. This is different from the evaporative flux predicted by the Craig-Gordon model. In addition, the Craig-Gordon model does not consider effects of convection on vapor isotopic ratios in the PBL. However, convection is an important process that 1) transports PBL air to the free troposphere, and 2) brings dry air from aloft to the PBL. The boundary layer air is therefore a mixture of evaporated vapor from the ocean surface, and the dry air from aloft. The extent of this mixing within the PBL is reflected by (2m) dew point, Td. Td is also useful because it is directly related to relative humidity with respect to the sea surface temperature (h2m,SST), moreso than is the 2 m relative humidity. Indeed, when h2m, SST was used in the multiple regression instead of Td, it was a significant predictor of δ2H. However, in both variance explained and AIC, the multiple regression that incorporated Td performed better. Both because it performs better in the multiple regression, and because it is
a measurable quantity, we prefer Td to h2m,SST and have retained it in the paper.

Indeed the Craig-Gordon model does not parametrize effects of convection or boundary layer mixing but this is not its role. The extent of the boundary layer mixing is reflected implicitly by the 2 m dew point but similarly in all other humidity variables near the surface including the humidity gradient towards the ocean surface summarised by hsst (which uses the dew point at 2 m and the saturation vapour pressure at SST). The 2 m dew point is in essence equivalent to the absolute humidity of the air parcel and contains no more information than the specific humidity. Furthermore, Td at 2m is not more directly related to hsst than to h2m, the reference saturation specific humidity is a different one in the two cases. For me the only useful argument that should be mentioned in the text as to why Td 2m could be an interesting variable to look at, is that it can be directly measured. A better performance of Td in the author's regression framework alone is not a good argument for using it.

P. 9-10 L. 7-35, 1-9

'We prefer Td to the classical variables Tss and h for determining isotopic evaporative fluxes. This choice is based on our understanding that the meteorological variable Td characterizes the bulk vapor content and isotopic ratio of the marine PBL, independent of the vapor temperature. When advected to the free troposphere, it is this vapor that will form precipitation. Additionally, through equilibrium fractionation Td also determines the isotopic ratio of the first condensate at the LCL, where Rayleigh distillation begins.

This paragraph is very confusing: 1) It is not clear that Td is Td at 2m, 2) I do not agree that Td at 2m characterises the bulk vapour content and isotopic ratio of the marine PBL, this is a very much simplified view 3) Neither the temperature nor the dew point temperature along an air parcel trajectory can be assumed to be conserved. The humidity of the trajectory (and thus also the dew point temperature) changes due to mixing and rain out. The authors did not look at the first condensate along the trajectory in their analysis and did not consider rain out along the trajectory explicitly.

Within the marine PBL, several inter-related factors/processes are at work to determine the starting point of a Rayleigh trajectory. What is meant by Rayleigh trajectory?

The first is the isotopic flux of evaporation from the sea surface. Most studies estimate this flux using the classic model by Craig and Gordon (1965). In that model, three variables control the evaporative flux: the sea surface temperature, Tss, δ2H above the laminar layer, and the humidity hss above the laminar layer (e.g., at 2m), defined relative to Tss. In essence hss is a humidity gradient just reformulated and expressed in the form of a relative humidity (fraction). Additionally the diffusivity of 2H is needed in the Craig-Gordon model for the non-equilibrium fractionation factor (alphak) and depending on which formulation of alphak the 10 m wind speed. Also the isotope composition of the ocean water is needed but can be approximated to be constant at 0‰.

Though hss is not a measured quantity nor one that is normally modeled, it is determined by Td above the laminar layer and Tss. Not a directly measured quantity that is true but calculated from 2 directly measurable quantities (dew point temperature and SST). And it is of course normally modelled since all numerical models use it (although in the form of a gradient) in their surface latent heat flux parameterization.

Hence isotopic fluxes can be determined with the classical model using Tss, and Td and delta_2H above the laminar layer as input variables. From a physical point of view, Tss determines the amount of equilibrium fractionation at the water-air interface. Td and vapor δ2H, as well as Tss, control kinetic fractionation as vapor diffuses across the laminar layer. I agree with this.

It should be noted that when Tss is large, Td tends to be large as well, as a result of their change with latitude and season. This is a very general statement and might be true at long (>monthly) timescales but not at the event timescale. Td at 2 m varies strongly at the synoptic timescale whereas Tss does not.

Td and δ2H are also correlated, which will be discussed below. Therefore, all three variables controlling the evaporative flux, Tss, and hss and δ 2H above the laminar layer, are associated directly or indirectly with Td, making Td a good indicator of evaporation conditions.

The second process is convergence. At a moisture source location, low level air is moist due to evaporation near the sea surface. Convergence and uplift transports low-level moist air into the free troposphere where it mixes with dry, isotopically depleted air descending from surrounding regions resulting in strong humidity and temperature gradients near the sea surface (below 2 m). "Convergence" is misused in this context in my opinion.
In contrast, the specific humidity and isotopic ratios in the bulk of the PBL above 2m are relatively constant, resulting from the relative contributions of vertical transport of moist low-level and descending air (Fan, 2016). Td and δ2H at 2m both reflect the outcome of this mixing process, and so it follows that they are positively correlated. I agree that we expect positive correlation between Td at 2m at the source and δ2H. But I do not understand what the authors exactly mean to imply with process 2.

The third process is condensation at the LCL. The temperature of the air mass, which equals or is very slightly less than the local dew point, determines the amount of isotopic fractionation and thus the isotopic ratio of the first condensate. Why is the temperature of the air mass equal the local 2m dew point? What is meant by local? At the moisture source? So, do we have saturated conditions at the moisture source all the time? I would rather expect an air mass temperature that is higher than Td2m except in the case of fog/in a cloud. Td at LCL is not equal to Td2m unless the air parcel has not experienced any humidity change since its last passage in the boundary layer, which is very unrealistic.

It is this isotopic composition that defines the beginning of the Rayleigh part of the trajectory. Only Td;2m, not Tss nor h2m, is directly associated with the condensation temperature at the LCL (which differs only slightly from Td;2m due to the pressure difference between 2 m and the LCL and its effect on saturation specific humidity). This is confusing. What do the authors mean by the condensation temperature is directly associated with Td2m? Td2m and TdLCL are 2 different variables.

Since all three processes before Rayleigh distillation are either directly or indirectly related to Td, we consider Td a better indicator for the source conditions than either Tss or h. I do not agree with this statement. Process 1 is reflected in all moist variables, process 2 as far as I understood what the authors mean (boundary layer mixing) as well, and process 3 is in my opinion irrelevant in this discussion concerning the physical reasons for choosing Td as representing the moisture source conditions.
It is difficult, however, to theoretically assess the sensitivity of precipitation δ2H to variations in source Td, because this would require quantification of the theoretical relationship of Td to δ2H through each of the three processes and perhaps their combinations. We here report the first empirical sensitivity of 3.23‰ °C-1 (Table 1) for δ2H relative to Td. At the sea surface, for Tss between 0 and 25 °C, equilibrium fractionation as a function of temperature yields sensitivities between 1.1-1.6‰°C-1 (Majoube, 1971). However, a large part of this fractionation may be offset by condensation at the LCL. Consequently, the observed

sensitivity probably reflects primarily the fraction of vapor contributed by dry, isotopically depleted descending air that converges within the PBL. Mixing with the dry air causes a decrease in Td, which affects the δ2H of the PBL in two ways: 1) making the PBL air dry and isotopically depleted, and 2) isotopically depleting the evaporative flux by enhancing kinetic fractionation (an effect of low relative humidity). Both mechanisms produce a positive association between δ2H and Td, consistent with the sign of our observed partial coefficient (Table 1).' I do not agree with 2, δ2H of the evaporation flux (δ2He) becomes more enriched with decreasing δ2Hv due to the isotope gradient. See the Figure below, x-axis represents δ2Hv, y-axis δ2He from ocean evaporation as computed using the Craig-Gordon model, the equilibrium fractionation factor from Majoube, 1971, the non-equilibrium fractionation factor from Merlivat and Jouzel, 1979, a wind-speed of 6 ms-1 and a sea surface temperature of 15°C. In blue the δ2He(δ2Hv) relation for a hsst of 80%, in black hsst=60% and in red hsst=40%. The lines intersect at δ2Hv=~-83‰ which is the equilibrium vapour equivalent of ocean water (0‰). In this situation (δ2Hv=~-83‰), there is no isotope gradient or humidity gradient effect.

[Figure]

I recommend careful revision of this text. In my opinion it can also be shortened substantially. The authors need to make it clear that a) Td at 2m is used, b) that Td at 2m is not necessarily equivalent to Td at LCL and along the trajectory, the discussion around Td at LCL is not relevant in this part which focuses on the moisture source processes and not the transport and rain out along the air parcel's trajectory, c) Td at 2m is used because it is a measureable quantity, equivalent to using specific humidity at 2m and as such it partly reflects the classically used moisture source parameters. The physical process linking Td and hsst being that strong ocean evaporation occurs when there is a strong humidity gradient towards the ocean surface, that is when hsst is low. A strong humidity gradient at the synoptic time scale is very often achieved through advection of cold dry air over the ocean surface, that is when specific humidity at 2m is low as well.

2) The simplifications involved in the used moisture source diagnostics compared to the more detailed method of Sodemann et al. 2008 should be mentioned explicitly in the manuscript (my previous comment 4): 1) The method adopted by the authors assumes that an air parcel is not further back-traceable once it has been located in the boundary layer, 2) the authors assume very strong mixing in the boundary layer and a dominant effect of recent evaporation on the humidity in the boundary layer (since all humidity taken up by the trajectory is assumed to have been evaporated in the last model data time step and at this location), 3) This strongly enhances moisture sources that are located close to the the measurement site and neglects more remote source locations. These 3 points should be mentioned in the manuscript.

   Only considering the latest passage of an air parcel in the boundary layer would account for 5-30% in rare cases up to 70% of the final specific humidity of an air parcel in the framework of Sodemann et al. 2008. Even if one argues that a trajectory is not further back-traceable once it has been in the boundary layer it is rather simplistic to assume that the air parcel has taken up all its humidity from surface evaporation at its latest position in the boundary layer.

3) The proposed method for defining the starting points of the trajectories at the measurement site contains a conceptual gap (see my previous comments 8 and 9): the observational and model worlds are interweaved without a thorough validation. It is not in the scope of this paper to show that the used reanalysis data show realistic condensation rate profiles at the observational site. But it should be clearly stated that it is assumed that the reanalysis' data representation of the rain out process during an event is consistent with the observations from the cloud radar. For me this is an important source of uncertainty in the presented method and should at least be explicitly mentioned.

After the requested changes with respect to the assumptions and implications of the chosen method have been made, I recommend publication of this overall very nice and interesting manuscript.

---

## Author Response (AR2)

**Authors' response to reviewer's minor revisions**

**Journal**: Atmospheric Chemistry and Physics
**Manuscript #:** acp-2016-539
**Title**: Annual variation in event-scale precipitation $\delta^2$H at Barrow, AK reflects vapor source region
**Authors**: Annie Putman, Xiahong Feng, Leslie J. Sonder, and Eric S. Posmentier
**Date**: Feb. 23, 2017

Dear Dr. Thomas Röckmann,

We are pleased to resubmit for publication the revised version of MS acp-2016-539, 'Annual variation in event-scale precipitation $\delta^2$H at Barrow, AK reflects vapor source region'. We appreciate the follow-up from the second reviewer to our first response. The reviewer's challenges and criticisms have helped us better express our thinking, particularly with respect to the use of 2 meter dew point as an indicator of evaporation conditions. Though the reviewer may not agree with our use of dew point as an indicator of the vapor source conditions, the authors feel that the articulation of the utility of dew point as a vapor source indicator is much improved in this iteration of the paper.

A point-by-point response to the comments of the reviewer are followed by a marked up manuscript. Note that line numbers in this document refer to the final manuscript. We appreciate your time and consideration as you evaluate our further revised manuscript and we look forward to hearing from you soon.

Kind Regards,

Annie Putman

Response to Reviewer #2 minor revisions
Feb 23, 2017

I would like to thank the authors for their detailed response to my first review. Most of my concerns have been addressed in a satisfactory way. However, I have three remaining points that need to be more carefully discussed in the manuscript and that I strongly recommend to consider before final publication:

*We appreciate the effort this reviewer has put into critically reading this manuscript. The reviewer's constructive comments have made our thinking and writing clearer. In reading the reviews, we realized that the revised manuscript still contained unclear or misleading statements, though not physically wrong. Therefore, additional revisions are made in response to this reviewer's comments. In the following our responses are in italicized red and the new text in plain red.*

1) The use of Td as a physically meaningful moisture source variable: the author's answer and the respective changes to the manuscript following my earlier comment 12.

*Our general response to 1):*

*We listed three processes that potentially affect the isotopic compositions of air mass in the PBL and of the first condensate at the LCL. The first one is evaporation. For this, the reviewer did not dispute, but thinks that Td at 2 m is no better than classical variables particularly hss. She/he does agree with the merit of Td at 2 m as a directly measurable quantity. The second process is mixing within the PBL. The reviewer does not disagree with us, but still thinks that hss can do the job just as well. The third process is condensation at the LCL. We think that the reviewer did not understand our argument, partly due to our lack of clarity. We have tried to make this clear in the new version.*

Several statements with respect to the use of Td as a relevant moisture source variable are very confusing or physically wrong. I list my concerns below. I copied the authors' changed text (blue) and added my comments to it (green).

The use of Td and related discussion in this paper reflects that our group has done a substantial amount of work to model and understand isotopic variations in the marine boundary layer (manuscripts in preparation), and our understanding continues to improve. We realize that our discussion about Td in the earlier version was not clear, and it is valid for this reviewer to solicit further explanation. For clarity for the reader, we have completely rewritten section 3.2 that pertains to Td, and we hope the new discussion in the revised version is clearer.

In short, the idea of using Td is to indicate the moisture conditions within the PBL, as this is the moisture that forms the first condensate. This is different from the evaporative flux predicted by the Craig-Gordon model. In addition, the Craig-Gordon model does not consider effects of convection on vapor isotopic ratios in the PBL. However, convection is an important process that 1) transports PBL air to the free troposphere, and 2) brings dry air from aloft to the PBL. The boundary layer air is therefore a mixture of evaporated vapor from the ocean surface, and the dry air from aloft. The extent of this mixing within the PBL is reflected by (2m) dew point, Td. Td is also useful because it is directly related to relative humidity with respect to the sea surface temperature (h2m,SST), moreso than is the 2 m relative humidity. Indeed, when h2m, SST was used in the multiple regression instead of Td, it was a significant predictor of δ2H. However, in both variance explained and AIC, the multiple regression that incorporated Td performed better. Both because it performs better in the multiple regression, and because it is a measurable quantity, we prefer Td to h2m,SST and have retained it in the paper.

Indeed the Craig-Gordon model does not parametrize effects of convection or boundary layer mixing but this is not its role. The extent of the boundary layer mixing is reflected implicitly by the 2 m dew point but similarly in all other humidity variables near the surface including the humidity gradient

towards the ocean surface summarised by hsst (which uses the dew point at 2 m and the saturation vapour pressure at SST). The 2 m dew point is in essence equivalent to the absolute humidity of the air parcel and contains no more information than the specific humidity. Furthermore, Td at 2m is not more directly related to hsst than to h2m, the reference saturation specific humidity is a different one in the two cases. For me the only useful argument that should be mentioned in the text as to why Td 2m could be an interesting variable to look at, is that it can be directly measured. A better performance of Td in the author's regression framework alone is not a good argument for using it.

*Thank you for the feedback. The comments and perspectives from this response have been incorporated into the revised version. In particular, noting that Td at 2m is directly measurable. Other aspects of this response have been covered in the specific comments below.*

P. 9-10 L. 7-35, 1-9
'We prefer Td to the classical variables Tss and h for determining isotopic evaporative fluxes. This choice is based on our understanding that the meteorological variable Td characterizes the bulk vapor content and isotopic ratio of the marine PBL, independent of the vapor temperature. When advected to the free troposphere, it is this vapor that will form precipitation. Additionally, through equilibrium fractionation Td also determines the isotopic ratio of the first condensate at the LCL, where Rayleigh distillation begins.
This paragraph is very confusing: 1) It is not clear that Td is Td at 2m, 2) I do not agree that Td at 2m characterises the bulk vapour content and isotopic ratio of the marine PBL, this is a very much simplified view 3) Neither the temperature nor the dew point temperature along an air parcel trajectory can be assumed to be conserved. The humidity of the trajectory (and thus also the dew point temperature) changes due to mixing and rain out. The authors did not look at the first condensate along the trajectory in their analysis and did not consider rain out along the trajectory explicitly.

*1) Td is changed to Td at 2m. 2) Yes, simplified yet largely correct. We do refer to the section of later discussions, and the reader can determine how simplified it is. As discussed in the paper, Td represents many processes in the PBL, and so the simplification is unavoidable. 3) We are not sure we understand this point. There is no rain out involved from the surface to the LCL.*

P. 9 L. 16-20
Our multiple linear regression attributes a substantial fraction of the variance in $\delta^2H$ to variations in $\bar{T}_d$ at 2 m (10.5%, Table 1). $\bar{T}_d$ at 2 m is used to indicate conditions at the vapor source, and is preferred to the classical variables $T_{SS}$, $h$, and the humidity $h_{SS}$ above the laminar layer (e.g., at 2 m), defined relative to $T_{SS}$. We prefer $T_d$ at 2 m because it is directly measurable and integrates three processes that determine the isotopic ratio of the first condensate at the lifted condensation level (LCL), where Rayleigh distillation begins.

Within the marine PBL, several inter-related factors/processes are at work to determine the starting point of a Rayleigh trajectory. What is meant by Rayleigh trajectory?

*Rayleigh distillation is more accurate, so the text has been changed, see above.*

The first is the isotopic flux of evaporation from the sea surface. Most studies estimate this flux using the classic model by Craig and Gordon (1965). In that model, three variables control the evaporative flux: the sea surface temperature, Tss, δ2H above the laminar layer, and the humidity hss above the laminar layer (e.g., at 2m), defined relative to Tss. In essence hss is a humidity gradient just reformulated and expressed in the form of a relative humidity (fraction). Additionally the diffusivity of 2H is needed in the Craig-Gordon model for the non-equilibrium fractionation factor (alphak) and depending on which formulation of alphak the 10 m wind speed. Also the isotope composition of the ocean water is needed but can be approximated to be constant at 0‰.

*We agree, and added 'primarily' to establish that the mentioned variables are a subset of all variables in the equation.*

P. 9 L. 21-24

The first process that determines the isotopic ratio of the first condensate is the isotopic flux of evaporation from the sea surface. The classical model by Craig and Gordon (1965) estimates the evaporative flux primarily using the sea surface temperature, $T_{ss}$ and the humidity $h_{ss}$ above the laminar layer (e.g., at 2 m), defined relative to $T_{ss}$, and $\delta^2 H$ above the laminar layer

Though hss is not a measured quantity nor one that is normally modeled, it is determined by Td above the laminar layer and Tss. Not a directly measured quantity that is true but calculated from 2 directly measurable quantities (dew point temperature and SST). And it is of course normally modelled since all numerical models use it (although in the form of a gradient) in their surface latent heat flux parameterization.

*This text has been changed to pointing out the relationship between hss and Td, and the relationship of Td and Tss.*

P. 9 L. 24-27

$T_d$ and $h_{ss}$ are related through the specific humidity and exhibit a correlation coefficient of 0.67 in our dataset. Likewise, $T_d$ and $T_{ss}$ are related on monthly and longer timescales, and exhibit a correlation coefficient of 0.46. The second process, described below, relates $T_d$ to $\delta^2 H$ above the laminar layer. Because $T_d$ is related to $h_{ss}$, $T_{ss}$ and $\delta^2 H$ above the laminar layer, it is a good proxy for the isotopic flux of evaporation.

Hence isotopic fluxes can be determined with the classical model using Tss, and Td and delta_2H above the laminar layer as input variables. From a physical point of view, Tss determines the amount of equilibrium fractionation at the water-air interface. Td and vapor δ2H, as well as Tss, control kinetic fractionation as vapor diffuses across the laminar layer. I agree with this.
It should be noted that when Tss is large, Td tends to be large as well, as a result of their change with latitude and season. This is a very general statement and might be true at long (>monthly) timescales but not at the event timescale. Td at 2 m varies strongly at the synoptic timescale whereas Tss does not.

*This is true, and the text has been changed to point out this timescale. See text above for new text.*

Td and δ2H are also correlated, which will be discussed below. Therefore, all three variables controlling the evaporative flux, Tss, and hss and δ 2H above the laminar layer, are associated directly or indirectly with Td, making Td a good indicator of evaporation conditions.

The second process is convergence. At a moisture source location, low level air is moist due to evaporation near the sea surface. Convergence and uplift transports low-level moist air into the free troposphere where it mixes with dry, isotopically depleted air descending from surrounding regions resulting in strong humidity and temperature gradients near the sea surface (below 2 m).
"Convergence" is misused in this context in my opinion.

*We recognize that using the word "convergence" without describing the PBL model developed and used in our group is unclear to readers. For greater clarity, the text has been simplified to mixing within the PBL, without specification of the mechanism driving the mixing.*

P. 9 L. 28

The second process is mixing of moist air near the ocean surface with drier, isotopically depleted

descending air (Fan, 2016).

In contrast, the specific humidity and isotopic ratios in the bulk of the PBL above 2m are relatively constant, resulting from the relative contributions of vertical transport of moist low-level and descending air (Fan, 2016). Td and δ2H at 2m both reflect the outcome of this mixing process, and so it follows that they are positively correlated. I agree that we expect positive correlation between Td at 2m at the source and δ2H. But I do not understand what the authors exactly mean to imply with process 2.

*The authors are expressing that both Td at 2m and d2H are affected by mixing within the PBL, so Td at 2m is a good proxy for d2H in the majority of the PBL, excepting the portion closest to the ocean surface which is very moist. Changes to the text have been made to clarify the reasoning.*

P. 9 L. 29-32

Mixing within the planetary boundary layer results in strong humidity and temperature gradients near the sea surface (well below 2 m), with more uniform specific humidity and isotopic ratios in the PBL above 2 m. The values $T_d$ and $\delta^2 H$ at 2 m reflect the relative proportion of the dry and isotopically depleted air in the PBL, so they are positively correlated. Therefore, $T_d$ at 2 m is also a proxy for the isotopic ratio of the air in the PBL.

The third process is condensation at the LCL. The temperature of the air mass, which equals or is very slightly less than the local dew point, determines the amount of isotopic fractionation and thus the isotopic ratio of the first condensate. Why is the temperature of the air mass equal the local 2m dew point? What is meant by local? At the moisture source? So, do we have saturated conditions at the moisture source all the time? I would rather expect an air mass temperature that is higher than Td2m except in the case of fog/in a cloud. Td at LCL is not equal to Td2m unless the air parcel has not experienced any humidity change since its last passage in the boundary layer, which is very unrealistic.

*See below.*

It is this isotopic composition that defines the beginning of the Rayleigh part of the trajectory. Only Td;2m, not Tss nor h2m, is directly associated with the condensation temperature at the LCL (which differs only slightly from Td;2m due to the pressure difference between 2 m and the LCL and its effect on saturation specific humidity). This is confusing. What do the authors mean by the condensation temperature is directly associated with Td2m? Td2m and TdLCL are 2 different variables.

*Given the questions asked by the reviewer, the section describing process three is clearly confusing. We did not mean to imply that Td equals or is TLCL, but rather that they are strongly related. The section has been re-written to address the confusing statements brought up by the reviewer. In general, we see a clear and strong linear relationship in our dataset between Td and TLCL, which is about 1:1 for all but the coldest temperatures.*

P. 9-10 L. 33-34, 1-2

The third process is condensation at the LCL. The temperature of the air mass at the LCL determines the amount of isotopic fractionation and thus the isotopic ratio of the first condensate. Of the vapor source variables, $T_d$ at 2 m, not $T_{ss}$ nor $h_{ss}$, is strongly related to the condensation temperature at the LCL. On an event scale, $T_d$ at 2 m and $T_{LCL}$ are correlated with a coefficient of 0.71.

Since all three processes before Rayleigh distillation are either directly or indirectly related to Td, we consider Td a better indicator for the source conditions than either Tss or h. I do not agree with this statement. Process 1 is reflected in all moist variables, process 2 as far as I understood what the authors mean (boundary layer mixing) as well, and process 3 is in my opinion irrelevant in this discussion concerning the physical reasons for choosing Td as representing the moisture source conditions.

*The idea is that because all three processes are related to Td, that Td is a good indicator of the vapor*

*source conditions. In particular, Td has a more consistent relationship to TLCL (process 3) than hss or Tss, which is relevant as TLCL is the temperature where the first condensate forms. The relevance of the first condensate is that it's the starting point for Rayleigh distillation.*

It is difficult, however, to theoretically assess the sensitivity of precipitation δ2H to variations in source Td, because this would require quantification of the theoretical relationship of Td to δ2H through each of the three processes and perhaps their combinations. We here
report the first empirical sensitivity of 3.23‰ °C-1 (Table 1) for δ2H relative to Td. At the sea surface, for Tss between 0 and 25 °C, equilibrium fractionation as a function of temperature yields sensitivities between 1.1-1.6‰°C-1 (Majoube, 1971). However, a large part of this fractionation may be offset by condensation at the LCL. Consequently, the observed sensitivity probably reflects primarily the fraction of vapor contributed by dry, isotopically depleted descending air that converges within the PBL. Mixing with the dry air causes a decrease in Td, which affects the δ2H of the PBL in two ways: 1) making the PBL air dry and isotopically depleted, and 2) isotopically depleting the evaporative flux by enhancing kinetic fractionation (an effect of low relative humidity). Both mechanisms produce a positive association between δ2H and Td, consistent with the sign of our observed
partial coefficient (Table 1).' I do not agree with 2, δ2H of the evaporation flux (δ2He) becomes more enriched with decreasing δ2Hv due to the isotope gradient. See the Figure below, x-axis represents δ2Hv, y-axis δ2He from ocean evaporation as computed using the Craig-Gordon model, the equilibrium fractionation factor from Majoube, 1971, the
non-equilibrium fractionation factor from Merlivat and Jouzel, 1979, a wind-speed of 6 ms-1 and a sea surface temperature of 15°C. In blue the δ2He(δ2Hv) relation for a hsst of 80%, in black hsst=60% and in red hsst=40%. The lines intersect at δ2Hv=~-83‰ which is the equilibrium vapour equivalent of ocean water (0‰). In this situation (δ2Hv=~-83‰), there is no isotope gradient or humidity gradient effect.

[Figure]

*Thanks for the computation! The reviewer is correct if the vapor is more enriched from the vapor in equilibrium with the ocean water. But it almost never happens. When the vapor is more depleted than the equilibrium vapor, then the overall effect depends on which one is greater, the humidity or the isotopic ratio. Our experience is that the humidity effect is often more dominant. This is reflected in a clarifying comment in the text.*

P.10 L. 3-15
Because the three processes that determine the isotope ratio of the first condensate before Rayleigh

distillation are either directly or indirectly related to $T_d$ at 2 m, and $T_d$ at 2 m is directly measurable, we consider $T_d$ a better indicator for the source conditions than either $T_{SS}$ or $h_{SS}$. It is difficult, however, to theoretically assess the sensitivity of precipitation $\delta^2H$ to variations in source $T_d$ at 2 m, because this would require quantification of the theoretical relationship of $T_d$ to $\delta^2H$ through each of the three processes and their combinations. We report here the first empirical sensitivity of 3.23‰ °C$^{-1}$ (Table 1) for $\delta^2H$ relative to $T_d$ at 2 m. For $T_{SS}$ between 0 and 25 °C, equilibrium fractionation yields sensitivities between 1.1-1.6 ‰°C$^{-1}$ (Majoube, 1971). However, condensation at the LCL likely offsets most of the fractionation that occurred during evaporation at the sea surface. Consequently, the observed sensitivity likely reflects the fraction of vapor contributed by dry, isotopically depleted air that mixes in the PBL. Mixing with the dry air causes a decrease in $T_d$, which affects the $\delta^2H$ of the PBL in two ways: 1) making the PBL air dry and isotopically depleted, and 2) isotopically depleting the evaporative flux by enhancing kinetic fractionation (low relative humidity makes evaporative flux isotopically depleted and low isotopic ratios of ambient air makes it enriched, but the former often out competes the latter (Fan, 2016)). Both mechanisms produce a positive association between $\delta^2H$ and $T_d$, consistent with the sign of our observed partial coefficient (Table 1).

I recommend careful revision of this text. In my opinion it can also be shortened substantially. The authors need to make it clear that a) Td at 2m is used, b) that Td at 2m is not necessarily equivalent to Td at LCL and along the trajectory, the discussion around Td at LCL is not relevant in this part which focuses on the moisture source processes and not the transport and rain out along the air parcel's trajectory, c) Td at 2m is used because it is a measureable quantity, equivalent to using specific humidity at 2m and as such it partly reflects the classically used moisture source parameters. The physical process linking Td and hsst being that strong ocean evaporation occurs when there is a strong humidity gradient towards the ocean surface, that is when hsst is low. A strong humidity gradient at the synoptic time scale is very often achieved through advection of cold dry air over the ocean surface, that is when specific humidity at 2m is low as well.

*We agree with most comments from this reviewer and have made changes accordingly. Though efforts have been made to condense the text, we feel that our hypothesis that Td is a source condition indicator should be fully articulated. With the help and challenge of this reviewer, we hope that this version does a better job explaining our reasoning. The future work will continue to test if our hypothesis is a good one and if so what affects the Td~d2H relationship quantitatively. Though the revised text may not be as short as this reviewer would like, but we hope the changes are sufficient.*

2) The simplifications involved in the used moisture source diagnostics compared to the more detailed method of Sodemann et al. 2008 should be mentioned explicitly in the manuscript (my previous comment 4): 1) The method adopted by the authors assumes that an air parcel is not further back-traceable once it has been located in the boundary layer, 2) the authors assume very strong mixing in the boundary layer and a dominant effect of recent evaporation on the humidity in the boundary layer (since all humidity taken up by the trajectory is assumed to have been evaporated in the last model data time step and at this location), 3) This strongly enhances moisture sources that are located close to the measurement site and neglects more remote source locations. These 3 points should be mentioned in the manuscript. Only considering the latest passage of an air parcel in the boundary layer would account for 5-30% in rare cases up to 70% of the final specific humidity of an air parcel in the framework of Sodemann et al. 2008. Even if one argues that a trajectory is not further back-traceable once it has been in the boundary layer it is rather simplistic to assume that the air parcel has taken up all its humidity from surface evaporation at its latest position in the boundary layer.

*These are all valid points. We have added some caveats to the methods section to cover the ramifications of simplifying the vapor source identification process relative to Sodemann et al 2008.*

P.4 L.11-12
However, it assumes strong mixing in the PBL such that the PBL reflects recent evaporation conditions. As well, the method may bias the results towards vapor sources proximal to the sampling site.

3) The proposed method for defining the starting points of the trajectories at the measurement site contains a conceptual gap (see my previous comments 8 and 9): the observational and model worlds are interweaved without a thorough validation. It is not in the scope of this paper to show that the used reanalysis data show realistic condensation rate profiles at the observational site. But it should be clearly stated that it is assumed that the reanalysis' data representation of the rain out process during an event is consistent with the observations from the cloud radar. For me this is an important source of uncertainty in the presented method and should at least be explicitly mentioned.

*Yes. This is the topic of a different paper, but a comment has been added to the Methods section to alert readers to this potential source of error.*

P.4 L. 25-27
This method assumes that the reanalysis' data's spatial and temporal representation of a precipitation event is consistent with the observations from the cloud radar. An analysis of this relationship is described in Putman (2013).

After the requested changes with respect to the assumptions and implications of the chosen method have been made, I recommend publication of this overall very nice and interesting manuscript.

[revised manuscript text omitted]